# Spatial joint profiling of DNA methylome and transcriptome in tissues

Chin Nien Lee[1,2,7 ✉], Hongxiang Fu[1,3,4,7], Angelysia Cardilla[1,4], Wanding Zhou[1,3 ✉] & Yanxiang Deng[1,5,6 ✉]

The spatial resolution of omics analyses is fundamental to understanding tissue biology[1–3]. The capacity to spatially profile DNA methylation, which is a canonical epigenetic mark extensively implicated in transcriptional regulation[4,5], is lacking. Here we introduce a method for whole-genome spatial co-profiling of DNA methylation and the transcriptome of the same tissue section at near single-cell resolution. Applying this technology to mouse embryogenesis and the postnatal mouse brain resulted in rich DNA–RNA bimodal tissue maps. These maps revealed the spatial context of known methylation biology and its interplay with gene expression. The concordance and distinction in spatial patterns of the two modalities highlighted a synergistic molecular definition of cell identity in spatial programming of mammalian development and brain function. By integrating spatial maps of mouse embryos at two different developmental stages, we reconstructed the dynamics that underlie mammalian embryogenesis for both the epigenome and transcriptome, revealing details of sequence-, cell-type- and region-specific methylation-mediated transcriptional regulation. This method extends the scope of spatial omics to include DNA cytosine methylation, enabling a more comprehensive understanding of tissue biology across development and disease.

DNA methylation is a key epigenetic mechanism that modulates gene expression and lineage specification by regulating chromatin structure and accessibility to transcriptional machinery[4,5]. This epigenetic modification varies dynamically across cell types, developmental stages and environmental conditions[6]. Abnormal cytosine-methylation patterns are associated with various diseases, including cancer[7], autoimmune diseases[8] and inflammation[9]. Furthermore, altered DNA-methylation levels have been observed in ageing tissues[10], contributing to biomarkers and predictive clock models for chronological and biological age[11].

Despite advances in single-cell methylome analysis[5,12], the lack of spatial information regarding intact tissues limits our understanding of gene regulation during tissue development and disease progression. Emerging spatial multi-omics technologies now enable profiling of molecular features in native tissue microenvironments, significantly enhancing our understanding of biological complexity in situ. However, current spatial methods are limited to histone modifications, chromatin accessibility, transcriptomes and selected protein panels[1–3]. Direct spatial mapping of DNA methylation has not been available, leaving this crucial epigenetic layer underexplored.

Here we introduce a technology for spatial joint profiling of the DNA methylome and transcriptome (spatial-DMT) on the same tissue section at near single-cell resolution. We used spatial-DMT to profile mouse embryos and postnatal mouse brains. The spatial maps uncovered intricate spatiotemporal regulatory mechanisms of gene expression in a native tissue context. Spatial-DMT investigates interactive molecular hierarchies in development, physiology and pathogenesis in a spatially resolved manner.

## Spatial-DMT design and workflow

Spatial-DMT combines microfluidic in situ barcoding[1], cytosine deamination conversion[12] and high-throughput next-generation sequencing to achieve spatial methylome profiling directly in tissue. The experimental workflow of spatial-DMT is illustrated in Fig. 1a,b, with a step-by-step protocol provided in Extended Data Fig. 1 and detailed reagent information provided in Supplementary Tables 1–4.

Specifically, hydrochloric acid (HCl) was applied to fixed frozen tissue sections to disrupt the nucleosome structures and remove histones to improve Tn5 transposome accessibility for DNA-methylation profiling[13]. Next, Tn5 transposition was performed and adapters containing a universal ligation linker were inserted into genomic DNA (gDNA). To further reduce the size of the large gDNA fragment and improve yield, we adopted a multi-tagmentation strategy as previously described[13]. Specifically, we implemented two rounds of tagmentation to balance DNA yield with experimental time and minimize the risk of RNA degradation. mRNAs were then captured by the biotinylated reverse transcription primer (poly-biotinylated deoxythymidine (dT) primer with unique molecular identifiers (UMIs) and a universal linker

[1]Department of Pathology and Laboratory Medicine, University of Pennsylvania, Philadelphia, PA, USA. [2]Institute of RNA innovation, Perelman School of Medicine, University of Pennsylvania, Philadelphia, PA, USA. [3]Center for Computational and Genomic Medicine, The Children's Hospital of Philadelphia, Philadelphia, PA, USA. [4]Department of Bioengineering, University of Pennsylvania, Philadelphia, PA, USA. [5]Epigenetics Institute, Perelman School of Medicine, University of Pennsylvania, Philadelphia, PA, USA. [6]Institute of Aging, Perelman School of Medicine, University of Pennsylvania, Philadelphia, PA, USA. [7]These authors contributed equally: Chin Nien Lee, Hongxiang Fu. ✉e-mail: chinnien.lee@pennmedicine.upenn.edu; wanding.zhou@pennmedicine.upenn.edu; yanxiang.deng@pennmedicine.upenn.edu

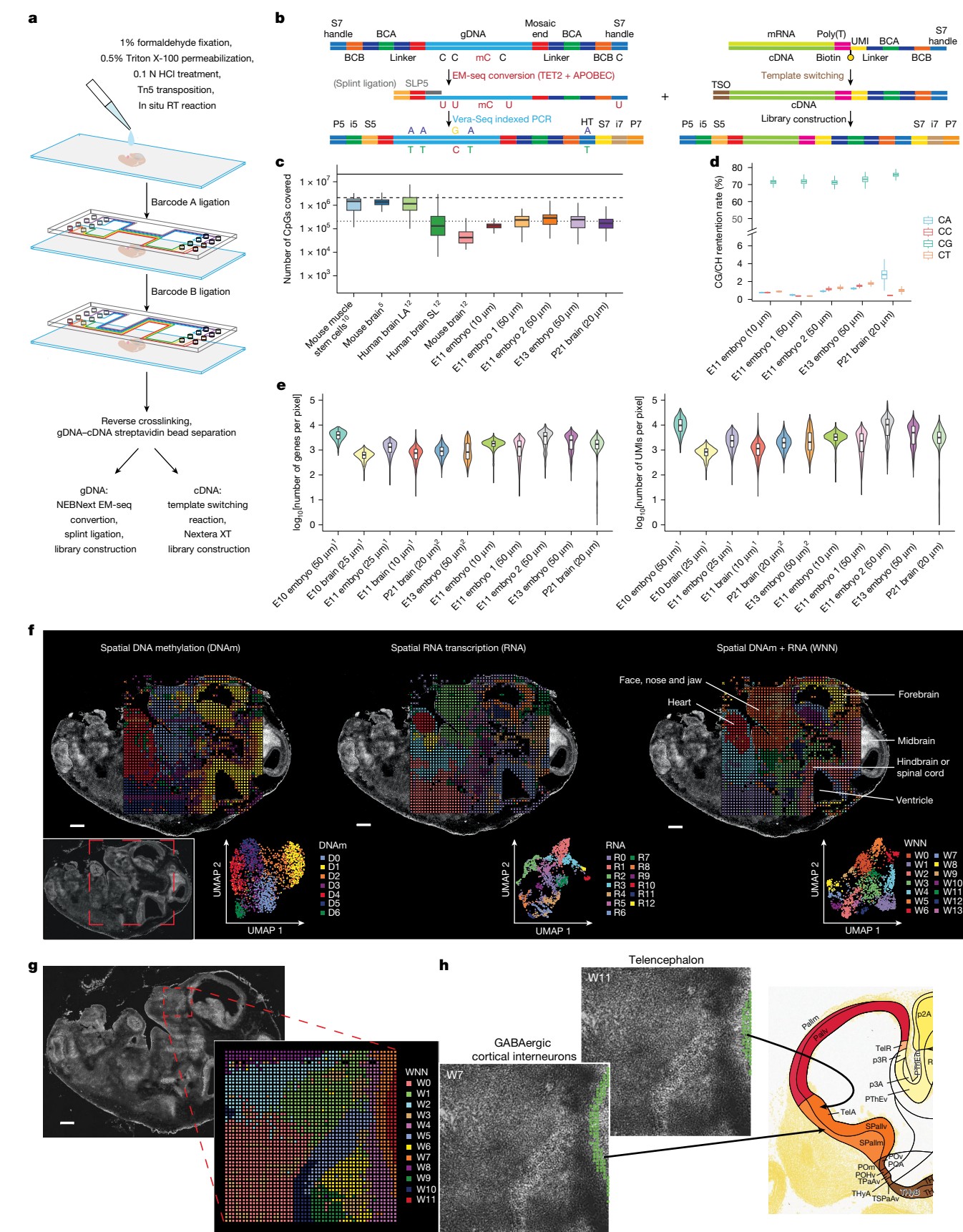

**Fig. 1 | See next page for caption.**

**Fig. 1 | Overview of co-profiling of DNA methylation and transcriptome in tissues. a**, Spatial-DMT workflow. A tissue section is fixed, permeabilized, treated with HCl treatment, and subjected to Tn5 transposition and reverse transcription (RT). Spatial barcodes are sequentially ligated, after which cDNA and gDNA are separated using streptavidin beads. cDNA is processed using template switching, after which a cDNA library is prepared; gDNA is processed using enzymatic methyl-seq (EM-seq) conversion and splint ligation, after which a library is constructed. **b**, Chemistry workflow of DNA methylation and RNA library preparation. BCA, barcode A; BCB, barcode B; TSO, template switch oligo. **c**, Comparison of the number of CpGs per tissue pixel per cell across spatial-DMT and other single-cell DNA-methylation datasets[5,10,12,14] (number of cells: mouse muscle stem cells[10], $n$ = 260; mouse brain[5], $n$ = 103,560; human brain LA[12], $n$ = 1,049; human brain SL[12], $n$ = 1,920; mouse brain[12], $n$ = 491; number of pixels: E11 embryo (10 µm), $n$ = 2,493; E11 embryo 1 (50 µm), $n$ = 1,954; E11 embryo 2 (50 µm), $n$ = 1,947; E13 embryo (50 µm), $n$ = 1,699; P21 brain (20 µm), $n$ = 2,235). The solid line indicates the maximum number of CpGs in the mouse genome, large dashed line indicates 10% of the total number of CpGs and small dashed line indicates 1% of the total number of CpGs. **d**, Box plots showing CG/CH retention rates in E11 and E13 embryos and P21 brain tissues. Number of pixels are as stated in part **c** for these datasets. **e**, Number of genes (left) and UMIs (right) per pixel. For data from previous studies[1,2], the numbers of pixels are as follows (left to right): ref. 1, $n$ = 901, 1,789, 1,837, 1,840; ref. 2, $n$ = 2,373, 2,187. For the other datasets, number of pixels are as stated in part **c**. The box plots show the median (centre line), first and third quartiles (box limits) and

1.5× interquartile range (whiskers). **f**, Spatial clusters (top) and UMAP analysis (bottom) of DNA methylation (DNAm, left), RNA transcription (RNA, middle) and integrated data (spatial DNAm + RNA (WNN), right) in E11 mouse embryo (pixel size, 50 µm; ROI (red dashed box), 5 × 5 mm²; technical replicates, $n$ = 2). Integrated analysis reveals more-refined spatial clusters and distinct anatomical regions, including brain, spinal cord, heart and craniofacial regions. Scale bars, 500 µm. D, DNA; R, RNA; W, WNN. **g**, Spatial mapping and joint clustering of DNA methylation and RNA data in E11 facial and forebrain regions (pixel size, 10 µm; ROI (red dashed box), 1 × 1 mm²). Scale bar, 500 µm. **h**, Spatial mapping of GABAergic cortical interneurons and telencephalon cells on the basis of deconvolution of transcriptomic pixels using a scRNA-seq reference[25]. Cell-type distributions align with the anatomic references from the Allen developing mouse atlases[55]. Pallm, mantle zone of the pallium; Pallv, ventricular zone of the pallium; POA, preoptic alar plate; POHv, ventricular zone of preopto-hypothalamic band; POm, mantle zone of preoptic area; POv, ventricular zone of preoptic area; PThEm, mantle zone of prethalamic eminence; PThEv, ventricular zone of prethalamic eminence; p2A, alar plate of prosomere 2; p3A, alar plate of prosomere 3; p3R, roof plate of prosomere 3; SPallm, mantle zone of the subpallium; SPallv, ventricular zone of the subpallium; TelA, alar plate of evaginated telencephalic vesicle; TelR, roof plate of evaginated telencephalic vesicle; TH, thalamus; THyA, alar part of terminal hypothalamus; THyB, basal part of terminal hypothalamus; TPaAv, ventricular zone of TPaA (terminal paraventricular area of THyA); TSPaAv, ventricular zone of TSPaA (terminal subparaventricular area of THyA).

sequence; Supplementary Table 1), followed by reverse transcription to synthesize the complementary DNA (cDNA). Both genomic fragments and cDNA in the tissue were then ligated to spatial barcodes sequentially. In brief, two sets of spatial barcodes (barcodes A1–A50 and B1–B50; Supplementary Tables 2 and 3) flowed perpendicularly to each other in the microfluidic channels. They were covalently conjugated to the universal linker through the templated ligation. This results in a two-dimensional grid of spatially barcoded tissue pixels in the region of interest (ROI), each defined by the unique combination of barcodes A$i$ and B$j$ ($i$ = 1–50, $j$ = 1–50; barcoded pixels, $n$ = 2,500). Barcoded gDNA fragments and cDNA were then released after reverse crosslinking. Afterwards, the biotin-labelled cDNA was enriched by the streptavidin beads and separated from the gDNA-containing supernatant. Subsequently, cDNA was subjected to a template switching reaction and the cDNA library was constructed (Fig. 1b, right, and Extended Data Fig. 1). gDNA was treated with enzymatic methyl-sequencing (EM-seq) conversion, splint ligation and DNA library construction (Fig. 1b, left, Extended Data Fig. 1 and Methods).

To minimize DNA damage during methylome profiling, we used enzymatic methyl-seq, an enzyme-based alternative to bisulfite conversion (Fig. 1b and Extended Data Fig. 1). In this process, modified cytosines (the sum of 5-methylcytosine and 5-hydroxymethylcytosine) were both oxidized by the ten–eleven translocation methylcytosine dioxygenase 2 (TET2) protein and protected from deamination by the APOBEC protein, whereas unmodified cytosines were deaminated to uracil. After the conversion, a SLP5 adapter with ten random H (A, C or T) nucleotides was ligated to the 3′ end of the gDNA, adding the other PCR handle[12]. The gDNA fragments were then amplified using uracil-literate VeraSeq Ultra DNA polymerase. Because C residues on the PCR handle of barcode B were changed to T residues after deamination and PCR, we modified the P7 primer (N70X-HT) by replacing C residues with T residues to complement the A residues on the PCR handle and amplify the fragments. Barcodes have been designed such that no crosstalk under C-to-T deamination is possible.

To demonstrate the ability of spatial-DMT to co-profile DNA methylation and transcription in complex tissues, we profiled mouse embryos on embryonic days 11 (E11) and 13 (E13) and mouse brain on postnatal day 21 (P21). Mouse embryos were profiled at two pixel resolutions: 50 µm for the head and upper body and 10 µm for zoomed-in mapping of the facial and forebrain region. Postnatal mouse brains were profiled at 20 µm pixel resolution. We developed

a computational pipeline to preprocess and analyse the data (Supplementary Fig. 1).

## Data quality of spatial-DMT

Spatial-DMT is reproducible. DNA methylation and RNA expression had high concordance (Pearson's $r$ = 0.9836 for DNA methylation and $r$ = 0.9752 for RNA expression; Methods) between the replicate E11 embryo maps with matched body parts (Extended Data Fig. 2a). Uniform manifold approximation and projection (UMAP) co-embedding of the two replicate spatial maps revealed co-localization of pixels from similar body parts but from different datasets (Extended Data Fig. 2b,c). Notably, marker genes, including *Frem1* (face), *Ank3* (brain) and *Trim55* (heart), showed spatial expression patterns consistent with their known tissue-specific expression profiles (Extended Data Fig. 2d–f).

DNA-methylation analysis generated 2.8–3.9 billion raw reads per sample for E11 (10 µm and 50 µm), E13 (50 µm) and P21 brain (20 µm) samples (Supplementary Table 5). Low-signal pixels were filtered on the basis of the knee-plot cut-off threshold (Extended Data Fig. 3a). After stringent quality control, 32.2–65.7% of reads (887,671,712–1,882,630,968) were retained, yielding 355,069–753,052 reads per pixel across 1,699–2,493 pixels in the E11, E13 and P21 samples (Supplementary Table 5). On average, 136,639–281,447 CpGs were covered per pixel across E11, E13 and P21 samples (Supplementary Table 5), comparable to previous single-cell DNA-methylation studies of mouse embryos and brain samples[5,10,12,14] (Fig. 1c). Duplication rates ranged from approximately 20 to 53% across samples (Extended Data Fig. 3b). The retention rate, defined as the percentage of unconverted cytosines owing to methylation or incomplete conversion[12], of mitochondrial DNA was minimal (below 1%; Extended Data Fig. 3c). The CpG retention rates were consistently between 70% and 80% across all samples, whereas the methylation level of non-CpG sites (mCH; H denotes A, C or T) remained low (mCA < 1% in embryos; mCA ≈ 3–4% in the postnatal brain; Fig. 1d and Extended Data Fig. 3d). The mCA level was higher in the P21 mouse brain, consistent with the known increase in non-CpG methylation in postnatal neuronal tissues[5,15]. Analysis of the methylation-free linker sequences showed that more than 99% of cytosines were successfully converted, further confirming conversion efficiency (Extended Data Fig. 3e). Furthermore, no evidence of RNA contamination, such as poly(A), poly(T) or template switching oligonucleotide sequences, was detected in the DNA-methylation libraries (Extended Data Fig. 3f).

We further assessed the genomic distribution of CpG coverage and found it to be uniformly distributed across genomic regions (Extended Data Fig. 4a). Furthermore, methylation levels across various chromatin states were consistent with known biology and comparable to those reported in published databases[5,10,12,14,16] (Extended Data Fig. 4b). For instance, DNA-methylation levels were low at transcription start sites but increased upstream and downstream of these regions (Extended Data Fig. 4c). Collectively, our approach yielded accurate and unbiased profiling of DNA methylation across the genome.

Simultaneously, we generated high-quality RNA data from the same tissue sections, enabling direct comparisons between transcriptional activity and epigenetic states. Specifically, we identified expression of 23,822–28,695 genes in our spatial maps (Supplementary Table 5). At the pixel level, the average number of detected genes per pixel ranges from 1,890 (E11 embryo, 10 μm; 3,596 UMIs) to 4,626 (E11 embryo, 50 μm; 16,709 UMIs), comparable to previous spatial transcriptomic studies of the mouse embryo and brain[1,2] (Fig. 1e and Supplementary Table 5). Lower-resolution pixels (for example, 50 μm) captured more UMIs and expressed genes, probably owing to the inclusion of more cells in each pixel.

## Spatial co-profiling of mouse embryos

Mammalian embryogenesis is an intricately programmed process with complex gene-expression dynamics regulated by epigenetic mechanisms, including DNA methylation, at spatiotemporal scales[17]. We applied spatial-DMT to produce spatial tissue maps of the DNA methylation and gene expression of the E11 mouse embryo. The DNA methylome and transcriptome define cell identity independently, with pixels clustered by methylation read-outs sampled from variably methylated regions (VMRs)[18] and expression levels from variably transcribed genes (Fig. 1f, left and middle, and Extended Data Fig. 4d). The two modalities can be integrated to achieve improved discrimination of intercellular and spatial diversity using a weighted nearest neighbour (WNN) method[19] (Fig. 1f, right). Superimposing the WNN clusters over histological images suggests clear correspondence with anatomical structures, for example, craniofacial region (W0), hindbrain–spinal cord (W2) and embryonic heart (W6), consistent with the tissue histology (Fig. 1f, bottom left).

The correlations between single-modality (RNA and DNA methylation) clusters and their integrated WNN clustering results were evaluated (Fig. 1f and Extended Data Fig. 4e,f). Each modality captured distinct yet complementary aspects of cellular identity and their integration through WNN analysis yielded clusters with enhanced resolution. RNA-expression profiles had a broader dynamic range, which may result in distinct clustering granularity across modalities. To quantify the relative contributions of each modality to the integrated clusters, we computed modality weights for individual spatial pixels (Extended Data Fig. 4g). This analysis revealed that some clusters were defined predominantly by gene expression (for example, W6, cardiac tissue), whereas others by DNA methylation (for example, W11, craniofacial region). Collectively, these results highlight the advantage of spatial multi-omics integration in resolving cellular heterogeneity missed by single-modality analyses.

We examined VMR methylations specific to the brain (W2), craniofacial (W0) and heart (W6) regions (Fig. 1f) and neighbouring gene expression (Fig. 2a,b, cluster W2; Extended Data Fig. 6a,b, cluster W0; and Extended Data Fig. 7a,b, cluster W6). Spatial cluster-specific gene expression is frequently associated with low DNA methylation at the neighbouring VMRs, as exemplified by signature genes *Runx2*, *Mapt*, and *Trim55*, which mark the craniofacial regions (jaw and upper nasal), the brain and spinal cord, and the heart, respectively (Fig. 2c and see Extended Data Figs. 5a, 6c and 7c for additional signature gene examples). Testing the epigenetically regulated genes from each cluster for Gene Ontology (GO) enrichment identified the developmental process

related to the anatomical structures (Extended Data Figs. 5b, 6d and 7d). Although the canonical negative correlations between DNA methylation and RNA expression have been observed in many genes, we also identified genes, for example, *Ank3*, *Atp11c*, *Cyfip2*, *Lmln* and *Khdrbs2*, for which expression is positively correlated with the methylation levels of the associated VMRs (Fig. 2d). For instance, *Ank3*, primarily located at the axonal initial segment and nodes of Ranvier in neurons of the central and peripheral nervous systems[20], had high levels of expression and DNA methylation in the brain region (Fig. 2c). The positive correlations between DNA methylation and RNA expression have been documented at enhancers[21], gene bodies[22] and Polycomb targets[23], suggesting a complex mechanism of DNA methylation in transcriptional regulation, contingent on its interaction with transcriptional factor (TF) binding and histone modifications.

To directly assess the influence of cell-type-specific methylation on TF binding, we performed motif enrichment analysis on differentially hypomethylated VMRs and examined the gene expression of each TF gene. Our findings uncovered a relationship between the expression of TFs and the DNA methylation at their binding sites. For example, TFs associated with heart development, *Hand2*, *Tbx20* and *Meis1*, were expressed in cluster W6, the corresponding hypomethylated VMRs of which were enriched in the binding motifs of these TFs (Fig. 2e, right). Similar sets of tissue-specific TFs are identified for other embryo structures, for example, *Ebf1* and *Pbx1* in the brain and spinal cord and *Sox9*, *Ebf1* and *Zeb2* in the craniofacial region (Fig. 2e). Notably, *Ebf1* was expressed and its binding motif was also enriched across all three tissue regions (Fig. 2e). EBF1 was reported as an interaction partner for TET2 and was suggested as a sequence-specific mechanism for regulating DNA methylation in cancer[24].

To precisely resolve the fine structure of transcriptional regulation in the craniofacial and forebrain regions of the E11 embryo, we used a 10 μm pixel size microfluidic chip to produce a spatial map with near single-cell resolution (Fig. 1g,h and Extended Data Fig. 8a). Integration of our spatial dataset with a single-cell reference[25] revealed a strong concordance between the two datasets (Extended Data Fig. 8b,c). For example, W11 maps to neuroectoderm and glial cells in the single-cell reference, W7 to CNS neurons, W5 to olfactory sensory neurons and W10 to epithelial cells (Extended Data Fig. 8b,c). Cell-type deconvolution using the single-cell RNA-sequencing (scRNA-seq) reference[25] identified distinct clusters that correspond precisely to known anatomical structures of the developing mouse brain. Notably, two spatially defined clusters, W7 and W11, captured key telencephalic compartments. W11 was enriched for telencephalon progenitors in the ventricular zone of the pallium, a neurogenic niche characterized by active cell division and proliferation, whereas W7 corresponded to γ-aminobutyric-acid-releasing (GABAergic) cortical interneurons localized in the mantle zone, where newborn neurons migrate, accumulate and differentiate to establish cortical architecture[26] (Fig. 1h and Extended Data Fig. 8d). GO enrichment analysis further supported these regional identities, highlighting biological processes associated with neurogenesis and progenitor proliferation in W11, and neuron projection and migration in W7 (Extended Data Fig. 8e). Moreover, the cell-type lineage tree constructed from the scRNA-seq reference confirmed the developmental trajectory, positioning telencephalon progenitors as direct precursors to GABAergic cortical interneurons[25]. Beyond the forebrain, our spatial analysis also resolved refined sensory structures in the developing olfactory system (cluster W5 and W10; Fig. 1g and Extended Data Fig. 8a). Sensory neurons were notably enriched in W5 (Extended Data Fig. 8f,g), spatially localized adjacent to the forebrain. This spatial pattern aligns closely with the established developmental trajectory of the olfactory system, in which olfactory sensory neurons progressively form connections with the forebrain as embryogenesis progresses[27]. Together, these findings provide a high-resolution view of neural and sensory system formation, underscoring the power of spatial-DMT in resolving intricate anatomical structures and capturing the spatiotemporal dynamics.

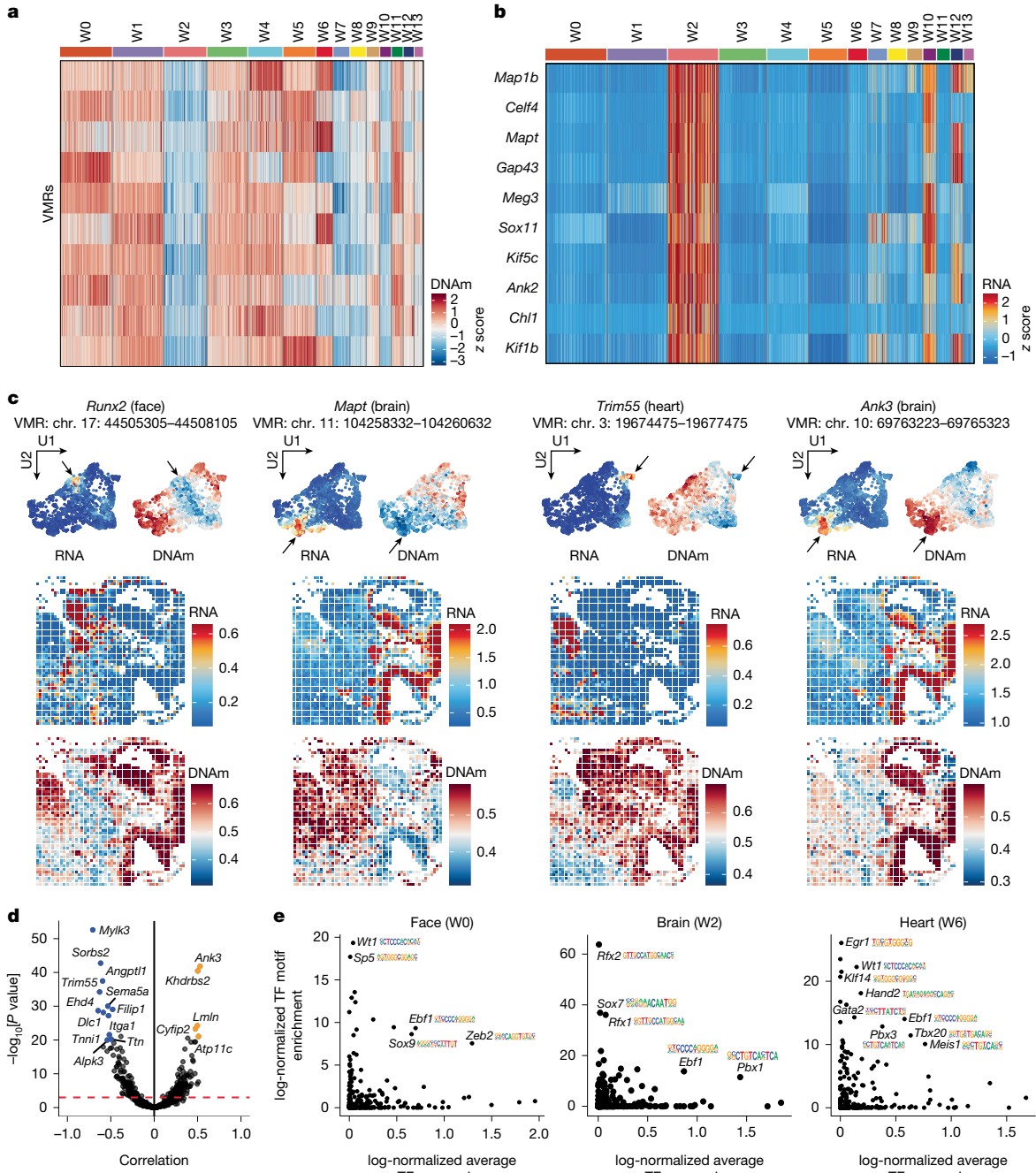

**Fig. 2 | Analysis of DNA methylation and gene expression in E11 embryo.**
**a**, Heat map of DNA methylation levels for the top ten differentially methylated genomic loci in brain and spinal cord (cluster W2) of E11 embryo. Each row represents a specific genomic locus and each column represents a different cluster. The colour scale indicates the z scores of DNA-methylation levels. **b**, Heat map of expression levels for nearby genes corresponding to the genomic loci in **a**. Each row represents a specific gene and each column represents a different cluster. The colour scale indicates the z scores of gene-expression levels. **c**, UMAP visualization and spatial mapping of DNA methylation (methylation percentage) and RNA-expression levels (log-normalized expression) for selected marker genes across different clusters. U1, UMAP 1; U2, UMAP 2. **d**, Correlation analysis between DNA methylation and RNA expression of nearby genes. Negative correlations (blue) indicate a repressive effect of methylation, whereas positive correlations (orange) indicate activation. Correlation coefficient and P value were calculated using the two-sided Pearson correlation test and P values were adjusted for multiple comparisons using the Benjamini–Hochberg method. **e**, Scatterplots of enriched TF motifs in the respective clusters. The y axis shows the enrichment P values from Homer one-sided hypergeometric test of TF motifs and the x axis shows the average gene expression of the corresponding TFs.

## Methylation and transcription dynamics

Mammalian embryogenesis is a precisely timed process with dynamic DNA methylation and gene expression underlying cell differentiation and tissue development[16]. By using pseudotime and spatial-DMT analyses on embryos of two different gestational ages (E11 and E13),

we can investigate the dynamics of DNA methylation, gene expression and their interactions at both spatial and temporal scales. We first analysed the developing brain, focusing on the differentiation trajectory from oligodendrocyte progenitors to premature oligodendrocytes. Spatial mapping of the pseudotime of each pixel revealed the organized migration of oligodendrocyte progenitor cells from the subpallium

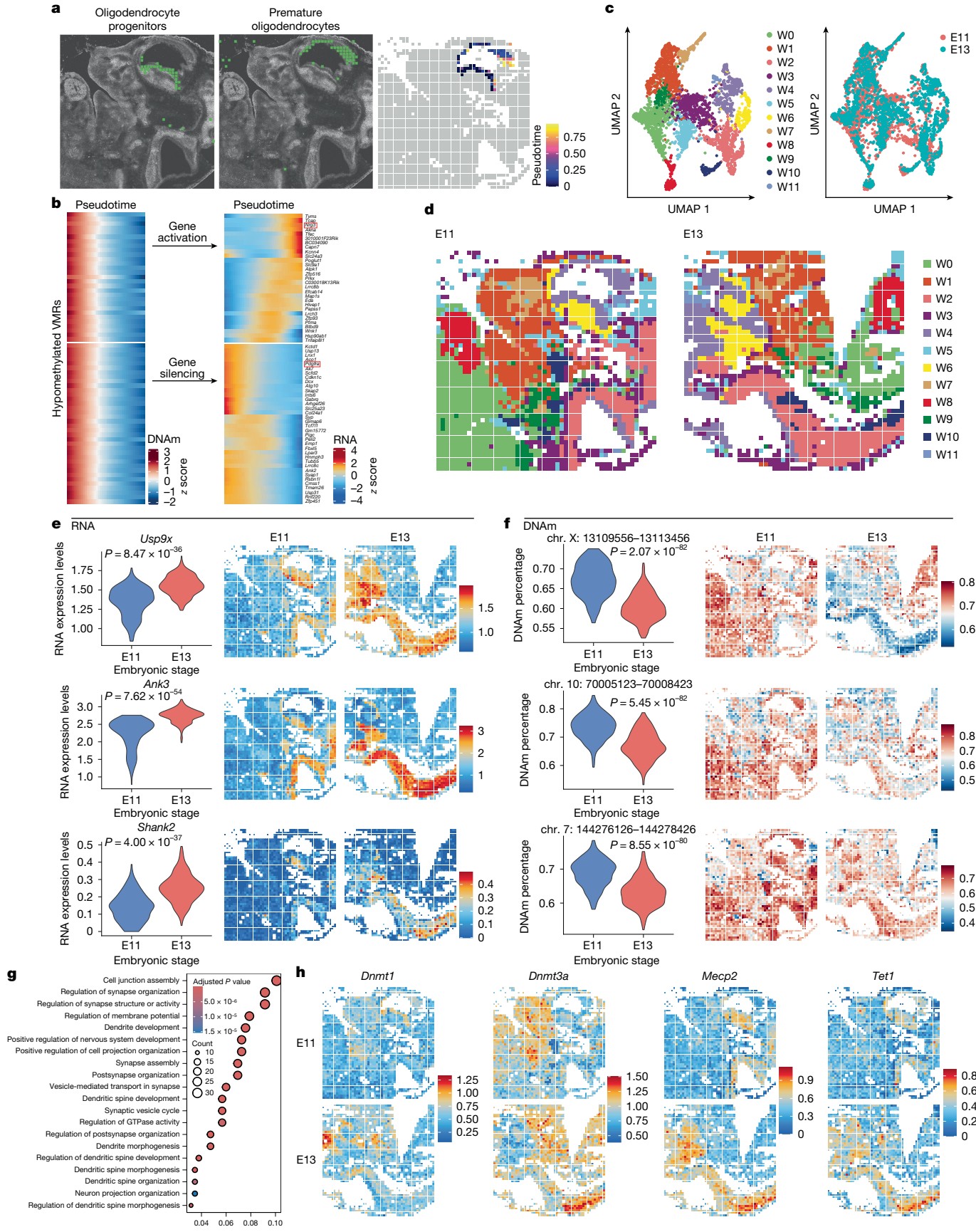

**Fig. 3** | See next page for caption.

**Fig. 3 | Spatiotemporal dynamics of DNA methylation and RNA transcription during embryogenesis. a**, Spatial mapping of oligodendrocyte progenitors (left) and premature oligodendrocytes (middle) identified by label transfer from mouse embryo scRNA-seq data[17] to spatial-DMT, with pseudotemporal reconstruction of the oligodendrogenesis plotted in space (right). **b**, Pseudotime heat maps of VMRs that become demethylated (left) and expression changes of nearby genes from oligodendrocyte progenitors to premature oligodendrocytes (right). Each row represents a specific genomic locus (left) and nearby genes (right), with columns representing tissue pixels ordered by pseudotime. The colour scale indicates the *z* scores of DNA methylation and gene expression. **c**,**d**, UMAP visualization (**c**) and spatial distribution (**d**) of integrated E11 and E13 WNN analysis. Spatial tissue pixels from different developmental stages conform

well and match with the tissue types. **e**,**f**, Comparative analysis of RNA-expression levels (log-normalized expression) with two-sided Wilcoxon rank-sum test, unadjusted $P = 8.47 \times 10^{-36}$, $P = 7.62 \times 10^{-54}$, $P = 4.00 \times 10^{-37}$ from top to bottom (**e**) and DNA methylation (methylation percentage) with two-sided Wilcoxon rank-sum test, unadjusted $P = 2.07 \times 10^{-82}$, $P = 5.45 \times 10^{-82}$, $P = 8.55 \times 10^{-80}$ from top to bottom (**f**) for upregulated genes in E13 brain and spinal cord. **g**, GO enrichment analysis from one-sided hypergeometric test of biological processes related to demethylated and upregulated genes in the brain and spinal cord from E11 to E13 stages. **h**, Spatial mapping of the expression levels (log-normalized expression) of DNA-methylation-related enzymes in the brain and spinal cord regions across E11 and E13 stages.

to the pallium during oligodendrogenesis (Fig. 3a and Extended Data Fig. 5c). This pseudotemporal process is associated with coupled DNA methylation and gene expression in different patterns. For example, loss of DNA methylation can be both associated with gene activation (for example, *Nrg3*, an oligodendrocyte marker[28]) and silencing (for example, *Pdgfra*, an oligodendrocyte precursor marker[29]; Fig. 3b, red boxes). The presence of both the positive and the negative couplings between DNA methylation and gene expression is aligned with the above comparisons across spatial pixels, reinforcing the regulatory diversity of mouse embryogenesis and oligodendrogenesis[29].

To further illustrate the temporal molecular dynamics of embryonic development, we performed spatial mapping of the E13 mouse embryo (Extended Data Fig. 9a). First, we validated our RNA dataset by integrating it with a published spatial ATAC–RNA (ATAC, assay for transposase-accessible chromatin) reference[2], which revealed a strong concordance (Supplementary Fig. 2a). Co-clustering the two datasets identified 11 distinct cluster populations, each corresponding to the histological location in both samples (Supplementary Fig. 2b,c). By integrating spatial data from E11 and E13 embryos (Fig. 3c,d), we identified genes upregulated in E13 (Fig. 3e and Extended Data Fig. 9b,d, left) associated with notable loss of DNA methylation (Fig. 3f and Extended Data Fig. 9c,d, right). These genes are implicated in the corresponding tissue functions. For example, *Usp9x*, *Ank3* and *Shank2* are critically involved in neuronal development, synaptic organization, morphogenesis and transmission[30–32] (Fig. 3e,f). *Ctnna1*, *Pecam1* and *Lamb1* are pivotal for maintaining cardiac tissue integrity[33], vascular development[34] and cardiac tissue structuring[35], respectively (Extended Data Fig. 9b,c). Functional analysis of upregulated genes further corroborated their association with biological processes, for example, synapse organization, neuron projection organization and dendritic spine morphogenesis in the brain (Fig. 3g), and sphingolipid metabolic processes in the heart (Extended Data Fig. 9e). Our tissue map data more precisely timed these gene and pathway activations to a specific stage of embryo development, tissue location and shed light on their epigenetic regulatory mechanisms. Notably, besides regulators of tissue development, some DNA methylation writers[36], readers[37] and eraser enzymes[38], for example, *Dnmt1*, *Dnmt3a*, *Mecp2* and *Tet1*, showed higher expression in the E13 embryo, suggesting elevated biochemical activity that drove the global DNA methylation dynamics (Fig. 3h).

## Co-mapping of mCH–mCG–RNA in mouse brain

mCH methylation, particularly mCA, is uniquely abundant in the brain[5,15]. To evaluate the spatial heterogeneity of non-CpG cytosine methylation, we applied spatial-DMT to the cortical and hippocampal regions of a P21 mouse brain (Fig. 4a). Initial analysis of global mCA and mCG levels revealed relatively low methylation in the dentate gyrus (DG), cornu ammonis (CA)1/2 and CA3 regions, compared with the cortex (Extended Data Fig. 3d).

The spatial distribution of methylome and transcriptome clusters matched anatomical structures shown by the histology image, reflecting the arealization of this brain region (Fig. 4b and Extended Data

Fig. 10a). Expression of known cell-specific marker genes is spatially distributed in regions in which these cell types were enriched. For instance, *Prox1*, a TF crucial for neurogenesis and the maintenance of granule cell identity[39], was prominently expressed in the DG (Fig. 4f, right). *Satb1*, which has a role in cortical neuron differentiation and layer formation[40], was strongly enriched in the cerebral cortex (Extended Data Fig. 10b, right). *Bcl11b*, a TF essential for neuronal progenitor cell differentiation[41], demonstrated elevated expression in the hippocampus (Extended Data Fig. 10c, right).

To elucidate the regulatory roles of mCG and mCA on gene transcription, we compared the two modifications for the signature genes of each cluster against all other clusters combined. By correlating the modification levels with gene expression, we identified genes potentially regulated by mCG, mCA or both (Fig. 4c–e). *Prox1* and *Bcl11b* expression was significantly associated with both mCG and mCA (Fig. 4c,f and Extended Data Fig. 10c), broadly marking the DG and CA1/2 regions, respectively. By contrast, *Ntrk3*, a receptor tyrosine kinase crucial for nervous system function[42], was highly expressed in the hippocampal CA1/2 and DG regions, correlating primarily with mCG levels but not with mCA (Fig. 4d,g). Similarly, *Satb1* expression in the cortex was strongly correlated with mCG but not with mCA levels (Extended Data Fig. 10b). The silencing of *Cux1*, a TF involved in neuronal development and function[43], had a negative correlation with CA and CG hypermethylation in the CA3 region. By contrast, in the CA1/2 region, *Cux1* expression showed a negative correlation only with CA hypermethylation and seemed independent of mCG levels (Fig. 4e,h). Across both sequence contexts, negative correlations between DNA methylation and gene expression are more prevalent than positive ones, highlighting the predominantly repressive nature of these epigenetic modifications (Extended Data Fig. 10d). Collectively, mCG and mCA regulate transcription in a gene-specific manner, jointly defining the cell identity across cell types and brain regions.

Further analysis of neuronal and glial populations revealed cell-type- and region-specific transcriptomic and epigenetic variation. For example, *Syt1* and *Rbfox3* are broadly expressed across neurons in all cortical layers (Supplementary Fig. 3a). By contrast, *Cux2*, *Cux1* and *Satb2* were highly expressed in upper-layer neurons, whereas *Bcl11b* was enriched in the deeper cortical layers (Supplementary Fig. 3b,c). Oligodendrocytes (*Mbp* and *Plp1*) and fibrous astrocytes (*Gfap*) were specifically enriched in the corpus callosum and hippocampal regions (Supplementary Fig. 3d,e). To further resolve cell identities, we integrated our spatial-DMT dataset with a reference scRNA-seq dataset[44] to associate spatial clusters with previously defined cell types. For instance, W0, W3 and W5 were identified as oligodendrocytes, DG granule neurons and telencephalon-projecting excitatory neurons, respectively (Supplementary Fig. 4a–c). Aggregated gene-expression and DNA-methylation levels across pixels in each cluster were highly correlated with the corresponding scRNA-seq transcriptomes[44] (Supplementary Fig. 4d) and single-cell DNA methylome profiles[5] (Supplementary Fig. 4e), as illustrated by W3 (DG) and W5 (CA1) (Fig. 4b, right). Cell-type deconvolution using the same scRNA-seq reference[44] revealed spatial organization of diverse cell types, consistent with

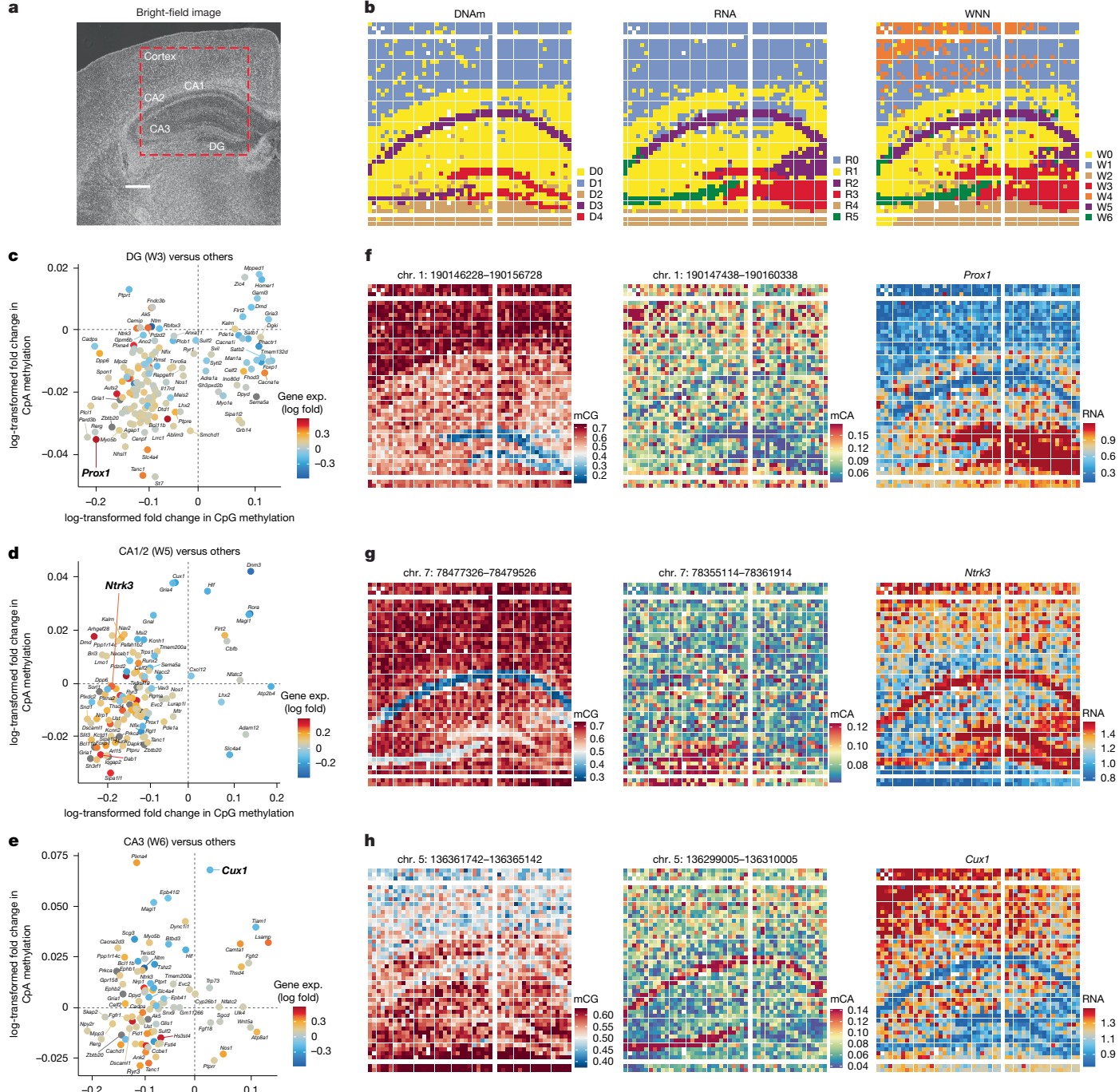

**Fig. 4 | Spatial DNA-methylation and RNA-transcription analyses in the P21 mouse brain. a**, Bright-field image of the P21 mouse brain section (pixel size, 20 μm; ROI area, 2 × 2 mm²) showing the analysed regions, including the DG, CA1, CA2, CA3 and cortex (*n* = 1). Scale bar, 500 μm. **b**, Spatial distribution of clusters identified in DNA-methylation data (left), RNA-transcription data (middle) and integrated DNA and RNA data using WNN analysis (right). **c**–**e**, Scatter plot displaying the relationship between CpG and CpA methylation changes (log-transformed fold change) and gene-expression changes (log-transformed fold change; gene exp. (log fold)) for marker genes in DG (**c**), CA1/2 (**d**) and CA3 (**e**). Each dot represents a specific gene, coloured by the log-fold change in gene expression, with red indicating upregulation and blue indicating downregulation. These plots demonstrate how differential methylation at CpG and CpA sites correlates with changes in gene expression, illustrating the complex regulatory landscape across different brain regions. **f**–**h**, Spatial mapping of CpG (left) and CpA (middle) methylation and RNA-expression levels (right) for selected genes in different brain regions.

known brain anatomy (Supplementary Fig. 4f). For example, cortical excitatory neurons showed expected laminar distribution: TEGLU7, TEGLU8, TEGLU4 and TEGLU3 were enriched in layers 2/3, 4, 5 and 6, respectively. In the hippocampus, TEGLU24 and TEGLU23 mapped to CA1/2 and CA3 regions, whereas in granule neurons, DGGRC2 localized to the DG. Furthermore, DEGLU1 was enriched in the thalamic region.

All cell types were mapped to expected brain anatomical regions in agreement with the scRNA-seq reference[44]. These observations underscore the robustness of spatial-DMT in resolving complex tissue architectures and simultaneously profiling the epigenome and transcriptome with their spatial context, which remains challenging for conventional anatomical and single-cell approaches.

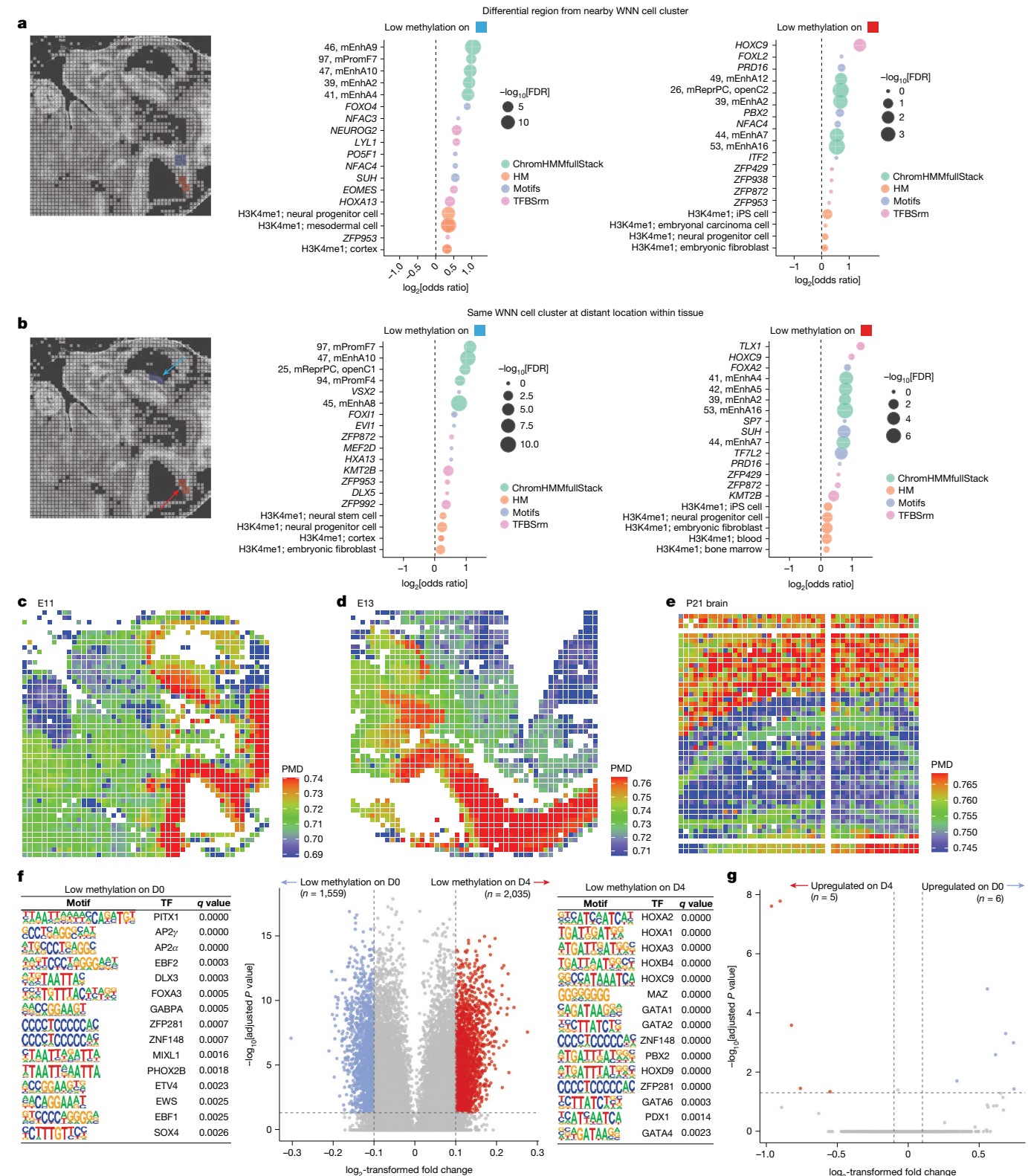

**Fig. 5 | Spatial-DMT resolves mitotic history and subtle region-specific epigenetic variations. a**, Differential region analysis between adjacent WNN cell clusters (left). Chromatin features enriched at hypomethylated genome loci in the blue region (middle) and those enriched in the red region (right) are shown. A summary of the ChromHMMfullStack (full-stack chromatin hidden Markov model) annotation is provided in Supplementary Table 6. HM, histone modification; TFBSrm, transcription factor binding sites from roadmap dataset. **b**, Differential region analysis in the same WNN (W7) cell cluster but across spatially distant locations (left). Chromatin features enriched at hypomethylated loci in the blue (middle) and red (right) regions are shown. False discovery rate

(FDR) from one-sided Fisher's exact test. **c**–**e**, Methylation levels at PMDs in the E11 mouse embryo (**c**), E13 mouse embryo (**d**) and P21 mouse brain (**e**). **f**, Enriched TF motifs from Homer one-sided hypergeometric test *q* value (on the basis of Benjamini–Hochberg procedure) in differentially hypomethylated VMRs between D0 and D4 cluster pixels derived from cluster R3 in the E11 embryo (two-sided Wilcoxon rank-sum test, adjusted *P* value on the basis of Bonferroni correction). **g**, Differentially expressed genes corresponding to the D0 and D4 pixels originated from the R3 cluster in the E11 embryo (Wilcoxon rank-sum test (two-sided), adjusted *P* value on the basis of Bonferroni correction).

## Resolving regional epigenetic variations

The spatially resolved methylation landscape enables the identification of subtle, region-specific epigenetic variations that may otherwise be undetected. To illustrate this, we performed regional differential methylation analysis, comparing the ventricular and mantle zone of the hindbrain–spinal cord (Fig. 5a). This analysis identified differential methylation at promoters and enhancers specific to neural progenitor cells, as shown by the enrichment of corresponding methylated histone H3 K4 (H3K4me1). These differential methylated loci co-localized with enhancers functionally implicated in brain and neural tube development (mEnhA9), which are associated with TF binding sites crucial for neuronal differentiation, including for the TFs FOXO4 (ref. 45), NEUROG2 (ref. 46) and HOXC9 ref. 47).

Spatial methylation profiles may reveal subtle differences between distantly located pixels in the same WNN cell cluster. We examined two W7 pixel groups located in the forebrain and spinal cord, respectively (Fig. 5b). In the forebrain, regions with loss of methylation were enriched for binding sites of FOXI1, TFs crucial for auditory development[48]. By contrast, spinal-cord-specific hypomethylation correlated with occupancy by TLX1, a TF essential for spinal cord maturation and neuronal differentiation[49]. These findings demonstrate that spatial-DMT can resolve subtle epigenetic heterogeneity across distinct tissue microenvironments.

## Elucidating mitotic history differences

Besides capturing regional TF-mediated cell state differences, spatial-DMT enables the analysis of diverse cell traits through other methylation-based features. For example, partially methylated domains (PMDs), which lose methylation over successive mitotic divisions, serve as indicators of mitotic activity[50]. By spatially mapping PMD methylation in E11 and E13 embryos (Fig. 5c,d), we identified distinct regional patterns. For instance, regions such as the forebrain and hindbrain–spinal cord had higher PMD methylation, whereas embryonic heart tissue demonstrated lower PMD methylation levels, reflecting active cardiogenesis at these developmental stages[51] (Fig. 5c,d). Notably, we observed spatial gradients in PMD methylation, decreasing from the centre to the periphery in the heart, and from the mantle to the ventricular zone in the forebrain and hindbrain–spinal cord. These gradients potentially reflect the spatial organization of progenitor cells and their differentiation trajectories. In the P21 brain (Fig. 5e), cortical layers displayed higher PMD methylation, consistent with reduced proliferative capacity typical of differentiated neurons[52]. By contrast, the DG has comparatively lower PMD methylation, consistent with the presence of neural stem and progenitor cells in the subgranular zone that continue to undergo mitotic division and neurogenesis[53].

Finally, the spatially resolved variations in the methylome and transcriptome can complement each other in distinguishing differentiated cell states (Fig. 1f, left and middle). Despite originating from the same RNA-defined clusters (R3), DNA methylation-defined cluster 0 (D0) and 4 (D4) cells had distinct subpopulations when stratified by their VMRs. Motif enrichment analysis of low-methylation sites in D0 and D4 identified regulatory elements associated with facial and cardiac morphogenesis (Fig. 5f), whereas corresponding gene-expression changes were limited (Fig. 5g). This may reflect epigenetically primed subpopulations that share similar transcriptional states. Together, these findings highlight the power of spatial-DMT in delineating cell identity beyond the resolution of transcriptomic analysis alone.

## Discussion

Understanding how epigenetic regulation is spatially organized in tissues remains a major challenge. Although single-cell DNA methylation and transcriptomic technologies have revealed diverse regulatory architectures in both health and disease[4,5,16], they require tissue dissociation, disrupting the native context. Here we developed spatial-DMT, a pioneering spatial multi-omics technology that enables simultaneous profiling of DNA methylation and RNA expression in the same intact tissue sections, preserving cellular and anatomical architectures.

Spatial-DMT generated high-quality data, with strong reproducibility across technical replicates and concordance with established single-cell and spatial transcriptomic references. When applied to mouse embryos at E11 and E13, it revealed methylome dynamics during embryogenesis. In the postnatal brain, it resolved the neuronal mCH landscape, consistent with elevated mCA levels in mature neurons[5]. With 10 μm resolution, spatial-DMT approaches single-cell granularity, enabling precise delineation of fine tissue structures and region-specific epigenetic features that are typically obscured by tissue dissociation. Our data indicate that gene expression may be differentially influenced by mCG and mCA for different cell types in a spatially restricted manner, potentially owing to the distinct genomic distribution of these methylation marks. Although the observed associations are correlative, they offer valuable insights into the complex interplay between regulatory elements on gene expression. This integrative data analysis also demonstrated that each modality captures distinct yet complementary aspects of cellular states, enhancing cell state differentiation beyond what is achievable with single-modality approaches. This may reflect regulatory redundancy, in which distinct epigenetic mechanisms converge to produce similar transcriptional outcomes. Conversely, the opposite scenario is equally plausible—transcriptional states may diverge despite similar epigenetic landscapes, as epigenetic regulation represents only one layer influencing gene expression. As a result, spatial-DMT offers a cost-effective strategy for high-resolution, unbiased genome-wide methylome analysis.

Future development of spatial-DMT may incorporate additional molecular modalities, including chromatin conformation (HiC), accessibility (ATAC-seq), histone modifications (CUT&Tag), metabolome (mass spectrometry imaging) and surface proteins (CITE-seq), to enhance the resolution of gene regulatory mechanisms in situ. The current enzymatic approach does not distinguish between 5-methylcytosine and 5-hydroxymethylcytosine, this limitation may be overcome by using emerging methods that enable simultaneous measurement of the full spectrum of cytosine base modifications[54]. Coupling our method with long-read sequencing technologies, such as Nanopore or PacBio, could simplify detection and improve coverage of epigenetic modifications. Adaptation of the protocol to formalin-fixed paraffin-embedded (FFPE) specimens would broaden its clinical use, whereas optimizing tagmentation and data-processing pipelines will further improve sensitivity and interpretability. The development of advanced spatial computational deconvolution strategies is essential to resolve signals from lower-resolution pixels, minimize potential biases and enhance biological interpretability. Broader application across tissues, developmental stages and species will further establish the generalizability of spatial-DMT for spatial multi-omics profiling in both basic and translational research.

In summary, spatial-DMT advances spatial omics analyses by enabling the simultaneous mapping of DNA methylation and transcription in the native tissue context. This approach provides a powerful framework for exploring epigenetic regulation, facilitates the discovery of methylation biomarkers, deepens our understanding of disease mechanisms and benefits both fundamental and translational research.

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

## Methods

### Tissue slide preparation

Mouse C57 embryo sagittal frozen sections (MF-104-11-C57 and MF-104-13-C57) were purchased from Zyagen. Freshly collected E11 or E13 mouse embryos were snap frozen in optimal cutting temperature (O.C.T.) compounds and sectioned at 7–10 μm thickness. Tissue sections were collected on poly-L-lysine-coated glass slides (Electron Microscopy Sciences, 63478-AS).

Juvenile mouse brain tissue (P21) was obtained from the C57BL/6 mice housed in the University of Pennsylvania Animal Care Facilities under pathogen-free conditions. All procedures used were approved by the Institutional Animal Care and Use Committee.

Mice were euthanized at P21 using $CO_2$ inhalation, followed by transcranial perfusion with cold Dulbecco's PBS (DPBS). After isolation, brains were embedded in Tissue-Tek O.C.T. compound and snap frozen on dry ice and a 2-methylbutane bath. Coronal cryosections of 8–10 μm were mounted on the back of Superfrost Plus microscope slides (Fisher Scientific, 12-550-15).

### Preparation of transposome

Unloaded Tn5 transposome (C01070010) was purchased from Diagenode and the transposome was assembled following the manufacturer's guidelines. The oligonucleotides used for transposome assembly were: Tn5ME-B, 5′-/5Phos/CATCGGCGTACGACTAGATGTGTATAAGAGACAG-3′; Tn5MErev, 5′-/5Phos/CTGTCTCTTATACACATCT-3′.

### DNA barcode sequences, DNA oligonucleotides and other key reagents

DNA oligonucleotides used for PCR and library construction are shown in Supplementary Table 1. All DNA barcode sequences are provided in Supplementary Tables 2 (barcode A) and 3 (barcode B) and all other chemicals and reagents are listed in Supplementary Table 4.

### Fabrication of the polydimethylsiloxane microfluidic device

Chrome photomasks were purchased from Front Range Photomasks, with a channel width of either 20 or 50 μm. The moulds for polydimethylsiloxane (PDMS) microfluidic devices were fabricated using standard photolithography. The manufacturer's guidelines were followed to spin-coat SU-8-negative photoresist (Microchem, SU-2025 and SU-2010) onto a silicon wafer (WaferPro, C04004). The heights of the features were about 20 and 50 μm for 20- and 50-μm-wide devices, respectively. PDMS microfluidic devices were fabricated using the SU-8 moulds. We mixed the curing and base agents in a 1:10 ratio and poured the mixture onto the moulds. After degassing for 30 min, the mixture was cured at 66–70 °C for 2–16 h. Solidified PDMS was extracted from the moulds for further use. The detailed protocol for the fabrication and preparation of the PDMS device can be found in our previous research[24].

### Spatial joint profiling of DNA methylation and RNA transcription

Frozen tissue slides were quickly thawed for 1 min in a 37 °C incubator. The tissue was fixed with 1% formaldehyde in PBS containing 0.05 U ml$^{-1}$ RNase inhibitor (Enzymatics) for 10 min and quenched with 1.25 M glycine for another 5 min at room temperature. After fixation, tissue was washed twice with 1 ml of DPBS–RNase inhibitor and cleaned with deionized $H_2O$.

The tissue was subsequently permeabilized with 100 μl of 0.5% Triton X-100 plus 0.05 U ml$^{-1}$ RNase inhibitor for 30 min at room temperature, then washed twice with 200 μl DPBS–RNase inhibitor for 5 min each. After permeabilization, the tissue was treated with 100 μl of 0.1 N HCl for 5 min at room temperature to disrupt histones from the chromatin, then washed twice with 200 μl of wash buffer (10 mM Tris-HCl pH 7.4, 10 mM NaCl, 3 mM $MgCl_2$, 1% BSA and 0.1% Tween 20) plus 0.05 U ml$^{-1}$ RNase inhibitor for 5 min at room temperature. Next,

50 μl of transposition mixture (5 μl of assembled transposome, 16.5 μl of 1× DPBS, 25 μl of 2× Tagmentation buffer, 0.5 μl of 1% digitonin, 0.5 μl of 10% Tween 20, 0.05 U ml$^{-1}$ RNase inhibitor (Enzymatics) and 1.87 μl nuclease-free water) was added and incubated at 37 °C for 60 min. After 60 min incubation, the first round of transposition mixture was removed and a second round of 50 μl of fresh transposition mixture was added and incubated for another 60 min at 37 °C. To stop the transposition, 200 μl of 40 mM EDTA with 0.05 U ml$^{-1}$ RNase inhibitor was added with incubation for 5 min at room temperature. After that, 200 μl 1× NEB3.1 buffer plus 1% RNase inhibitor was used to wash the tissue for 5 min at room temperature. The tissue was then washed again with 200 μl of DPBS–RNase inhibitor for 5 min at room temperature before proceeding with the in situ reverse transcription reaction.

### In situ reverse transcription

For the in situ reverse transcription, the following mixture was added: 12.5 μl 5× reverse transcription buffer, 4.05 μl RNase-free water, 0.4 μl RNase inhibitor (Enzymatics), 1.25 μl 50% PEG-8000, 3.1 μl 10 mM dNTPs, 6.2 μl 200 U μl$^{-1}$ Maxima H Minus Reverse Transcriptase, 25 μl 0.5× DPBS–RNase inhibitor and 10 μl 100 μM reverse transcription primer (biotinylated-dT oligo). The tissue was incubated for 30 min at room temperature, then at 45 °C for 90 min in a humidified container. After the reverse transcription reaction, tissue was washed with 1× NEB3.1 buffer plus 1% RNase inhibitor for 5 min at room temperature.

### In situ barcoding

For in situ ligation with the first barcode (barcode A), the first PDMS chip was covered at the tissue ROI. For alignment purposes, a 10× objective (KEYENCE BZ-X800 fluorescence microscope, BZ-X800 Viewer Software) was used to take the bright-field image. The PDMS device and tissue slide were clamped tightly with a custom acrylic clamp. Barcode A was first annealed with ligation linker 1 by mixing 10 μl of 100 μM ligation linker, 10 μl of 100 μM individual barcode A and 20 μl of 2× annealing buffer (20 mM Tris-HCl pH 7.5–8.0, 100 mM $NaCl_2$ and 2 mM EDTA). For each channel, 5 μl of ligation master mixture was prepared with 4 μl of ligation mixture (27 μl T4 DNA ligase buffer, 0.9 μl RNase inhibitor (Enzymatics), 5.4 μl 5% Triton X-100, 11 μl T4 DNA ligase and 71.43 μl RNase-free water) and 1 μl of each annealed DNA barcode A (A1–A50, 25 μM). Vacuum was applied to flow the ligation master mixture into the 50 channels of the device and cover the ROI of the tissue, followed by incubation at 37 °C for 30 min in a humidified container. Then the PDMS chip and clamp were removed after washing the tissue with 1× NEB 3.1 buffer for 5 min. The slide was then washed with deionized water and dried using compressed air.

For in situ ligation with the second barcode (barcode B), the second PDMS chip was covered at the ROI and a bright-field image was taken with the 10× objective. An acrylic clamp was applied to clamp the PDMS and tissue slide together. Annealing of barcode B (B1–B50, 25 μM) and preparation of the ligation mixture are the same as barcode A. The whole device was incubated at 37 °C for 30 min in a humidified container. The PDMS chip and clamp were then removed, and the slide was washed with deionized water and dried using compressed air. A bright-field image was then taken for further alignment.

### Reverse crosslinking

For tissue lysis, the ROI was digested with 100 μl of the reverse crosslinking mixture (0.4 mg ml$^{-1}$ proteinase K, 1 mM EDTA, 50 mM Tris-HCl pH 8.0, 200 mM NaCl and 1% SDS) at 58–60 °C for 2 h in a humidified container. The lysate was then collected in a 0.2-ml PCR tube and incubated on a 60 °C shaker overnight.

### gDNA and cDNA separation

For gDNA and cDNA separation, the lysate was purified with Zymo DNA Clean and Concentrator kit and eluted with 100 μl nuclease-free water. Next, 40 μl of Dynabeads MyOne Streptavidin C1 beads were used and

washed three times with 1× B&W buffer containing 0.05% Tween 20 (50 µl 1 M Tris-HCl pH 8.0, 2,000 µl 5 M NaCl, 10 µl 0.5 M EDTA, 50 µl 10% Tween 20 and 7,890 µl nuclease-free water). After washing, beads were resuspended in 100 µl of 2× B&W buffer (50 µl 1 M Tris-HCl pH 8.0, 2,000 µl 5 M NaCl, 10 µl 0.5 M EDTA and 2,940 µl nuclease-free water) containing 2 µl of SUPERase In RNase inhibitor, then mixed with the gDNA–cDNA lysate and allowed to bind for 1 h with agitation at room temperature. A magnet was then used to separate the beads, which bind to the cDNA that contains dT, from the supernatant that contains the gDNA.

## gDNA library generation

Supernatant (200 µl) was collected from the above separation process for further methylated gDNA detection and library construction. Next, 1 ml of DNA binding buffer was added to the 200 µl supernatant and purified with the Zymo DNA Clean and Concentrator kit again, then eluted in 84 µl (3 × 28 µl) nuclease-free water. The NEBNext enzymatic methyl-seq conversion module (EM-seq) was then used to detect methylated DNA in the sample by converting unmethylated cytosines to uracil; the manufacturer's guidelines were followed. Then, 28 µl of DNA sample was aliquoted into each PCR tube, TET2 reaction mixture (10 µl TET2 reaction buffer containing reconstituted TET2 reaction buffer supplement, 1 µl oxidation supplement, 1 µl DTT, 1 µl oxidation enhancer and 4 µl TET2) was added to the DNA sample on ice. In brief, 5 µl of diluted 1:1,300 of 500 mM Fe (II) solution was added to the mixture and incubated for 1 h at 37 °C in a thermocycler. After the reaction, the sample was transferred to ice and 1 µl of stop reagent from the kit was added. The sample was then incubated for another 30 min at 37 °C. TET2 converted DNA was then purified with 90 µl of solid-phase reversible immobilization (SPRI) beads and eluted with 16 µl nuclease-free water. The thermocycler was preheated to 85 °C, 4 µl formamide was added to the converted DNA and incubated for 10 min at 85 °C in the preheated thermocycler. After the reaction, the heated sample was immediately placed on ice to maintain the open chromatin structure, then reagents from the kit were added (68 µl nuclease-free water, 10 µl APOBEC reaction buffer, 1 µl BSA and 1 µl APOBEC) to deaminate unmethylated cytosines to uracil for 3 h at 37 °C in a thermocycler. Deaminated DNA was then cleaned up using 100 µl (1:1 ratio) of SPRI beads and eluted in 20 µl nuclease-free water.

## Splint ligation

The gDNA tube was heat-shocked for 3 min at 95 °C and immediately put on ice for 2 min. Then, 10 µl of 0.75 µM pre-annealed Splint Ligate P5 (SLP5) adapter was added. This adapter was diluted from a 12 µM stock, which contained 6 µl of 100 µM SLP5RC oligo, 8.4 µl of 100 µM SLS5ME-A-H10 oligo, 5 µl of 10× T4 RNA Ligase Buffer and 30.6 µl nuclease-free water in a PCR tube that was incubated at 95 °C for 1 min, then gradually cooled by −0.1 °C s$^{-1}$ to 10 °C on a thermocycler. Next, 80 µl of ligation master mixture was added to the gDNA tube at room temperature. The mixture contained 40 µl preheated 50% PEG-8000, 12.5 µl SCR buffer (666 mM Tris-HCl pH 8.0 and 132 mM MgCl$_2$ in nuclease-free water), 10 µl of 100 mM DTT, 10 µl of 10 mM ATP, 1.25 µl of 10,000 U ml$^{-1}$ T4 PNK and 6.25 µl of 400,000 U ml$^{-1}$ T4 ligase. The splint ligation mixture was then splinted into five 0.2-ml PCR tubes, 20 µl per tube. The tubes were shaken at 1,000 rpm for 10 s and spun down, then incubated for 45 min at 37 °C, followed by 20 min at 65 °C to inactivate the ligase. For splint ligation indexing PCR, 80 µl of the PCR reaction mixture was mixed in each splint ligated tube. The mixture contained 20 µl 5× VeraSeq GC Buffer, 4 µl 10 mM dNTPs, 3 µl VeraSeq Ultra Enzyme, 5 µl 20× EvaGreen dye, 2 µl of 10 µM N501 primer and 2 µl of 10 µM N70X-HT primer (Supplementary Table 1). The mixture was then aliquoted into a clean PCR tube with 50 µl volume and run on a thermocycler with the setting below, 98 °C for 1 min, then cycling at 98 °C for 10 s, 57 °C for 20 s and 72 °C for 30 s, for 13–19 cycles, followed by 72 °C for 10 s. The reaction was removed once the

quantitative PCR (qPCR) signal began to plateau. The amplified PCR products were pooled and purified with a 0.8× volume ratio of SPRI beads (bead-to-sample ratio) and the completed DNA library was eluted in 15 µl nuclease-free water.

## cDNA library generation

The separated beads containing cDNA were used for cDNA library generation. In brief, 400 µl of 1× B&W buffer with 0.05% Tween 20 was used to wash the beads twice. Then, the beads were washed once with 400 µl of 10 mM Tris-HCl pH 8.0 containing 0.1% Tween 20 for 5 min at room temperature. Streptavidin beads with bound cDNA molecules were placed on a magnetic rack and washed once with 250 µl nuclease-free water before being resuspended in a template switching oligonucleotide solution (44 µl 5× Maxima reverse transcription buffer, 44 µl of 20% Ficoll PM-400 solution, 22 µl of dNTPs, 5.5 µl of 100 mM template switch oligo, 11 µl Maxima H Minus reverse transcriptase, 5.5 µl of RNase inhibitor (Enzymatics) and 88 µl nuclease-free water). Resuspended beads were then incubated for 30 min with agitation at room temperature and for 90 min at 42 °C, with gentle agitation. After the reaction, beads were washed with 400 µl of 10 mM Tris pH 8.0 containing 0.1% Tween 20 and washed without resuspension in 250 µl nuclease-free water. Water was removed on the magnetic rack and the beads were resuspended in the PCR solution (100 µl of 2× Kappa Master mix, 8.8 µl of 10 µM primers 1 and 2 and 92.4 µl nuclease-free water). Next, the beads were mixed well and 50 µl of the PCR mixture was split into four 0.2-ml PCR tubes. The PCR programme was run as follows: 95 °C for 3 min and cycling at 98 °C for 20 s, 65 °C for 45 s and 72 °C for 3 min, for a total of five cycles, followed by 4 °C on hold. After five cycles of PCR reaction, four PCR tubes were placed on a magnetic rack and 50 µl of the clear PCR solution was transferred to four optical-grade qPCR tubes, adding 2.5 µl of 20× Evagreen dye to each tube. The sample was run on a qPCR machine with the following conditions: 95 °C for 3 min, cycling at 98 °C for 20 s, 65 °C for 20 s and 72 °C for 3 min, for 13–17 cycles, followed by 72 °C for 5 min. The reaction was removed once the qPCR signal began to plateau. The amplified PCR product was purified with a 0.8× volume ratio of SPRI beads and eluted in 20 µl nuclease-free water.

A Nextera XT DNA Library Prep Kit was used for cDNA library preparation. In brief, 2 µl (2 ng) of purified cDNA (1 ng µl$^{-1}$), 10 µl Tagment DNA buffer, 5 µl Amplicon Tagment mix and 3 µl nuclease-free water were mixed and incubated at 55 °C for 5 min. Then, 5 µl NT buffer was added to stop the reaction with incubation at room temperature for 5 min. PCR master mix (15 µl 2× N.P.M. Master mix, 1 µl of 10 µM P5 primer (N501) and 1 µl of 10 µM indexed P7 primer (N70X) and 8 µl nuclease-free water) was added. The PCR reaction was run with the following programme: 95 °C for 30 s, cycling at 95 °C for 10 s, 55 °C for 30 s, 72 °C for 30 s and 72 °C for 5 min, for a total of 12 cycles. The PCR product was then purified with a 0.7× ratio of SPRI beads and eluted in 15 µl nuclease-free water to obtain the cDNA library.

## Library quality check and next-generation sequencing

An Agilent Bioanalyzer D5000 ScreenTape was used to determine the size distribution and concentration of the library before sequencing. Next-generation sequencing was conducted on an Illumina NovaSeq 6000 sequencer and NovaSeq X Plus system (150 bp paired-end mode).

## Data preprocessing

For RNA-sequencing data, Read 2 was processed to extract barcode A, barcode B and the UMIs. Using the STARsolo pipeline[56] (v.2.7.10b), these processed data were mapped to the mouse genome reference (mm10). This step generated a gene matrix that captures both gene-expression and spatial-positioning information, encoded through the combination of barcodes A and B. The gene matrix was then imported into R for downstream spatial transcriptomic analysis using Seurat package (v.5.1.0)[57].

For DNA-methylation data, adaptor sequences were trimmed before demultiplexing the FASTQ files using the combination of barcodes A and B. We used the BISulfite-seq CUI Toolkit (BISCUIT) (v.0.3.14)[58] to align the DNA sequences to the mouse reference genome (mm10). Methylation levels at individual CG and CH sites were stored as continuous values between 0 and 1, representing the fraction of methylated reads after quality filtering. These processed CG–CH files were then analysed independently using the MethSCAn pipeline[18] to identify VMRs[59], defined as fused genome intervals with methylation-level variance in the top 2%. We used default parameter settings when running MethSCAn, and the MethSCAn filter min-sites parameter was determined from the read coverage knee plot (Extended Data Fig. 3a). The methylation levels and residuals of VMRs were then imported into R for downstream DNA-methylation analysis.

## Clustering and data visualization

We mapped the exact location of pixels on the bright-field tissue image using a custom Python script (https://github.com/zhou-lab/Spatial-DMT-2024/tree/main/Data_preprocess/Image), before removing additional empty barcodes on the basis of read-count thresholds determined by the knee plot (Extended Data Fig. 3a). Clustering and data visualization were conducted using R in RStudio.

For RNA data, we used the SCTtransform function in the Seurat package (v.5.1.0), built using a regularized negative binomial model, for normalization and variance stabilization. Dimensionality reduction was performed using RunPCA function with the SCTtransformed assay. We then constructed the nearest-neighbour graph using the first 30 principal components with the FindNeighbors function and identified clusters with the default Leiden method in the FindClusters function. Finally, a UMAP embedding was computed using the same principal components with RunUMAP function.

Owing to the inherent sparsity of DNA-methylation data, it is impractical to analyse methylation status solely at the individual CpG level. Binary information at sparse loci cannot be used directly to construct a feature matrix suitable for downstream analysis. In our study, we adopted the VMR framework, which divides the genome into variable-sized tiles and calculates the average methylation level across CpGs in each tile for each pixel[18]. This approach results in a continuous-valued matrix, in which rows correspond to pixels and columns represent genomic tiles, with values ranging from 0 to 1. VMR methylation levels and residuals were then imputed using the iterative principal component analysis approach as suggested in the MethSCAn instructions. Initially, missing residual values were replaced with zero and missing methylation levels were replaced with the average values for that VMR interval. The principal component analysis approach was iteratively applied until updated values stabilized to a threshold. The imputed residual matrix for VMRs was then imported into the existing Seurat object as another modality. Similar to the RNA-clustering pipeline, dimensionality was reduced using the RunPCA function. The first ten principal components from the residual matrix were used for clustering and UMAP embedding.

To visualize clusters in their spatial locations, the SpatialDimPlot function was used after clustering on the basis of gene expression or VMR residuals. UMAP embedding was visualized with the DimPlot function. The FindMarkers function was applied to select genes and VMRs that were differentially expressed or methylated for each cluster. For spatial mapping of individual VMR methylation levels or gene expression, we applied the smoothScoresNN function from the FigR package[60]. The SpatialFeaturePlot function was then used to visualize VMR methylation levels and gene expression across all pixels. To illustrate the relationships between clustering results from different modalities, we generated the confusion matrix and alluvial diagram using the pheatmap and ggalluvial R package[61].

## Integrative analysis of DNA methylation and RNA data

To integrate spatial DNA methylation and RNA data, WNN analysis in Seurat was applied using the FindMultiModalNeighbors function[19]. On the basis of the WNN graph, clustering, UMAP embedding and spatial mapping of identified clusters were performed for integrated visualization.

For the integration of spatial transcriptomics data of E11 and E13 mouse embryos, the top 3,000 integration features were selected, followed by the use of PrepSCTIntegration and IntegrateData functions to generate an integrated dataset. Similarly, to integrate with public single-cell transcriptomic data[25,44], we first identified anchors using the FindIntegrationAnchors function in Seurat, followed by data integration using the IntegrateData function. To integrate DNA-methylation data, common VMRs between both developmental stages were obtained and the integrated CCA method from the IntegrateLayers function was used to join the methylation data from the two developmental stages. A Wilcoxon signed-rank test was performed to compare the methylation levels and gene-expression differences between the two time points.

## TF motif enrichment

To perform TF motif enrichment, we first used the MethSCAn diff function on distinct groups of cells to identify differentially methylated VMRs on the basis of the clustering assignment. The HOMER[62] findMotifsGenome function was then applied to analyse the enrichment of known TF motifs using its default database. We followed the same parameter settings used in MethSCAn, with motif lengths of 5, 6, 7, 8, 9, 10, 11 and 12.

## CpGs enrichment analysis

Enrichment analysis of individual CpGs in the differential regions (Fig. 5a,b) was performed using knowYourCG (https://github.com/zhou-lab/knowYourCG), which provides a comprehensive annotation database for each CpG, including chromatin states, TF binding sites, motif occurrences, PMD annotations and more. To avoid inflated odds ratios for high-coverage data, genomic uniformity was quantified using fold enrichment, defined as the ratio of observed overlaps to expected overlaps. The expected number of overlaps was calculated as: (number of CpGs sequenced × number of CpGs in the chromatin state feature)/ total number of CpGs in the genome.

## Correlation and GO enrichment analysis

Correlation analysis was performed for different clusters. We first used the findOverlaps function in GenomicRanges package (v.4.4)[63] to map VMRs to overlapped genes. Then, the Pearson correlation test was applied to obtain the correlation between mapped genes and corresponding VMRs. The Benjamini–Hochberg procedure was used to adjust all $P$ values.

GO enrichment analysis was conducted using the enrichGO function from clusterProfiler package[64] (v.4.2). For GO enrichment in the comparative analysis of E11 and E13 mouse embryos, the FindMarkers function in Seurat package was used to find differential genes and VMRs in the same cluster from integrated data across two developmental stages. Differentially upregulated genes (false discovery rate ≤ 0.05) with demethylated VMRs were used for the GO analysis.

## Reporting summary

Further information on research design is available in the Nature Portfolio Reporting Summary linked to this article.

## Data availability

Raw and processed data reported in this paper are deposited in the Gene Expression Omnibus (GEO) with accession code GSE270498. Resulting FASTQ files were aligned to the mouse reference genome

(mm10). Published data for data quality comparison and integrative data analysis include a single-cell atlas of mouse embryos (https://oncoscape.v3.sttrcancer.org/atlas.gs.washington.edu.mouse.rna/downloads and https://omg.gs.washington.edu/), a mouse brain atlas (http://mousebrain.org/adolescent/downloads.html) and the Allen Mouse Brain Atlas (https://developingmouse.brain-map.org/).

## Code availability

The data analysis pipeline and code to reproduce analyses are available at GitHub (https://github.com/zhou-lab/Spatial-DMT-2024/)[65] and Zenodo (https://doi.org/10.5281/zenodo.15843594)[66].

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

**Acknowledgements** We acknowledge support from the Packard Fellowship for Science and Engineering (to Y.D.), the pilot award from the Epigenetics Institute at the University of Pennsylvania (to Y.D.), the National Institute of Allergy and Infectious Diseases of the National Institutes of Health under award number DP2AI177913 (to Y.D.) and the National Institute of Health and National Institute of General Medical Sciences (R35-GM146978) (to W.Z.). The content is solely the responsibility of the authors and does not necessarily represent the official views of the National Institutes of Health. C.N.L. was supported in part by the Institute for RNA Innovation of the Perelman School of Medicine at the University of Pennsylvania.

**Author contributions** C.N.L. and Y.D. were responsible for the methodology. C.N.L. conducted all spatial joint profiling of DNA methylation and transcription experiments in E11 (50 μm), E11 (10 μm), E11 replicate (50 μm), E13 embryos and P21 mouse brain. A.C. assisted with experiments, including reagent preparation for E11 (50 μm), E13 embryos and P21 mouse brain. C.N.L., H.F., W.Z. and Y.D. performed computational analysis for E11 (50 μm), E11 (10 μm), E11 replicate (50 μm), E13 embryos and P21 mouse brain DNA methylation and RNA data. The original draft was prepared by C.N.L., H.F., W.Z. and Y.D. All authors reviewed, edited and approved the paper.

**Competing interests** Y.D. and C.N.L. are inventors on a provisional patent application related to this work (application number: 63/809,054). Y.D. is a scientific adviser at AtlasXomics. The other authors declare no competing interests.

**Additional information**
**Correspondence and requests for materials** should be addressed to Chin Nien Lee, Wanding Zhou or Yanxiang Deng.

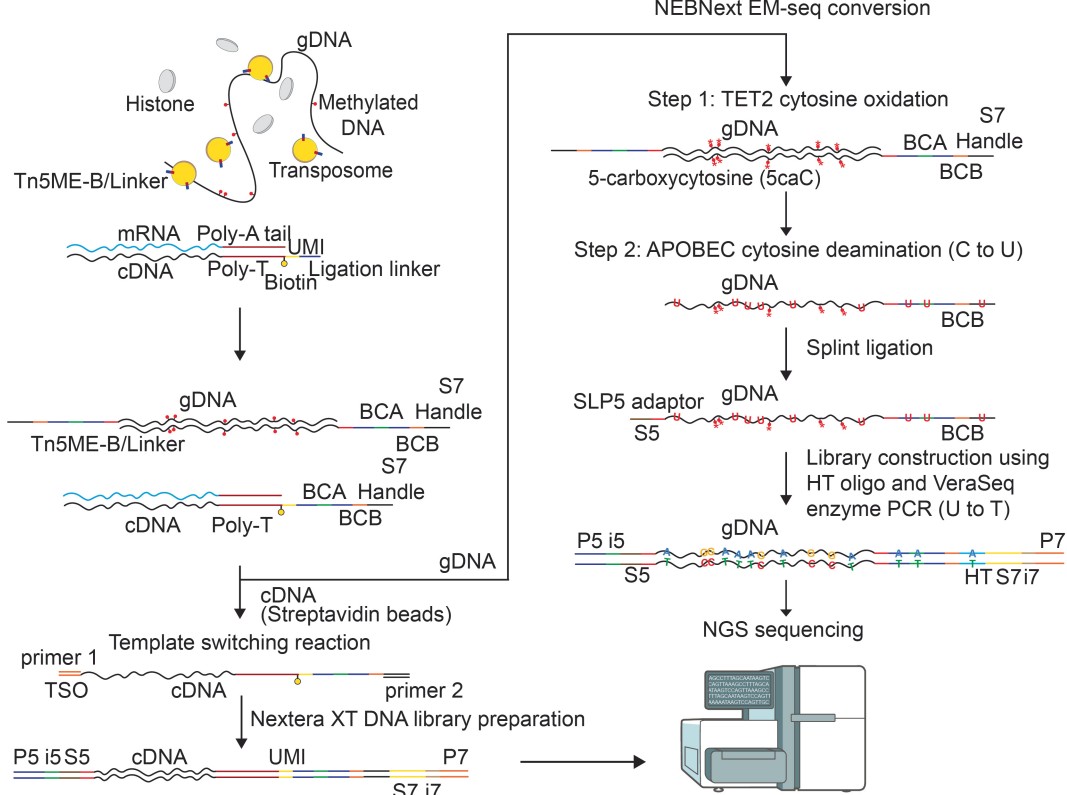

**Extended Data Fig. 1 | Chemistry workflow of Spatial-DMT.** The diagram illustrates the chemistry workflow for simultaneous DNA methylation and RNA library preparation in Spatial-DMT. Initially, tissue sections are fixed and permeabilized to preserve tissue architecture and molecular integrity while allowing reagent penetration. Following permeabilization, sections undergo HCl treatment to disrupt the protein structure and remove nucleosome histones to improve Tn5 transposome accessibility. Next, Tn5 transposition integrates adapters into genomic DNA, and in situ reverse transcription (RT) converts mRNA into complementary DNA (cDNA) directly within the tissue. Subsequently, Barcode A and Barcode B are sequentially ligated to label and spatially encode DNA and mRNA molecules. Streptavidin bead separation then facilitates the differential enrichment and processing of nucleic acids: cDNA molecules undergo a template switching reaction, followed by Nextera XT library preparation, while genomic DNA (gDNA) is subjected to NEBNext EM-seq conversion for methylation detection, splint ligation, and subsequent library construction.

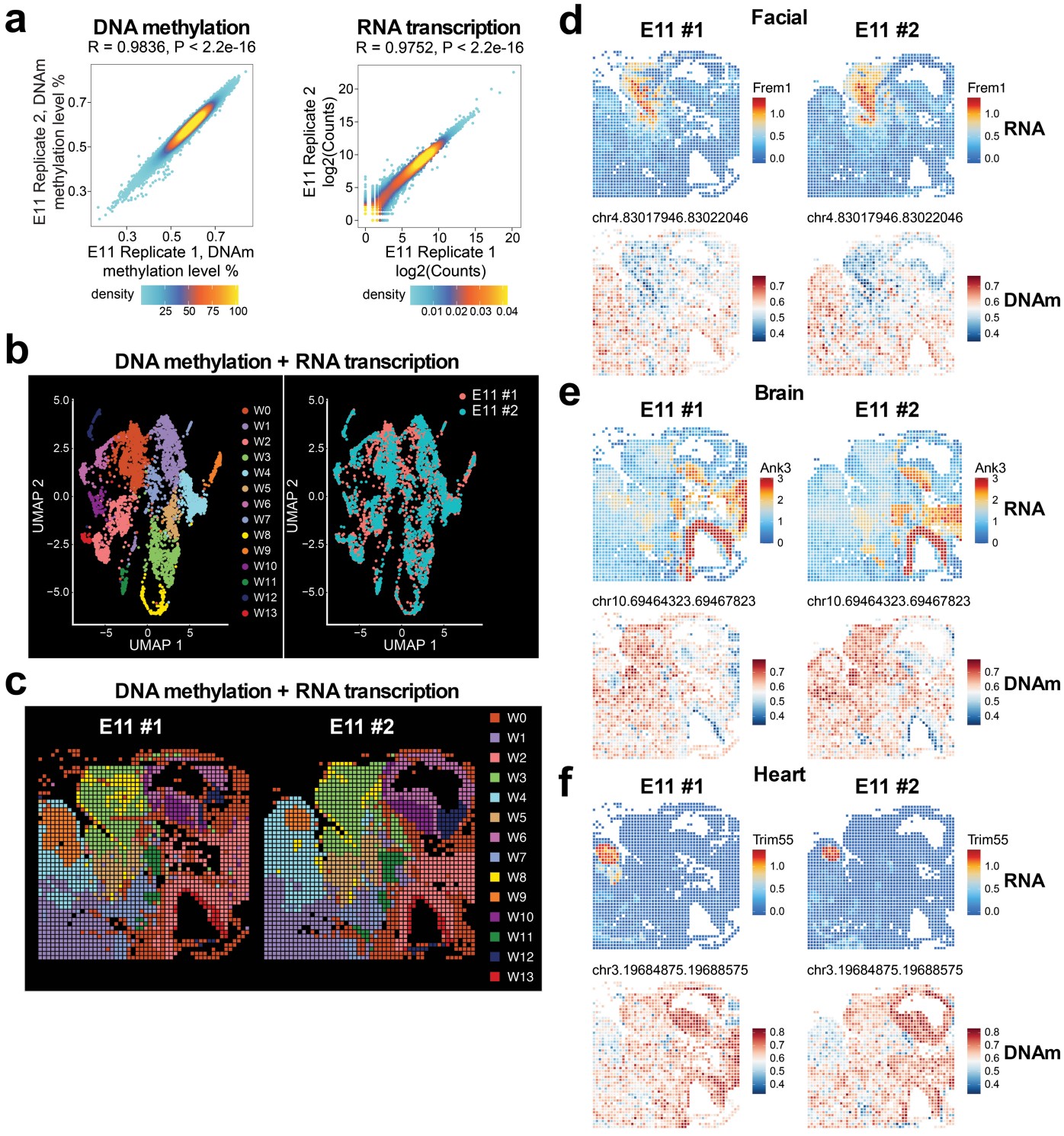

**Extended Data Fig. 2 | Reproducibility of Spatial-DMT. a**, Correlation analysis between replicates based on DNA methylation (left) and RNA transcription (right) data from E11 embryo samples with two-sided Pearson correlation test. **b, c**, Integrative analysis (**b**) and spatial distribution (**c**) of clusters from two independent Spatial-DMT experiments using E11 embryo samples. **d-f**, Spatial mapping of DNA methylation and RNA expression levels for selected marker genes in facial (**d**), brain (**e**), and heart (**f**) regions from two independent Spatial-DMT experiments using E11 embryo samples.

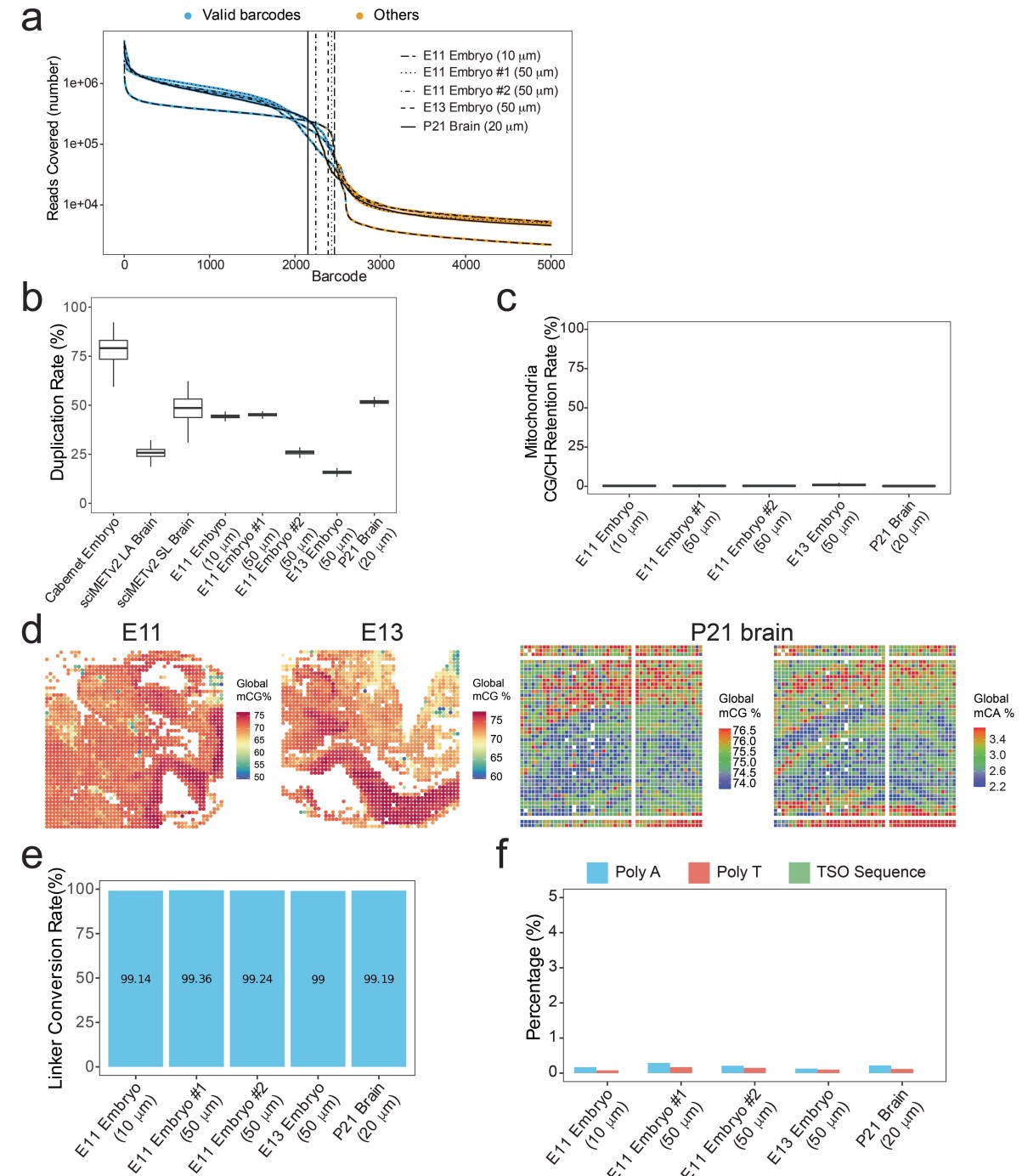

**Extended Data Fig. 3 | Quality control metrics for Spatial-DMT datasets.**
**a**, Barcode rank plots showing the distribution of reads per spatial barcode for E11, E13 embryos, and P21 brain samples. Valid barcodes are shown in blue, and filtered barcodes are shown in orange. **b**, Box plots, where the central line represents the median, the lower and upper hinges indicate the first and third quartiles, and the whiskers extend to the most extreme data points within 1.5 times the interquartile range (IQR) from the hinges, showing the duplication rate for E11, E13 embryos, and P21 brain samples, and other single-cell DNA methylation datasets[12,67]. The y-axis represents the percentage of duplicated reads (n = 278 cells for Cabernet Embryo, n = 927 cells for SciMETv2 LA Brain, n = 1619 cells for SciMETv2 SL Brain, n = 2,493 for E11 embryo (10 µm), n = 1,954

for E11 embryo 1 (50 µm), n = 1,947 for E11 embryo 2 (50 µm), n = 1,699 for E13 embryo (50 µm), and n = 2,235 for P21 brain (20 µm)). **c**, Box plots showing the mitochondrial retention rate for E11, E13 embryos, and P21 brain samples. The y-axis represents the percentage of reads mapped to the mitochondrial genome. **d**, Spatial distribution heatmaps of global DNA methylation levels in E11, E13 embryos (mCG), and P21 mouse brain (mCG and mCA). **e**, The conversion rate of unmethylated cytosine in the linker sequence in E11, E13 embryos, and P21 mouse brain. **f**, Percentage of reads with poly A (≥ 30 adenine), poly T (≥ 30 thymine), and TSO sequence (AAGCAGTGGTATCAACGCAGAGTACATGGG) in the DNA libraries of E11, E13 embryos, and P21 mouse brain.

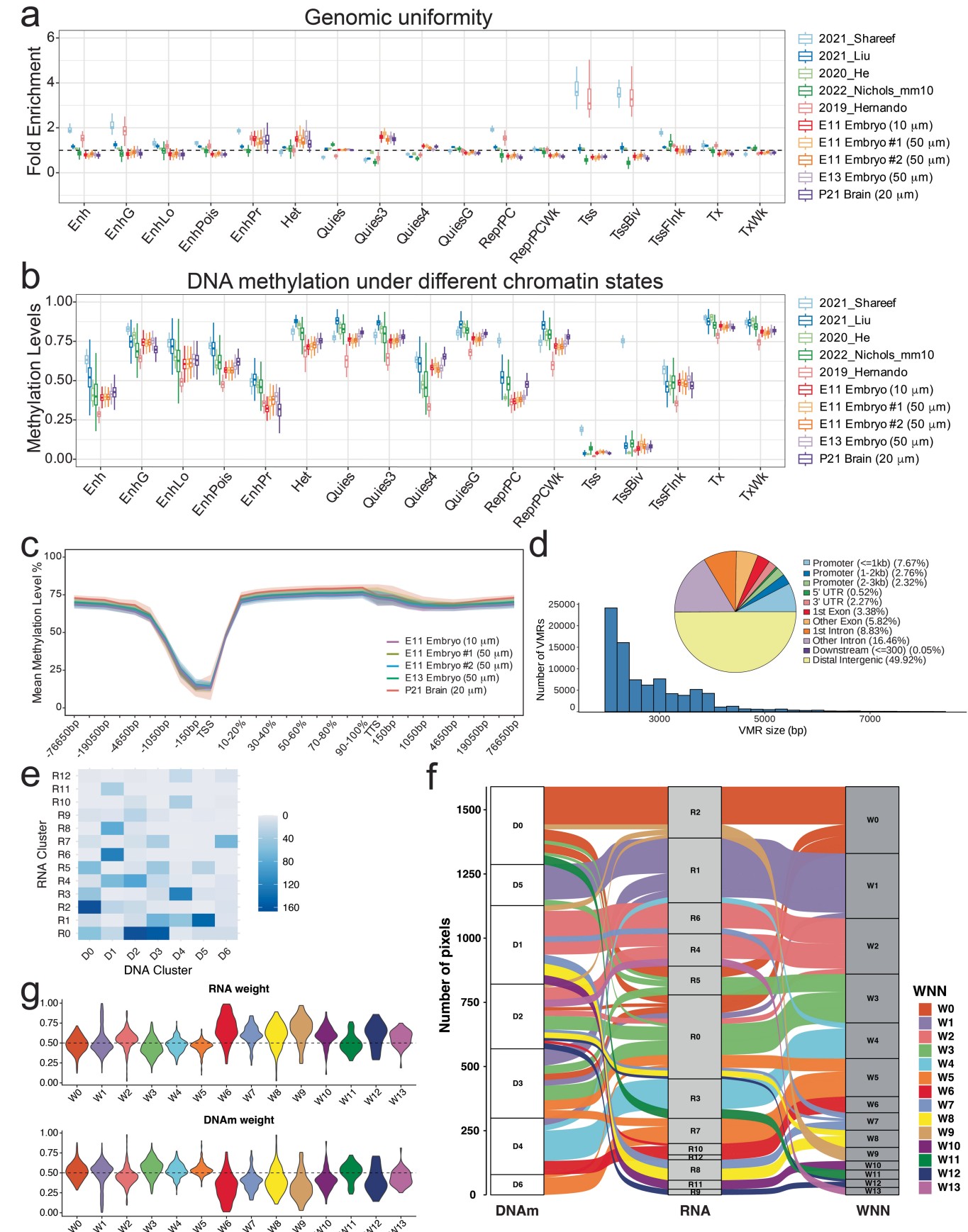

**Extended Data Fig. 4** | See next page for caption.

**Extended Data Fig. 4 | Further analysis of DNA methylation data quality of Spatial-DMT. a**, Genomic uniformity analysis showing the fold enrichment (observed overlaps divided by expected overlaps) for different genomic features, comparing E11, E13 embryos, and P21 brain samples with reference datasets[5,10,12,14,16] (n = 32 cells for 2021_Shareef, n = 103560 cells for 2021_Liu, n = 107 samples for 2020_He, n = 491 cells for 2022_Nichols_mm10, n = 260 cells for 2019_Hernando, n = 2,493 for E11 embryo (10 μm), n = 1,954 for E11 embryo 1 (50 μm), n = 1,947 for E11 embryo 2 (50 μm), n = 1,699 for E13 embryo (50 μm), and n = 2,235 for P21 brain (20 μm), same for b). **b**, DNA methylation levels under different chromatin states, comparing E11, E13 embryos, and P21 brain samples with reference datasets[5,10,12,14,16]. The y-axis represents the methylation levels, with different chromatin states on the x-axis. These chromatin states include active states such as active transcription start site (TSS)-proximal promoter states (TssFlink), actively-transcribed states (Tx, TxWk), enhancer states (Enh, EnhG, EnhLo, EnhPois, EnhPr). Inactive states consist of constitutive heterochromatin (Het), quiescent states (Quies, Quies3, Quies4, QuiesG), repressed Polycomb states (ReprPC, ReprPCWk), and bivalent regulatory states (TssBiv). Boxplots in a and b contain the central line, which represents the median, the lower and upper hinges indicate the first and third quartiles, and the whiskers extend to the most extreme data points within 1.5 times the interquartile range (IQR) from the hinges. **c**, Line plots showing the mean DNA methylation levels (%) across different genomic regions for E11, E13 embryos, and P21 brain samples. The x-axis represents the genomic regions, and the y-axis represents the mean methylation levels. Error bands represent mean values ± 1 standard deviation. **d**, Histogram and pie chart showing the size distribution and genomic annotation of variably methylated regions (VMRs) identified in the E11 embryo. The pie chart indicates the percentage of VMRs in different genomic features, such as promoters, exons, and intergenic regions. **e**, Confusion matrix illustrating the correspondence between clustering results derived from RNA transcription and DNA methylation data in the E11 embryo. **f**, Alluvial diagram showing the relationships among clusters identified by DNA methylation, RNA, and WNN integration in the E11 embryo. **g**, Modality weights indicating the relative contributions of gene expression and DNA methylation for each WNN cluster in the E11 embryo.

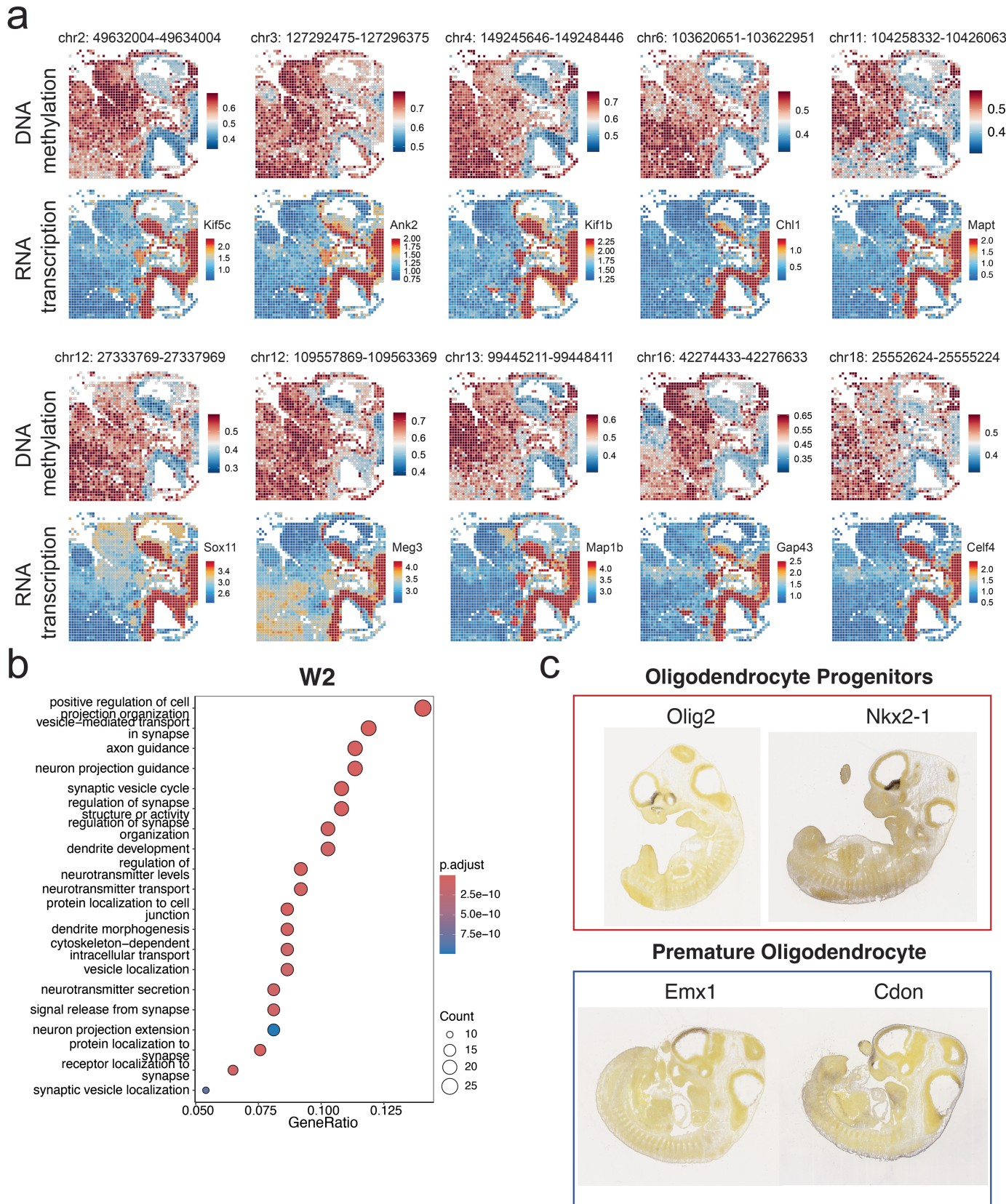

**Extended Data Fig. 5** | See next page for caption.

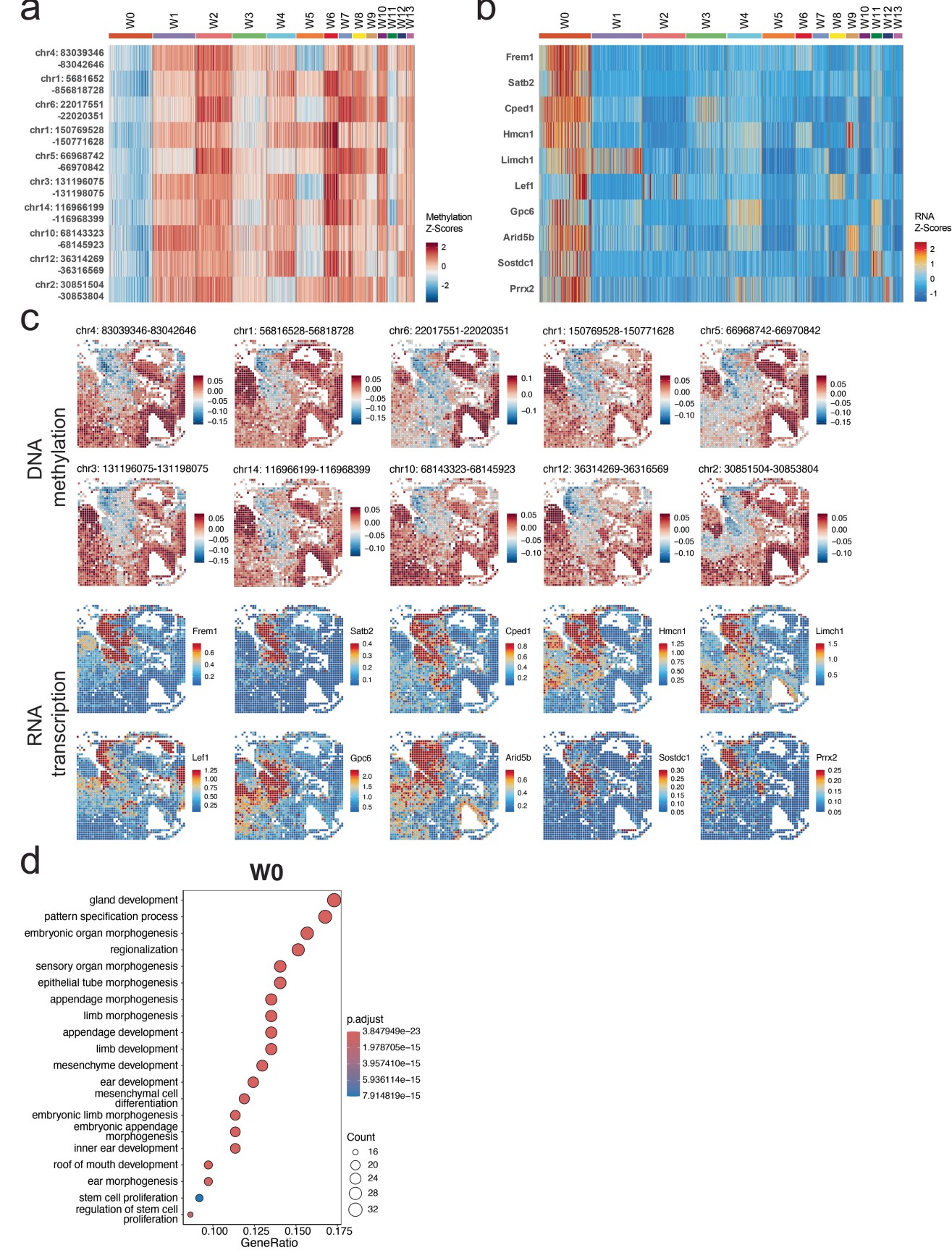

**Extended Data Fig. 6 |** See next page for caption.

**Extended Data Fig. 6 | Spatial mapping of DNA methylation and RNA expression in the craniofacial region of E11 mouse embryo. a**, Heatmap of DNA methylation levels for the top 10 differentially methylated genomic loci in the craniofacial region (cluster W0) of E11 mouse embryo. Each row represents a specific genomic locus, and each column represents a different cluster. The color scale indicates the Z-scores of DNA methylation levels. **b**, Heatmap of expression levels for nearby genes corresponding to the genomic loci in **a**. Each row represents a specific gene, and each column represents a different cluster. The color scale indicates the Z-scores of gene expression levels. **c**, Spatial mapping of DNA methylation and RNA expression levels for selected marker genes in the craniofacial region (W0) of E11 mouse embryo. **d**, GO enrichment analysis from one-sided hypergeometric test for differentially expressed genes in clusters W0. The dot plots show the top GO terms for biological processes enriched in each cluster. The x-axis represents the GeneRatio, and the y-axis lists the GO terms. The size of the dots indicates the count of genes associated with each term, and the color represents the adjusted p-value (p.adjust).

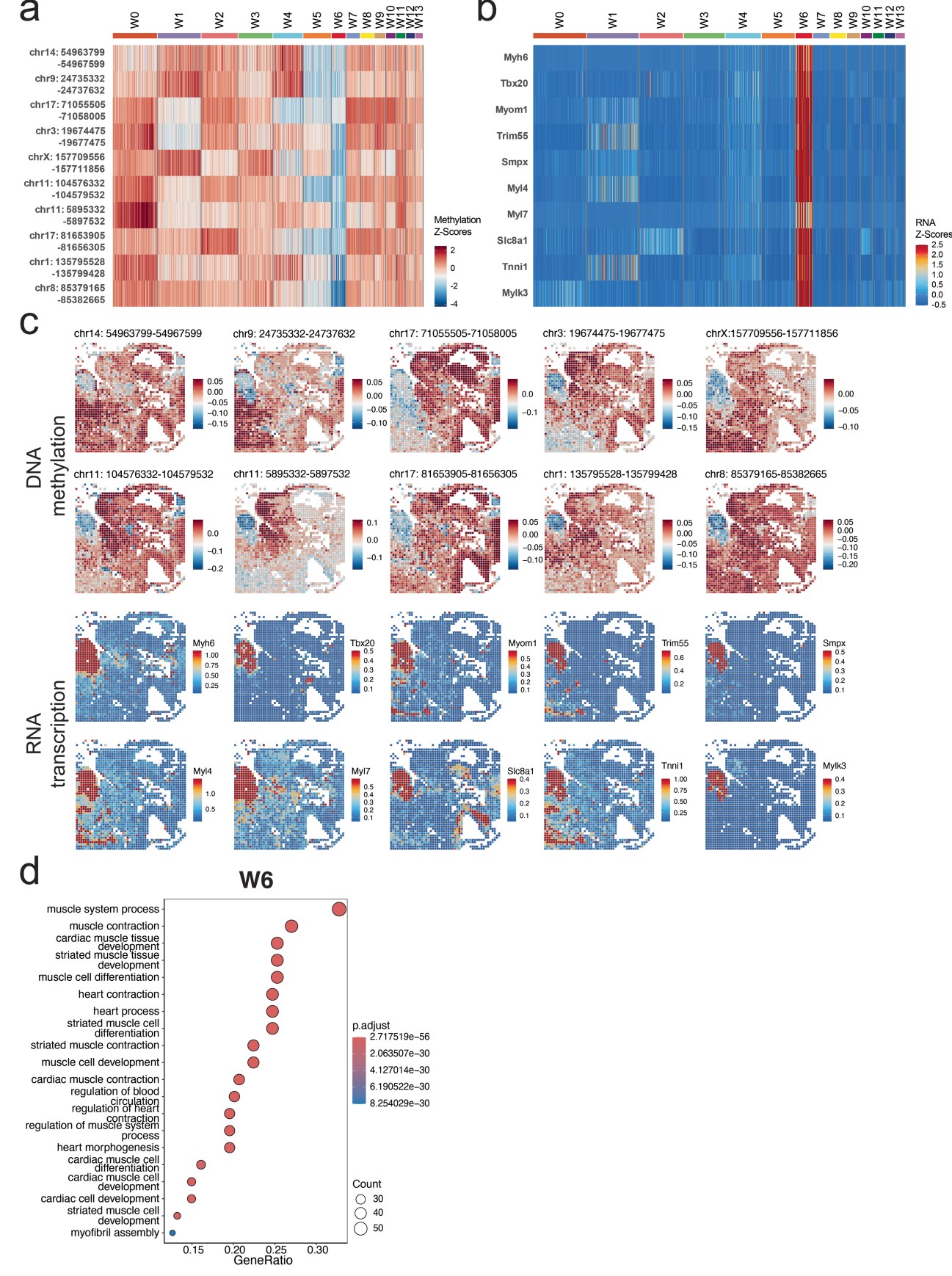

**Extended Data Fig. 7** | See next page for caption.

**Extended Data Fig. 7 | Spatial mapping of DNA methylation and RNA expression in the heart region of E11 mouse embryo. a**, Heatmap of DNA methylation levels for the top 10 differentially methylated genomic loci in the heart region (cluster W6) of E11 mouse embryo. Each row represents a specific genomic locus, and each column represents a different cluster. The color scale indicates the Z-scores of DNA methylation levels. **b**, Heatmap of RNA expression levels for nearby genes corresponding to the genomic loci in **a**. Each row represents a specific gene, and each column represents a different cluster. The color scale indicates the Z-scores of gene expression levels. **c**, Spatial mapping of DNA methylation and RNA expression levels for selected marker genes in the heart region (W6) of E11 mouse embryo. **d**, GO enrichment analysis from one-sided hypergeometric test for differentially expressed genes in clusters W6. The dot plots show the top GO terms for biological processes enriched in each cluster. The x-axis represents the GeneRatio, and the y-axis lists the GO terms. The size of the dots indicates the count of genes associated with each term, and the color represents the adjusted p-value (p.adjust).

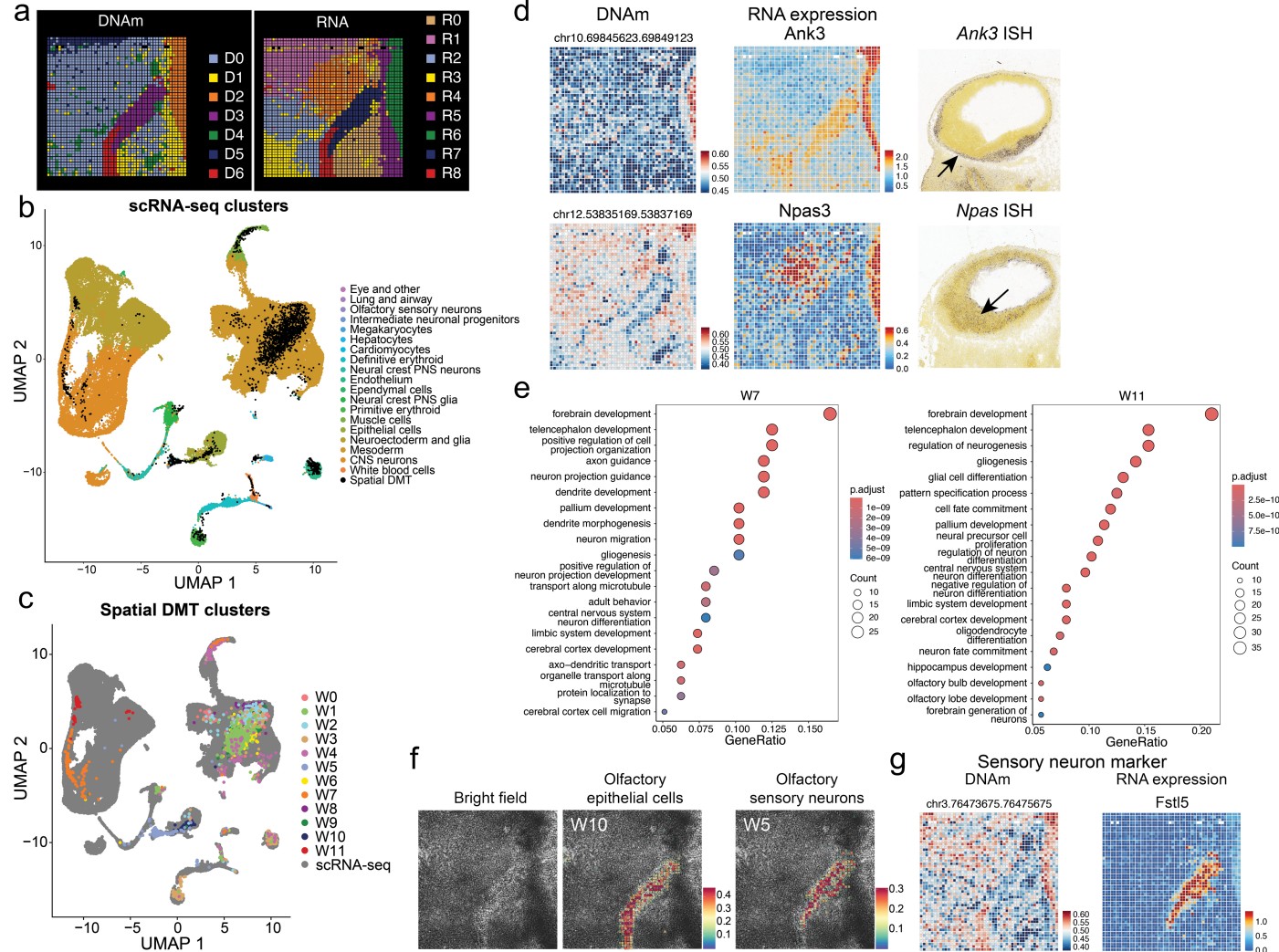

**Extended Data Fig. 8 | Further analysis of DNA methylation and RNA expression in E11 facial and forebrain regions (pixel size, 10 µm). a**, Spatial distribution of all clusters for DNA methylation (left) and RNA transcription (right). **b-c**, Integration of Spatial-DMT RNA data with scRNA-seq data from mouse embryo[25]. Cells are colored according to cell annotations from scRNA-seq data[25] (**b**) and unsupervised clustering results from Spatial-DMT (**c**). **d**, Spatial mapping of DNA methylation, RNA expression levels, and in situ hybridization (ISH) images from the Allen Developing Mouse Brain Atlas[55] show the spatial distribution of selected markers in clusters W7 and W11. **e**, GO enrichment analysis from one-sided hypergeometric test for differentially expressed genes in clusters W7 and W11. The dot plots show the top GO terms for biological processes enriched in each cluster. The x-axis represents the GeneRatio, and the y-axis lists the GO terms. The size of the dots indicates the count of genes associated with each term, and the color represents the adjusted p-value (p.adjust). **f**, Spatial distribution of olfactory epithelial cells and olfactory sensory neurons revealed by cell type decomposition of spatial transcriptomic pixels using a scRNA-seq reference[25]. **g**, Spatial mapping of DNA methylation and gene expression level of *Fstl5* (olfactory sensory neurons marker).

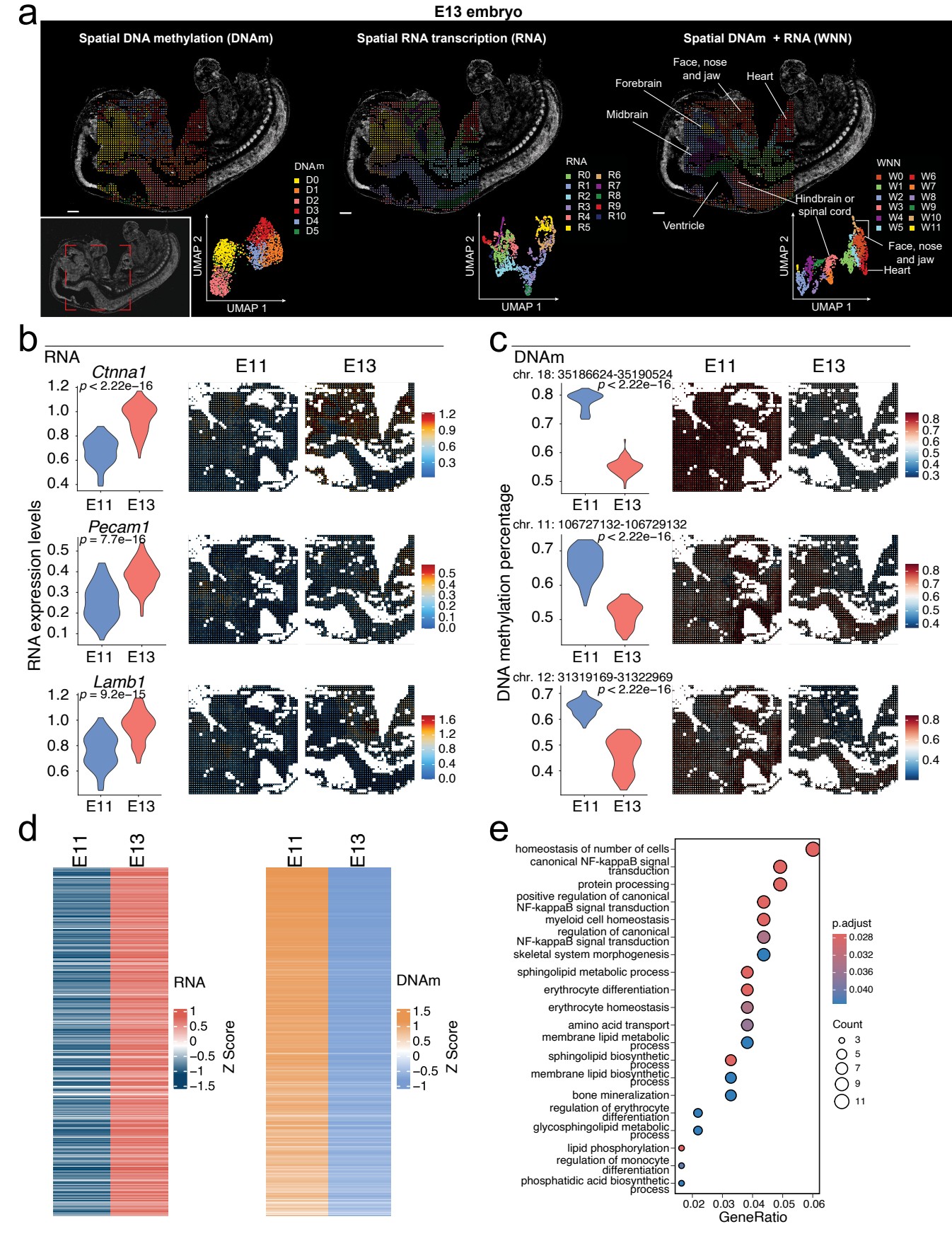

**Extended Data Fig. 9** | See next page for caption.

**Extended Data Fig. 9 | Comparative analysis of DNA methylation and RNA expression between E11 and E13 mouse embryos. a**, Spatial distribution and UMAP visualization of clusters identified from spatial DNA methylation data (left), spatial RNA transcription data (middle), and integrated DNA and RNA data using WNN analysis (right) for the E13 embryo. Each color represents a different cluster, illustrating the spatial distribution of methylation patterns, gene expression profiles, and the integrated data across different regions such as the forebrain, midbrain, hindbrain or spinal cord, heart, and face, nose and jaw region. Scale bars, 500 μm. **b, c**, Comparative analysis of RNA expression (**b**) and DNA methylation levels (**c**) for upregulated genes in E13 heart region, indicating significant changes in methylation and gene expression patterns during embryonic development with two-sided Wilcoxon test. **d**, Heatmaps comparing global RNA expression (left) and DNA methylation levels (right) in brain and spinal cord regions between E11 and E13 embryos. **e**, GO enrichment analysis from one-sided hypergeometric test for biological processes related to demethylated and upregulated genes in the heart region of E13 embryo. The size of the dots indicates the count of genes associated with each term, and the color represents the adjusted p-value (p.adjust).

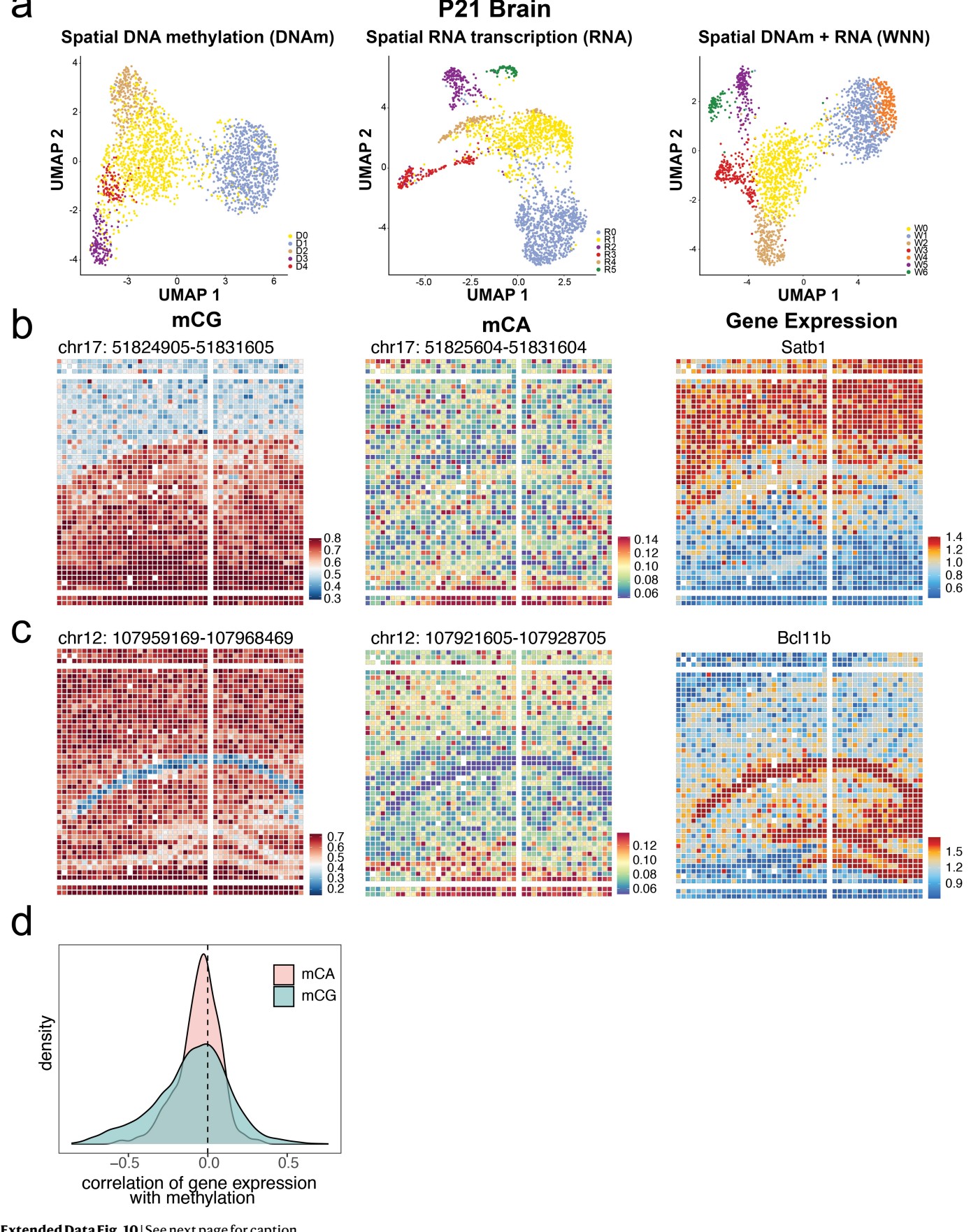

**Extended Data Fig. 10** | See next page for caption.

**Extended Data Fig. 10 | Analysis of DNA methylation and RNA expression in the P21 mouse brain. a**, UMAP visualization of clusters identified from spatial DNA methylation data (left), spatial RNA transcription data (middle), and integrated DNA and RNA data using WNN analysis (right) for the P21 brain. **b**, Spatial mapping of CpG (left) and CpA (middle) methylation levels, and RNA expression levels (right) for *Satb1* in the P21 brain. **c**, Spatial mapping of CpG (left) and CpA (middle) methylation levels, and RNA expression levels (right) for *Bcl11b* in the P21 brain. **d**, Density plot showing the correlation of gene expression with CpA and CpG methylation in the P21 mouse brain. The plot illustrates the distribution of correlation coefficients, with red indicating CpA methylation and green indicating CpG methylation.

# Reporting Summary

## Statistics

For all statistical analyses, confirm that the following items are present in the figure legend, table legend, main text, or Methods section.

| n/a | Confirmed | |
|---|---|---|
| ☐ | ☒ | The exact sample size (*n*) for each experimental group/condition, given as a discrete number and unit of measurement |
| ☐ | ☒ | A statement on whether measurements were taken from distinct samples or whether the same sample was measured repeatedly |
| ☐ | ☒ | The statistical test(s) used AND whether they are one- or two-sided <br> *Only common tests should be described solely by name; describe more complex techniques in the Methods section.* |
| ☐ | ☒ | A description of all covariates tested |
| ☐ | ☒ | A description of any assumptions or corrections, such as tests of normality and adjustment for multiple comparisons |
| ☐ | ☒ | A full description of the statistical parameters including central tendency (e.g. means) or other basic estimates (e.g. regression coefficient) AND variation (e.g. standard deviation) or associated estimates of uncertainty (e.g. confidence intervals) |
| ☐ | ☒ | For null hypothesis testing, the test statistic (e.g. *F*, *t*, *r*) with confidence intervals, effect sizes, degrees of freedom and *P* value noted <br> *Give P values as exact values whenever suitable.* |
| ☒ | ☐ | For Bayesian analysis, information on the choice of priors and Markov chain Monte Carlo settings |
| ☒ | ☐ | For hierarchical and complex designs, identification of the appropriate level for tests and full reporting of outcomes |
| ☐ | ☒ | Estimates of effect sizes (e.g. Cohen's *d*, Pearson's *r*), indicating how they were calculated |

*Our web collection on statistics for biologists contains articles on many of the points above.*

## Software and code

Policy information about availability of computer code

| Data collection | Keyence Imaging System BZ-X800, Illumina NovaSeq 6000 and NovaSeq X Plus system. |
|---|---|
| Data analysis | STARsolo (version 2.7.10b), BISCUIT (version 0.3.14), Seurat package (version 5.1.0), MethSCAn (version 1.0.0), FigR package (version 0.1.0), HOMER (version 4.11), GenomicRanges (version 4.4), clusterProfiler (version 4.2), R (version 4.3.1), RStudio (version 2024.04.0), knowYourCG (version 1.3.15), ggalluvial (version 0.12.5), pheatmap (1.0.12), BZ-X800 1.1.2.4, Slingshot v2.2.1. <br><br> The data analysis pipeline and code to reproduce analyses are available on GitHub (https://github.com/zhou-lab/Spatial-DMT-2024/). |

For manuscripts utilizing custom algorithms or software that are central to the research but not yet described in published literature, software must be made available to editors and reviewers. We strongly encourage code deposition in a community repository (e.g. GitHub). See the Nature Portfolio guidelines for submitting code & software for further information.

## Data

Policy information about availability of data

All manuscripts must include a data availability statement. This statement should provide the following information, where applicable:

- Accession codes, unique identifiers, or web links for publicly available datasets
- A description of any restrictions on data availability
- For clinical datasets or third party data, please ensure that the statement adheres to our policy

Raw and processed data reported in this paper are deposited in the Gene Expression Omnibus (GEO) with accession code GSE270498. Published data for data quality comparison and integrative data analysis include single cell atlas of mouse embryos (https://oncoscape.v3.sttrcancer.org/ atlas.gs.washington.edu.mouse.rna/downloads, https://omg.gs.washington.edu/), mouse brain atlas (http://mousebrain.org/adolescent/downloads.html), and Allen Mouse Brain Atlas (https://developingmouse.brain-map.org/).

## Research involving human participants, their data, or biological material

Policy information about studies with human participants or human data. See also policy information about sex, gender (identity/presentation), and sexual orientation and race, ethnicity and racism.

| Reporting on sex and gender | N/A |
|---|---|
| Reporting on race, ethnicity, or other socially relevant groupings | N/A |
| Population characteristics | N/A |
| Recruitment | N/A |
| Ethics oversight | N/A |

Note that full information on the approval of the study protocol must also be provided in the manuscript.

# Field-specific reporting

Please select the one below that is the best fit for your research. If you are not sure, read the appropriate sections before making your selection.

☒ Life sciences ☐ Behavioural & social sciences ☐ Ecological, evolutionary & environmental sciences

For a reference copy of the document with all sections, see nature.com/documents/nr-reporting-summary-flat.pdf

# Life sciences study design

All studies must disclose on these points even when the disclosure is negative.

| Sample size | No directly relevant. No sample size calculation was performed. Samples sizes were chosen primarily based on experiment length, sequencing costs. The current manuscript mainly described a new method for profiling spatially resolved DNA methylation and Transcription, the sample sizes are sufficient because each sample serves as a proof-of-concept for the new technology. |
|---|---|
| Data exclusions | No data were excluded from the study. |
| Replication | All attempts at replication was successful. For E11 mouse embryo, replicate experiments have been done on adjacent tissue sections to test the reproducibility of the new technology. Other experiments were performed once to serve as a proof-of-concept for the new technology. |
| Randomization | Randomization was not applicable because the focus of this paper is the development of a new spatial multiomics technology for profiling spatially resolved DNA methylation and transcription, it did not involve allocating samples/organisms/participants into experimental groups. |
| Blinding | Blinding was not applicable because the focus of this paper is the development of a new spatial multiomics technology for profiling spatially resolved DNA methylation and transcription, it did not involve group allocation, and by extension, blinding. |

# Reporting for specific materials, systems and methods

We require information from authors about some types of materials, experimental systems and methods used in many studies. Here, indicate whether each material, system or method listed is relevant to your study. If you are not sure if a list item applies to your research, read the appropriate section before selecting a response.

## Materials & experimental systems

| n/a | Involved in the study |
|---|---|
| ☒ ☐ | Antibodies |
| ☒ ☐ | Eukaryotic cell lines |
| ☒ ☐ | Palaeontology and archaeology |
| ☐ ☒ | Animals and other organisms |
| ☒ ☐ | Clinical data |
| ☒ ☐ | Dual use research of concern |
| ☒ ☐ | Plants |

## Methods

| n/a | Involved in the study |
|---|---|
| ☒ ☐ | ChIP-seq |
| ☒ ☐ | Flow cytometry |
| ☒ ☐ | MRI-based neuroimaging |

# Animals and other research organisms

Policy information about studies involving animals; ARRIVE guidelines recommended for reporting animal research, and Sex and Gender in Research

| Laboratory animals | All mice used were on C57BL/6 background. Animal were maintained in 12 h light/12 h dark cycle at room temperatures ranging between 20-25°C and humidities between 40-60%. P21 mouse was used in Spatial-DMT for the co-profiling of DNA methylation and Transcription. |
|---|---|
| Wild animals | No wild animals were used in this study. |
| Reporting on sex | Sex was not important for this study since the tissues were used to benchmark a new genomics protocol, which we anticipate would provide identical results regardless of sex. |
| Field-collected samples | No field collected sample were used in this study. |
| Ethics oversight | Juvenile mouse brain tissue (P21) was obtained from the C57BL/6 mice housed in the University of Pennsylvania Animal Care Facilities under pathogens-free conditions. All procedures used were pre-approved by the Institutional Animal Care and Use Committee. |

Note that full information on the approval of the study protocol must also be provided in the manuscript.

# Plants

| Seed stocks | No seed was use in this study |
|---|---|
| Novel plant genotypes | N/A |
| Authentication | N/A |

