## [Peer Review file · Nature]

Spatial joint profiling of DNA methylome and transcriptome in tissues

Corresponding Author: Dr Yanxiang Deng

Version 1:

Reviewer comments:

Referee #1

(Remarks to the Author)

In this paper, Lee and colleagues developed a new method for joint spatial profiling of DNA methylation and gene expression in tissue sections. The method is an extension of the deterministic barcoding in tissue for spatial omics sequencing (DBiT-seq) approach to include DNA methylation profiling and combine it with spatial RNA detection. The method addresses an important aim of combining profiling of gene expression with epigenetic signatures with spatial resolution and specifically focuses on DNA methylation, which is important for developmental and disease processes. The method itself appears to be well-designed and uses innovative solutions, such as enzymatic C → U conversion and separation of gDNA from cDNA. However, I have serious concerns about three things: 1) the resolution of the method, which appears to be 20-50 μm and is below the original DBiT-seq approach (10 μm) and is not at the scale of single cells, 2) quality of the data, especially of RNA, and as an extension, the ability to resolve cell types and 3) the use of this method in the paper to discover truly biologically novel and significant findings. Below are my specific comments and questions:

1) Resolution of the method. Why are authors using 20-50 μm resolution, when the DBiT-seq used 10 μm and was able to resolve single cells? The way barcodes are applied is essentially the same. This greatly limits biological discovery with this method.

2) Quality of the RNA data. Looks like there is a significant drop in the number of genes/spot compared to DBiT-seq method, which detected ~4000 genes/spot for 50 μm. The authors need to compare these methods in the paper and explain why there is a drop in detection (which is expected due to additional treatment and steps to obtain DNA methylation readouts). In addition, a comprehensive comparison of RNA detection metrics with other spatial transcriptomics methods would be important.

3) As the extension of the previous two points, the UMAP plots and resolution of clusters and cell types does not look as good as for DBiT-seq. The clusters in this paper do not look very resolved. The authors should try to co-cluster their data with DBiT-seq data and discuss how the datasets compare. Also, a comparison to single-cell datasets is important, but can be complicated by a low resolution (20-50 μm).

4) The duplication rate seems pretty high. How does it compare to single-cell methylation data? Also, I would expect much higher non-CG rates in postnatal cortex, which has been widely reported.

5) The labels in Extended Data Figure 2g are not clear. Where is the TSS on this plot?

6) Much more analysis can be done on the P21 brain, especially for the cortex. The authors should check for markers of neurons (Syt1, Rbfox3), oligos (Mbp, Plp1), fibrous astrocytes (Gfap), upper layer neurons (Cux2, Cux1, Satb2), deep-layer neurons (Bcl11b [right now it is shown mostly in the hippocampus], Tle4). These markers should be checked both in the RNA and promoter methylation data. They should be shown in the paper as proof that the method works since they are very spatially restricted in the cortex. The authors can also check for markers of other brain regions, such as striatum and thalamus, if they are in the tissue section they analyzed.

7) The biological insights from the analyses in the paper are lacking. Are the new specific developmentally regulated enhancers that were discovered using this method that control subcortical region (e.g. specific nuclei, subregions of the striatum and thalamus that are hard to dissect and resolve with single-cell sequencing) and cortical layer specification? How about the role of non-CG methylation compared to CG methylation in regulating gene expression in spatially restricted brain

regions and laminae?

Referee #2

(Remarks to the Author)

Summary of the Paper:

The paper introduces a new spatial multi-omics technology capable of joint profiling of the DNA methylome and transcriptome at near-single-cell resolution. This technology was applied to E11, E13 mouse embryos, and P21 mouse brains. The key novelty lies in its ability to simultaneously capture both RNA expression and DNA methylation, offering insights into the complex interplay between these two modalities. However, the lack of new biological insights resulted from such co-profiling and the lack of clear benchmarking against previously published technologies are major limitations of the paper, and there are a few significant concerns to be addressed:

Main Comments and Suggestions:

1. Technical Workflow and Methodological Considerations:

Use of HCl for Tn5 Tagmentation:

The authors used HCl to increase genomic DNA accessibility, followed by extensive Tn5 tagmentation prior to mRNA capture. However, this procedure may compromise RNA quality as HCl is known to reverse cross-linking in PFA-fixed samples. This concern is supported by the lower number of UMIs per pixel observed in their RNA data compared to the original DBiT-seq paper (Liu et al., 2020).

Suggestions:

Discuss whether the joint profiling of two modalities compromises the data quality of individual modalities by comparing the RNA data in spatial-DMT with DBiT-seq.

Elaborate on how much of the genome becomes accessible post-treatment. Are there specific genes or regions that remain inaccessible? Provide data to address these questions.

Evaluate if the treatment affects the transcriptome or DNA methylation, and if so, to what extent.

Consider adding experiments or data to clarify these concerns.

Choice of 50 μ m Microfluidic Device:

The use of a 50 μ m microfluidic device for E11 and E13 mouse embryos results in each pixel covering multiple cells, thus not achieving single-cell resolution.

Suggestions:

Discuss why a smaller, 10 μ m microfluidic device was not used to achieve closer-to-single-cell resolution. Address how averaging between neighboring cells may affect data interpretation and provide insights into potential biases introduced by this approach.

2. Analytical Limitations and Data Interpretation:

Simultaneous Profiling of 5hmC and 5mC:

While the authors claim that their method profiles 5hmC and 5mC simultaneously using TET oxidation followed by APOBEC deamination, the chemistry used cannot distinguish between endogenous 5fC, 5caC, 5hmC, and 5mC.

Suggestions:

Include a discussion on these limitations and the potential impact on data interpretation.

Correlation Between DNA Methylation and Gene Expression:

Different genes exhibit varying correlations with DNA methylation. The lack of differentiation between various methylation states (5mC and 5mC oxidation) could lead to misinterpretation.

Suggestions:

Conduct a more detailed analysis to distinguish the effects of different methylation states on gene expression.

Explore whether positive correlations observed are due to the presence of 5hmC, 5fC, or 5caC, which are generally associated with DNA de-silencing.

3. Novelty and Contribution:

Importance and Benefits of Spatial Multi-Omics Technology:

While the paper emphasizes the ability to co-profile DNA methylation and transcriptome, the authors should better highlight the novelty and significance of their work compared to previous studies.

Suggestions:

Provide a clearer comparison with the previous spatial epigenome-transcriptome co-profiling study, particularly focusing on the new contributions made by the ability to map DNA methylation spatially.

Emphasize the potential benefits of this technology over traditional single-cell profiling strategies, such as identifying novel spatial patterns and region-specific regulatory mechanisms.

Comparisons with Public Datasets:

Given the novel nature of this technology, validating these findings against publicly available datasets would add credibility.

Suggestions:

Integrate publicly available datasets to validate or benchmark the observed patterns, especially for differentially regulated genes in the brain.

4. Data Presentation and Clarity:

Workflow and Figures:

The use of multiple figures, such as Figure 1B, lacks clarity and requires revision for better interpretation.

Suggestions:

Revise Figure 1B for better clarity. Include a more detailed explanation of the “retention rate” in Figure 1.

Address why the number of CpGs detected is significantly lower compared to other studies (e.g., Hernando 2019 and Liu 2021).

Provide a clear definition for terms like “variably methylated regions” and “retention rate” to enhance readability.

5. Spatial and Temporal Resolution:

Spatial Resolution and Single-Cell Analysis:

The resolution of 20-50 μm does not fully achieve single-cell resolution, which needs to be clearly addressed.

Suggestions:

Discuss the potential to achieve improved spatial resolution and the limitations imposed by the current setup.

Provide details on the number of tissue slices and replicates analyzed for reproducibility. Address if the findings are based on a single sample or multiple biological replicates.

Pseudo-time Analysis:

The terms “temporal” and “dynamics” are used frequently, despite the absence of time-resolved data.

Suggestions:

Use more appropriate terminology that reflects the pseudo-temporal nature of the data.

6. Integration of Multi-Modal Data:

Joint Analysis of DNA Methylation and Transcriptome Data:

The authors showed that different genes in the mouse brain are regulated by mCG, mCA, or both. However, they do not provide a thorough explanation for why this regulation differs.

Suggestions:

Compare or integrate these findings with other datasets to validate the distinct roles of mCA and mCG in epigenetic regulation.

Conduct a more detailed analysis on why specific genes are differentially regulated by mCG or mCA.

7. Specific Figures and Data Points:

Figure 1F:

The authors overlay their spatial omics data with a histological image of a mouse embryo, but it's unclear how well the results align with known anatomy.

Suggestions:

Quantify the alignment of spatial data with known anatomical features.

Figure 4B, 4F, 4G, and 4H:

These figures display RNA and DNA methylation data integration, yet some expression patterns appear random.

Suggestions:

Provide further explanation or possible hypotheses for these seemingly random patterns.

Include benchmarking with known datasets to confirm the biological relevance of these clusters.

8. Additional Considerations:

Replicability and Statistical Analysis:

The lack of replicates or clear statistical validation is a significant gap.

Suggestions:

Include more biological replicates and detailed statistical analysis for all key findings to enhance reproducibility.

Referee #3

(Remarks to the Author)

In this manuscript, Lee & Fu et al., present spatial-DMT for spatial co-profiling of DNA methylation and transcriptome. The authors spent efforts to combine existing approaches for 5mC and transcriptomics measurement to enable joint spatial profiling of 5mC and RNA, which is certainly of high interest. The authors demonstrated the utilities of this technology to mouse embryogenesis and postnatal brain and revealed spatial context of known methylation and expression patterns. While the authors showed this new technology is able to identify the concordant and distinct DNA methylation and gene expression relationships in different spatial contents, the current manuscript lacks some necessary information from both technical and analytical aspects. Specific comments are listed below.

As a manuscript reporting a new technology, systematically (quantitative) evaluation and benchmarking is required. Some concerns about the specificity of this method are from lack of control or technical details. For example,

1. One key information for DNA methylation analysis – conversion rates – is missing throughout the manuscript. The accurate calculation of conversion rates can be performed on synthetic spike-in DNA with known 5mC sites. In addition, in

Line 130 and Fig. 1d, it is unclear what is “retention rate”. Is this meant to be “modification level”?

2. Did the authors performed technical replicates to demonstrate reproducibility?

3. Another missing piece is the comparison with public/reference datasets. For example, how many identified 5mC sites are also supported by previous methods? What is the genome-wide correlation for 5mC modification levels? This is particularly information is no positive control (spike-in DNA) can support the successful conversion of 5mC bases. The comparison of gene expression levels detected by this methods with prior methods is also lacking.

4. The authors used HCl to disrupt nucleosome to minimize biased tagmentation, which is supported by Fig. S2e. However, there is still non-negligible enrichments toward heterochromatin and depletion from enhancers. What background is used for calculating odds ratio, reference genome annotation or real data from WGS? If the reference annotation is used as background model, a positive control from public whole genome sequencing data should added to this panel.

5. Will HCl treatment affect the diffusion pattern of DNA and RNA?

6. In Lines 126-128 and Fig.1c, the comparison is unfair. For example, Shareef 2021 is expanded RRBS which was designed to capture only a subset of the genome but not whole genome as in this method. Also, the numbers from Nichols 2022 is inconsistent with the publication: sciMETv1.LA was reported to have 2,158,578 CG and sciMETV2.SL have 325,034-534,728 CG sites - which are 2-10 folds of the coverage by this method.

7. In addition, the authors should clearly note that there could be multiple cells in a 20um or 50um pixel, and the numbers from reference data are from single cells. Did the author observed 6.25-fold of #CG for 50um compared to 20um $((50/20)^2)$?

8. What is the cutoff in calling a 5mC site? As a pixel could capture fragments from multiple cells, are there sites methylated in one cell but unmethylated in another cell within one pixel?

9. Another technical concerns about the purity of DNA modality. Can streptavidin beads pull down complete remove cDNA from supernatant that will be used for DNA library preparation? Since a SLP5 adaptor ligation is used for ssDNA library prep after APOBEC treatment, will this SLP5 adaptor also ligated to cDNA that remained in DNA library (if the beads pull down is not 100.00%)? A simple way to check is the calculate the percentage of reads that start with a TSO sequence in DNA library.

10. Lines 86-87, the authors stated 30 min - 1 hour tagmentation for twice is the optimized condition for improving gDNA yield. Can the authors provide the comparisons of different conditions during their optimization. This information could be useful for readers as to balance yield and experimental time.

There are multiple places that could be improved from the analytical aspect. For example:

11. A general comment is that the histological images could be provided side-by-side with molecular profiles when applicable.

12. Lines 210-212, Fig2. abc and Fig. 3ef. It is inappropriate to compare the expression levels of genes with the overlapping VMR. There is no evidence the given VMR is regulating this gene, and what is there are multiple VMR overlapping with one gene but showing different methylation levels?

13. In Fig. 1f, while the inconsistency of clustering between 5mC and RNA (typically lower resolution in 5mC) is reasonable, but there is a subset of the cells was grouped as one cell type (R3) by RNA, but it was combined with R2 and classified as D0 in DNA (D0 and D4). What is the biological basis of RNA modality have higher resolution for D4 cells (R1, R3 and R12) but lower resolution for D0 cells (R2 & R3)? A consensus matrix can be provided to show the DNA-RNA clustering relationships more clearly.

14. Fig. 1f, there are a few straight horizontal lines in DNA spatial distribution that grouped as D3. Is this technical artifact or true biology?

15. What is the genome distribution patterns of VMR? Did the CG and CA VMR separately calculated? What are the statistics in calling these VMRs?

Minor:

1. The size of slides/tissue sections should be clearly labeled.

2. Line 194, TET2 full name is ten-eleven translocation methylcytosine dioxygenase 2.

3. Unit label are missing from multiple figure panels, for example, Fig. 2c, 3e,f, h.

4. Line 235, are there any literature support for the statement that Dnmt1 expression are positively associated with mitotic division?

5. Lines 244-246, Fig. S8d cannot support this statement.

6. Lines 272, Fig. 4h did show mCG-dependent RNA expression level.

Version 2:

Reviewer comments:

Referee #1

(Remarks to the Author)

In the resubmission of this paper by Lee et al, the authors have responded to most comments from me and other reviewers and added additional data and analyses. These additional details, figures and data alleviate some of my concerns, and I commend the authors for being responsive and comprehensive in addressing the comments.

However, I still have significant concerns about the quality of the data that the method can produce, some of which are actually highlighted by the additional data the authors provide. My concerns are especially about the quality of the RNA data, which does not seem to be very useful to identify well-known cell types that should be resolved due to their spatial distribution. Additionally, 10um experiments should provide near single-cell resolution, but the paper does not show that specific cell types can be identified from RNA-seq data. Also, there is no comparison of the DNA methylation data to sc DNA methylation data (there are several datasets available). More details below:

1. I have concerns with the quality of RNA data, especially for the postnatal brain. From Extended Data Fig. 10c, it is evident that the spatial clusters do not map well on the scRNA-seq data, with spatial clusters distributed widely across scRNA-seq clusters. It is in contrast to similar data from Zhang et al., Nature, 2023 (Figure 2d, 5c), which the authors cite, which shows much better correspondence between scRNA-seq and spatial clusters.
2. I did not see any data with UMAP/clustering of spatial P21 data to show that at least the main cell types can be identified (oligos, deep-layer neurons, granule neurons of the hippocampus etc).
3. The paper is almost fully focused on embryonic mouse analysis. It gives the impression that the technique only works well with this particular sample type, with very little data from other ages and specific brain regions. This is in contrast to similar papers that demonstrate the performance of their method with several types of samples. Using the method to analyze cortical lamina (the entire span of the cortex), as well as striatum or thalamus, in addition to the hippocampus would alleviate some of these concerns.
4. There is no comparison or integration of the spatial DNA methylation data with publicly available single-cell DNA methylation data. Without these analyses, it is impossible to judge whether the method truly works. Such analyses have been done with Spatial ATAC.

Referee #2

(Remarks to the Author)

Thanks for addressing my comments and questions satisfactorily. I do not have further concerns.

Referee #3

(Remarks to the Author)

The authors' responses addressed most of the issues, including quality control of the data, reproducibility of techniques, and consistency of published data. However there are still several issues.

Comments 4: While scWGBS data is a relevant comparator for methylome profiling, it may not adequately control for tagmentation or fragmentation biases, since it also suffers from sparse coverage and noise. Could you clarify why bulk WGS was not included as a control in your enrichment panel, and whether such a comparison might better reflect the expected uniform coverage? You interpret log2 odds ratios within ± 1 as indicating "non-significant" enrichment or depletion. Could you elaborate on whether statistical testing was performed to support this threshold, rather than relying on visual inspection alone?

Comments 8: Thank you for the clarification. However, I still have some concerns regarding the interpretation of continuous methylation values in the context of low cell numbers per pixel. Given that each pixel captures limited DNA fragments from only a few cells, the total read count per site is likely low and each CG site per cell can even only be either methylated or unmethylated. In this scenario, continuous values (e.g., 0.25, 0.5, 0.75) may simply reflect stochastic sampling noise rather than true biological heterogeneity.

Comment 13: Although DNAm and RNA-based cell identity do not have a simple subset-superset relationship - but there must be biology behind the observation. For example, epigenetic primed sub-population but share similar transcriptional states, and sub-types of distinct transcriptional states not divided by DNAm (but other epigenetic mechanisms). From the confusion matrix looks there are quite a few differences between the two modalities. One possibility is RNA signal has much larger signal range (e.g., expression level from 0 to thousands of transcripts per cell) but DNAm is nearly binary. One suggestion is to highlight the advantage of using WNN for spatial-multiomics analysis - for example, can you estimate the fraction of cell types within the pixel based on WNN and independent DNAm/RNA clustering? The confusion matrix should be included in the manuscript, along with quantitative comparison between WNN-based clustering and single-modality clustering results, which will be helpful to the readers.

Version 3:

Reviewer comments:

Referee #1

(Remarks to the Author)

Thank you for clarifying my questions and for additional analyses. I have no further questions, all issues I raised have been resolved.

Referee #3

(Remarks to the Author)

Thank you for your detailed responses to all my review questions. I don't have any questions.

Spatial joint profiling of DNA methylome and transcriptome in mammalian tissues

Responses to reviewers' comments

Overall summary of the revision:

We sincerely thank all the reviewers for their thoughtful and constructive feedback on our manuscript. In response, we have conducted additional experiments and analyses, and made substantial revisions that we believe fully address all concerns and significantly improve the overall quality of the work. Please find below a summary of the key revisions:

- Conducted a new experiment at 10 μm resolution, demonstrating near single-cell profiling capability in the E11 craniofacial and forebrain regions.
- Performed a technical replicate of the E11 sample, confirming the reproducibility of both DNA methylation and gene expression measurements.
- Performed additional benchmarking with publicly available single cell methylation, spatial transcriptomics, and single-cell RNA-seq datasets, supporting the robustness of our methylation and expression data.
- Enhanced our analysis of biological insights, including spatially regulated methylation patterns, partially methylated domains, and the relationship between methylation and gene expression across embryonic and postnatal brain regions.
- Addressed all methodological and analytical concerns, including spatial resolution limits, integration with public datasets, and the interpretation of variably methylated regions (VMRs).

Point-by-point responses to individual reviewer comments, along with corresponding updates in the revised manuscript, are provided below.

Table of Contents:

Referee #1	2
Referee #2	19
Referee #3	50
References	73

Referee #1 (Remarks to the Author):

In this paper, Lee and colleagues developed a new method for joint spatial profiling of DNA methylation and gene expression in tissue sections. The method is an extension of the deterministic barcoding in tissue for spatial omics sequencing (DBiT-seq) approach to include DNA methylation profiling and combine it with spatial RNA detection. The method addresses an important aim of combining profiling of gene expression with epigenetic signatures with spatial resolution and specifically focuses on DNA methylation, which is important for developmental and disease processes. The method itself appears to be well-designed and uses innovative solutions, such as enzymatic C → U conversion and separation of gDNA from cDNA. However, I have serious concerns about three things: 1) the resolution of the method, which appears to be 20-50 μm and is below the original DBiT-seq approach (10 μm) and is not at the scale of single cells, 2) quality of the data, especially of RNA, and as an extension, the ability to resolve cell types and 3) the use of this method in the paper to discover truly biologically novel and significant findings. Below are my specific comments and questions:

Response:

We would like to thank the reviewer for the positive feedback regarding our work. We have taken the concerns seriously and provided additional data and analysis to address these reservations regarding data quality and significance. Below is our response to the reviewer's specific comments and questions.

1. Resolution of the method. Why are authors using 20-50 μm resolution, when the DBiT-seq used 10 μm and was able to resolve single cells? The way barcodes are applied is essentially the same. This greatly limits biological discovery with this method.

Response:

We thank the reviewer for their comment on our method's resolution. In the first submission, we only showed data with 20 to 50 μm pixel resolution primarily to establish the feasibility of the technology. As the reviewer suggested, the barcode delivery methods are the same under different pixel resolutions. However, the DNA input is lower for smaller pixel sizes, which makes DNA recovery and data analysis more challenging. Although 10 μm pixel resolution avoids mixing cells of different epigenetic identities and offers cleaner data, they are limited in areas mapped. For instance, broader coverage at 50 μm resolution may be sometimes preferred for studying extensive regions such as whole embryos, whereas higher resolutions (e.g., 20 μm or 10 μm) are used to enable detailed analysis of localized regions.

Additionally, the number of cells captured per pixel varies based on several tissue-specific characteristics, including tissue species (e.g., human versus mouse tissues), cell density (e.g., densely populated hippocampal regions versus less dense cortical regions), cell size (e.g., smaller lymphocytes compared to larger neurons), and distribution and organization of the region of interest within the tissue. Therefore, we conclude that a bigger pixel size has its own application scenario, and the choice of resolution depends on the specific biological questions investigated.

Having established spatial co-mapping of DNA methylation and RNA transcriptome profiles on 20-50 μm , we followed the reviewer's suggestion to evaluate Spatial-DMT to finer pixel resolutions. We performed experiments at a 10 μm resolution, specifically mapping the craniofacial and forebrain regions of an E11 embryo (Fig. 1g-h, Extended Data Fig. 10). This higher resolution allowed us to resolve finer cellular subtypes that were challenging to distinguish using a 50 μm chip. A dedicated results section (Spatial co-profiling of DNA methylome and transcriptome in mouse embryo) discussing these new findings from the 10 μm analysis has now been included, underscoring that our Spatial-DMT technology is adaptable and capable of achieving higher resolution as needed.

“To precisely resolve the fine structure of transcriptional regulation at the craniofacial and forebrain regions of the E11 embryo, we employed the 10 μm pixel-size microfluidic chip to produce a near single-cell resolution spatial map (Fig. 1g, h and Extended Data Fig. 10a). Integration of our spatial dataset with a single-cell RNA sequencing (scRNA-seq) reference¹ from mouse embryo revealed a strong concordance between the two datasets (Extended Data Fig. 10b, c). Cell-type deconvolution using the scRNA-seq reference¹ identified distinct clusters that correspond precisely to known anatomical structures of the developing mouse brain. Notably, two spatially defined clusters, W7 and W11, captured key telencephalic compartments. W11 was enriched for telencephalon progenitors in the ventricular zone of the pallium, a neurogenic niche characterized by active cell division and proliferation, while W7 corresponded to GABAergic cortical interneurons localized in the mantle zone, where newborn neurons migrate, accumulate, and differentiate to establish cortical architecture²⁻⁴ (Fig. 1h and Extended Data Fig. 10d). Gene ontology (GO) enrichment analysis further supported these regional identities, highlighting biological processes associated with neurogenesis and progenitor proliferation in W11, and neuron projection and migration in W7 (Extended Data Fig. 10e). Moreover, the cell-type lineage tree constructed from the scRNA-seq reference confirmed the developmental trajectory, positioning telencephalon progenitors as direct precursors to GABAergic cortical interneurons¹. Beyond the forebrain, our spatial analysis also resolved refined sensory structures within the developing olfactory system (cluster W5 and W10) (Fig. 1g and Extended Data Fig. 10a). Sensory neurons were notably enriched in W5 (Extended Data Fig. 10f, g), spatially localized adjacent to the

forebrain. This spatial pattern closely aligns with the established developmental trajectory of the olfactory system, wherein olfactory sensory neurons progressively form connections with the forebrain as embryogenesis progresses⁵⁻⁷. Together, these findings provide a high-resolution view of neural and sensory system formation, underscoring the power of Spatial-DMT in resolving intricate anatomical structures and capturing the spatiotemporal dynamics.”

Fig. 1g-h:

Extended Data Fig. 10:

2. Quality of the RNA data. Looks like there is a significant drop in the number of genes/spot compared to DBiT-seq method, which detected ~4000 genes/spot for 50 μm . The authors need to compare these methods in the paper and explain why there is a drop in detection (which is expected due to additional treatment and steps to obtain DNA methylation readouts). In addition, a comprehensive comparison of RNA detection metrics with other spatial transcriptomics methods would be important.

Response:

We thank the reviewer for their concern regarding the quality of the RNA data from Spatial-DMT. In brief, we think our RNA data quality is comparable to that of previously published datasets, including DBiT-seq. The observed difference in the number of detected genes is likely attributable to biological variation between samples. Below we outline the additional analysis supporting this interpretation. First, we have directly compared our data with published DBiT-seq transcriptome (Liu et al., *Cell*, 2020)⁸. New analyses are presented in Fig. 1e and Supplementary Table 5. The number of genes and unique molecular identifiers (UMIs) detected per pixel in our method is comparable to those reported in prior datasets profiling the similar tissue and pixel resolution. For example, when both profiling E11 embryo tissues, our 10 μm data detected more genes per pixel than the 25 μm data by DBiT-seq, despite the smaller pixel size (Fig. 1e). Second, the only published dataset that has slightly higher gene detection than Spatial-DMT was based on DBiT-seq profiling the entire E10 mouse embryo. In comparison, our study only profiled half of an E11 embryo. Given that earlier-stage embryos typically have higher cell density and more active gene expression, each 50 μm pixel in the E10 dataset would be expected to capture more transcripts. Therefore, we think that the difference in developmental stage likely accounts for the variation in gene detection.

Lastly, we compared our data with another spatial co-profiling method that combines spatial gene expression with ATAC-seq (Di et al., *Nature*, 2023)⁹. The analysis shows comparable RNA data quality under the same resolution and similar sample types (E13 mouse embryo and P21 mouse brain). For example, when both profiling E13 embryo tissue using 50 μm pixel sizes, our data detected more genes per pixel than Di et al. 2023⁹ (Fig. 1e). Taken together, these observations suggest that, despite the additional treatments involved in our method to enable simultaneous DNA methylation profiling, the quality of our RNA data remains unaffected.

Fig. 1e:

3. As the extension of the previous two points, the UMAP plots and resolution of clusters and cell types does not look as good as for DBiT-seq. The clusters in this paper do not look very resolved. The authors should try to co-cluster their data with DBiT-seq data and discuss how the datasets compare. Also, a comparison to single-cell datasets is important, but can be complicated by a low resolution (20-50 μm).

Response:

We thank the reviewer for suggestions regarding additional comparison and co-clustering with DBiT-seq data, and the idea of comparison with single-cell datasets. It is important to note that the original UMAP plots in the DBiT-seq study combined multiple samples (Liu et al., *Cell*, 2020, Fig. 5)⁸, potentially enhancing the apparent clustering resolution. When comparing our data with only individual sample plots from the DBiT-seq paper (e.g., Figures 2C, 6C, and 6D)⁸, our clustering and cell-type resolutions are comparable.

Direct co-clustering our data with DBiT-seq is challenging due to differences in the tissue mapped (different developmental stages and anatomical regions), and inherent focus of the two technologies (RNA vs DNA methylation). However, we followed the reviewer's suggestion to compare Spatial-DMT with other, more recent spatial datasets and single-cell RNA seq data.

1) Co-clustering with Spatial-ATAC-RNA data: We integrated our E13 embryo dataset (50 μm resolution) with a comparable Spatial ATAC-RNA sequencing dataset (Di et al., *Nature*, 2023)⁹. The integrated data (Extended Data Fig. 12) demonstrates well-resolved spatial domains and clear, coherent cell-type clustering across both datasets.

2) Integration with single-cell RNA-seq data: We further integrated our P21 brain sample (20 μm resolution) with a publicly available mouse brain single-cell RNA-seq dataset, which resulted in consistent and refined clustering (Extended Data Fig. 15). Previous work indicates that a 20 μm resolution can approach single-cell resolution by capturing 1–3 cells^{9,10}.

3) Additional validation at higher spatial resolution: To provide additional evidence of spatial resolution capabilities, we included a new 10 μm resolution experiment from the E11 mouse embryo craniofacial and forebrain region and integrated it with a single-cell RNA-seq dataset (Extended Data Fig. 10b-c). These comprehensive analyses further confirm the quality, robustness, and accuracy of our Spatial-DMT approach.

Extended Data Fig. 12:

Extended Data Fig. 15:

Extended Data Fig. 10b-c:

4. The duplication rate seems pretty high. How does it compare to single-cell methylation data? Also, I would expect much higher non-CG rates in postnatal cortex, which has been widely reported.

Response:

Thank you for your comments regarding the duplication rate and non-CG methylation levels. Read duplication and reduced library complexity are indeed a limitation of most single-cell DNA methylation profiling methods, as DNA is allelic and subject to dropout in single-cell experiments. However, we employed multiple measures, including enzymatic conversion, to mitigate DNA loss during library preparation. To evaluate the duplication rate, we compared our data with recent single-cell methylation methods such as Cabernet¹¹ and sciMETv2¹² using both mouse embryo and human brain datasets. As shown in Extended Data Fig. 4b, our Spatial-DMT datasets generally have lower duplication rates than Cabernet and sciMETv2 when it was based on splint-ligation (SL), a technique shared by Spatial-DMT. One sample showed lower duplication rates than sciMETv2 with linear amplification, which is more protective of DNA but uses a more involved procedure, despite targeting different biological tissues. Based on these comparisons, we think that Spatial-DMT yields a duplication rate comparable to single-cell methylation approaches.

Extended Data Fig. 4b:

Prior bulk and single-cell studies of mouse brain indicated that global non-CpG methylation levels in the mouse brain range from approximately 0.2% to 7.6%, with around 3% specifically in isocortex neurons (Liu et al. *Nature*, 2021¹³, Lister et al. *Science*, 2013¹⁴). Similarly, a review by Kinde et al. PNAS 2015¹⁵ estimated the % CpH to range from 2-6% in postnatal mouse brains. In our study, the CpA methylation rate average around 3% in the P21 brain sample (Fig. 1d), which aligns with previous reports.

It is important to note that the non-CG methylation rates vary by developmental stage and cell type in the postnatal cortex¹⁴. To validate our non-CG methylation rates stratifying cell type and regional context, we have included a global CpA methylation level plot across pixels in Extended Data Fig. 4d. This analysis demonstrates a higher CpA retention rate in the cortex compared to the hippocampus, highlighting expected regional differences in cell composition with more neuron populations in the cortex. In addition, in the dentate gyrus (DG) region, we observed a CpA methylation rate of approximately 2%, consistent with previously reported values¹³.

5. The labels in Extended Data Fig. 2g are not clear. Where is the TSS on this plot?

Response:

Thanks for the comments regarding our lack of clarity. We have updated the plot explicitly labeling TSS in Extended Data Fig. 5c.

Extended Data Fig. 5c:

6. Much more analysis can be done on the P21 brain, especially for the cortex. The authors should check for markers of neurons (Syt1, Rbfox3), oligos (Mbp, Plp1), fibrous astrocytes (Gfap), upper layer neurons (Cux2, Cux1, Satb2), deep-layer neurons (Bcl11b [right now it is shown mostly in the hippocampus], Tle4). These markers should be checked both in the RNA and promoter methylation data. They should be shown in the paper as proof that the method works since they are very spatially restricted in the cortex. The authors can also check for markers of other brain regions, such as striatum and thalamus, if they are in the tissue section they analyzed.

Response:

We thank the reviewer for these constructive comments. We have now included additional marker analysis including neurons (Syt1, Rbfox3), oligodendrocytes (Mbp, Plp1), fibrous astrocytes (Gfap), upper-layer neurons (Cux2, Cux1, Satb2), and deep-layer neurons (Bcl11b), as suggested by the reviewer. Both RNA expression and promoter DNA methylation levels for these markers demonstrate expected spatial variation consistent with known brain biology (Extended Data Fig. 14). To discuss these findings, we wrote the following in the Results section (“Spatial co-mapping of mCH, mCG, and RNA transcription in the postnatal mouse brain”):

"Further analysis of neuronal and glial populations revealed cell-type- and region-specific transcriptomic and epigenetic variation. For example, Syt1 and Rbfox3 are broadly expressed across neurons in all cortical layers (Extended Data Fig. 14a). In contrast, Cux2, Cux1, and Satb2 were highly expressed in upper-layer neurons, whereas Bcl11b was enriched in the deeper cortical layers. Oligodendrocytes (Mbp and Plp1) and fibrous astrocytes (Gfap) were specifically enriched in the corpus callosum and hippocampal regions (Extended Data Fig. 14b, d)."

Extended Data Fig. 14:

Our mapped region did not include the striatum or thalamus, and therefore these regions were not analyzed.

7. The biological insights from the analyses in the paper are lacking. Are the new specific developmentally regulated enhancers that were discovered using this method that control subcortical region (e.g. specific nuclei, subregions of the striatum and thalamus that are hard to dissect and resolve with single-cell sequencing) and cortical layer specification? How about the role of non-CG methylation compared to CG methylation in regulating gene expression in spatially restricted brain regions and laminae?

Response:

We thank the reviewer for pointing out that our initial submission did not fully realize the potential of this technological innovation to deliver biological insights. In the revised

manuscript, we added new spatial maps of specific anatomic locations and significantly expanded our data analysis focusing on enhancer regulatory mechanisms, their implication to subcortical and cortical layer specification, and the interplay between CG and non-CG methylation in these mechanisms. These analyses are detailed below:

Analysis of spatial-temporal dynamics and (sub)cortical layer specification. As the reviewer suggested, spatial technology enables analysis of regions that are challenging to dissect and resolve using single-cell sequencing. We conducted spatial mapping at cellular resolution (10- μ m pixel size) to analyze the facial and forebrain region of an E11 mouse embryo and observed refined spatial patterns (Fig. 1g and Extended Data Fig. 10a), which revealed distinct clusters corresponding precisely to known anatomical regions of the developing mouse brain areas that difficult to dissect. Notably, clusters W7 and W11 delineated key telencephalic compartments: W11 was enriched for telencephalon cells in the ventricular zone of the pallium, a neurogenic region characterized by active cell division and proliferation; W7 was enriched for GABAergic cortical interneurons localized in the mantle zone, where newborn neurons migrate, accumulate, differentiate, and begin to form the initial cortical architecture^{4,16,17} (Fig. 1h, Extended Data Fig. 10d). Gene ontology (GO) enrichment analyses further supported these regional identities, highlighting biological processes such as neurogenesis and progenitor proliferation in W11, and neuron projection and migration in W7 (Extended Data Fig. 10e). Moreover, the cell-type lineage tree constructed from the scRNA-seq reference confirmed the developmental trajectory, clearly showing telencephalon progenitors as precursors to GABAergic cortical interneurons¹⁸. In addition to these brain structures, our spatial analysis also resolved refined sensory structures in the developing olfactory system (W5 and W10) (Fig. 1g and Extended Data Fig. 10a). Sensory neurons were notably enriched in cluster W5 (Extended Data Fig. 10f and g), positioned closer and adjacent to the forebrain. This spatial pattern closely aligns with established developmental trajectories of the olfactory system, wherein olfactory sensory neurons progressively connect to the forebrain as embryogenesis advances⁵⁻⁷. Collectively, our results demonstrate the capacity of Spatial-DMT to resolve intricate anatomical details and spatiotemporal dynamics of DNA methylation and gene expression profiles during neural and sensory system development.

Fig. 1g-h:

Extended Data Fig. 10:

Analysis of regulatory mechanisms bearing spatial tissue context. Similarly, spatial analysis enabled the resolution of subtle region-specific epigenetic variations overlooked in single-cell sequencing. For instance, regional differential methylation analysis comparing

the ventricular and mantle zone of hindbrain/spinal cord revealed subtle methylation differences at neural progenitor cell-specific promoters and enhancers, as shown by the enrichment of corresponding H3K4me1 marks (Extended Data Fig. 16d). These differential methylated loci colocalized with enhancers involved in brain and neural tube development (mEnhA9), which are associated with key neuronal differentiation transcription factors such as FOXO4¹⁹, NEUROG2²⁰, and HOXC9²¹. Additionally, spatial methylome profiling distinguished epigenetic signatures within identical WNN clusters located at distinct spatial positions (Extended Data Fig. 16e). In the forebrain, hypomethylation is found at the binding sites of FOXI1, transcription factors crucial for hearing and visual development²², whereas hypomethylated loci specific to the spinal cord region is associated with the binding of TLX1, a transcription factor essential for spinal cord development and neuronal differentiation²³. These findings further highlight the power of spatial methylome profiling in resolving subtle epigenetic variations that are related to spatial context.

Extended Data Fig. 16d-e:

Analysis of DNA-RNA interaction in cell identity definition. Spatial-DMT offers a unique opportunity for us to look at the spatial distribution of DNA-RNA interaction. Interestingly, we observed clustering inconsistencies between 5mC and RNA in the E11 embryo, highlighting complementary information provided by DNA methylation and gene expression profiling. For instance, RNA cluster R3 subdivided into two distinct DNA clusters, D0 and D4. DNA

methylation analysis of these clusters revealed significant differences, notably enriched in key developmental transcription factors. Specifically, cluster D0 exhibited enrichment for Pitx1, a transcription factor associated with facial morphogenesis²⁴, while cluster D4 showed enrichment for transcription factors such as Hoxa1 and Gata6, known regulators crucial for cardiovascular development^{25,26}. In contrast, differential RNA expression analysis between these same clusters yielded fewer distinct markers, resulting in lower spatial resolution. These findings underscore the unique, orthogonal insights provided by Spatial-DMT-seq and demonstrate its enhanced capability in distinguishing subtle but biologically meaningful epigenetic states during development (Extended Data Fig. 16f, g).

Extended Data Fig. 16f, g:

Analysis of DNA methylation biology in spatial context. The unique DNA methylation mapping by Spatial-DMT allows us to study DNA methylation biology with more structure context. For example, we examined partially methylated domains (PMDs), lamina-associated and late-replicating genomic regions known to progressively lose DNA methylation through mitotic divisions, making them useful indicators of cellular proliferation. By spatially mapping PMD methylation in E11 and E13 mouse embryos (Extended Data Fig. 16a, b), we identified distinct tissue-specific patterns. For instance, regions such as the forebrain and hindbrain/spinal cord exhibited higher PMD methylation, while embryonic heart tissue demonstrated lower PMD methylation levels, aligning with its known active heart chamber formation at these developmental stages²⁷. Intriguingly, we observed spatial gradients in PMD methylation, decreasing from the center to the periphery in the heart, and from the mantle to the ventricular zone in the forebrain and hindbrain/spinal cord. These gradients potentially reflect the spatial organization of progenitor cells and their differentiation trajectories. In the postnatal P21 brain (Extended Data Fig. 16c), cortical layers displayed higher PMD methylation, consistent with reduced proliferative capacity typical of differentiated neurons²⁸. In contrast, the dentate gyrus exhibits comparatively

lower PMD methylation, consistent with the presence of neural stem/progenitor cells in the subgranular zone that continues to undergo mitotic division and neurogenesis²⁹.

Extended Data Fig. 16a-c:

Comparison of CG vs non-CG regulation: Lastly, following reviewer’s suggestion, we systematically assessed the correlations between CpG and non-CpG (predominantly CpA) methylation patterns and gene expression across spatially defined brain regions (Fig. 4c-h). Notably, genes such as *Cux1* showed stronger correlations between CpA methylation and expression compared to CpG methylation in specific regions, suggesting a unique spatially restricted regulatory role for non-CpG methylation. While these observations provide new spatial insights, we acknowledge that fully delineating the distinct regulatory roles of these methylation marks requires further experimental validation beyond the scope of this study. Future functional perturbation studies could further clarify these mechanisms.

Fig. 4c-h:

These above analyses have been integrated into the revised Results and Discussion section, demonstrating additional biological insights yielded by Spatial-DMT.

Referee #2 (Remarks to the Author):

Summary of the Paper:

The paper introduces a new spatial multi-omics technology capable of joint profiling of the DNA methylome and transcriptome at near-single-cell resolution. This technology was applied to E11, E13 mouse embryos, and P21 mouse brains. The key novelty lies in its ability to simultaneously capture both RNA expression and DNA methylation, offering insights into the complex interplay between these two modalities. However, the lack of new biological insights resulted from such co-profiling and the lack of clear benchmarking against previously published technologies are major limitations of the paper, and there are a few significant concerns to be addressed:

Response:

We would like to thank the reviewer for the positive feedback and appreciate their concerns regarding biological insights and benchmarking. We take these comments seriously and have significantly expanded our data and analysis. We comprehensively benchmarked our Spatial-DMT technology against previously published technologies. Additionally, we further explored and expanded upon the biological insights gained from spatial co-profiling of DNA methylation and RNA expression. Detailed responses to specific points are provided below.

Main Comments and Suggestions:

1. Technical Workflow and Methodological Considerations:

(1) Use of HCl for Tn5 Tagmentation:

The authors used HCl to increase genomic DNA accessibility, followed by extensive Tn5 tagmentation prior to mRNA capture. However, this procedure may compromise RNA quality as HCl is known to reverse cross-linking in PFA-fixed samples. This concern is supported by the lower number of UMIs per pixel observed in their RNA data compared to the original DBiT-seq paper (Liu et al., 2020).

Suggestions:

Discuss whether the joint profiling of two modalities compromises the data quality of individual modalities by comparing the RNA data in spatial-DMT with DBiT-seq. Elaborate on how much of the genome becomes accessible post-treatment. Are there specific genes or regions that remain inaccessible? Provide data to address these questions. Evaluate if the treatment affects the transcriptome or DNA methylation, and if so, to what extent.

Consider adding experiments or data to clarify these concerns.

Response:

Thank you for your comments. After careful evaluation, we found that the HCl treatment does not compromise RNA integrity or data quality.

First, it is important to note that the 50 μm resolution DBiT-seq dataset was generated from an E10 mouse embryo, while our study profiled E11 and E13 embryos. Developmental stage differences may contribute to variations in gene detection. For a direct comparison, we compared Spatial-DMT and DBiT-seq data at the same spatial resolution (10 μm) and similar sample types (E13 mouse embryo). These comparisons demonstrated comparable data quality (Fig. 1e). Furthermore, our RNA data show comparable UMI counts and gene detection per pixel compared to previously published spatial ATAC and transcriptomic co-profiling results (E13 mouse embryo and P21 mouse brain) (Di et al., *Nature*, 2023)⁹, which did not employ HCl treatment (Fig. 1e).

Fig. 1e:

Regarding genome accessibility post-treatment, we assessed the genomic distribution of CpG coverage and found it to be uniformly distributed across genomic regions (Extended Data Fig. 5a). Methylation levels across various chromatin states are consistent with known biology and are comparable with those published databases^{12,13,30,31} (Extended Data Fig. 5a, b). These findings indicate that the HCl treatment does not introduce biases or selectively restrict access to specific genes or genomic regions.

Extended Data Fig. 5a-b:

(2) Choice of 50 μm Microfluidic Device:

The use of a 50 μm microfluidic device for E11 and E13 mouse embryos results in each pixel covering multiple cells, thus not achieving single-cell resolution.

Suggestions:

Discuss why a smaller, 10 μm microfluidic device was not used to achieve closer-to-single-cell resolution. Address how averaging between neighboring cells may affect data interpretation and provide insights into potential biases introduced by this approach.

Response:

Thank you for your comments regarding the resolution of our method. In the first submission, we employed a spatial resolution range of 20–50 μm for Spatial-DMT primarily to demonstrate the feasibility of simultaneously mapping DNA methylation and RNA transcriptome profiles within a tissue context. Although this resolution is not at the single-cell level, it has already provided significant biological insights in both embryonic and postnatal brain tissues. The choice of resolution depends on the specific biological questions being investigated. For instance, broader coverage at 50 μm resolution is beneficial for studying extensive regions such as a whole embryo, whereas higher resolutions (e.g., 20 μm or 10 μm) enable more detailed analysis of localized regions. Additionally, the number of cells captured per pixel varies based on several tissue-specific

characteristics, including tissue species (e.g., human versus mouse tissues), cell density (e.g., densely populated hippocampal regions versus less dense cortical regions), cell size (e.g., smaller lymphocytes compared to larger neurons), and distribution and organization of the region of interest within the tissue.

To further demonstrate the feasibility of achieving near single-cell resolution, we performed experiments at a 10 μm resolution specifically mapping the craniofacial and forebrain regions of an E11 embryo (Fig. 1g-h, Extended Data Fig. 10). This higher resolution allowed us to resolve finer cellular subtypes that were challenging to distinguish using a 50 μm chip. A dedicated results section discussing these new findings from the 10 μm analysis has now been included, underscoring that our Spatial-DMT technology is adaptable and capable of achieving higher resolution as needed.

“To precisely resolve the fine structure of transcriptional regulation at the craniofacial and forebrain regions of the E11 embryo, we employed the 10 μm pixel-size microfluidic chip to produce a near single-cell resolution spatial map (Fig. 1g, h and Extended Data Fig. 10a). Integration of our spatial dataset with a single-cell RNA sequencing (scRNA-seq) reference¹ from mouse embryo revealed a strong concordance between the two datasets (Extended Data Fig. 10b, c). Cell-type deconvolution using the scRNA-seq reference¹ identified distinct clusters that correspond precisely to known anatomical structures of the developing mouse brain. Notably, two spatially defined clusters, W7 and W11, captured key telencephalic compartments. W11 was enriched for telencephalon progenitors in the ventricular zone of the pallium, a neurogenic niche characterized by active cell division and proliferation, while W7 corresponded to GABAergic cortical interneurons localized in the mantle zone, where newborn neurons migrate, accumulate, and differentiate to establish cortical architecture²⁻⁴ (Fig. 1h and Extended Data Fig. 10d). Gene ontology (GO) enrichment analysis further supported these regional identities, highlighting biological processes associated with neurogenesis and progenitor proliferation in W11, and neuron projection and migration in W7 (Extended Data Fig. 10e). Moreover, the cell-type lineage tree constructed from the scRNA-seq reference confirmed the developmental trajectory, positioning telencephalon progenitors as direct precursors to GABAergic cortical interneurons¹. Beyond the forebrain, our spatial analysis also resolved refined sensory structures within the developing olfactory system (cluster W5 and W10) (Fig. 1g and Extended Data Fig. 10a). Sensory neurons were notably enriched in W5 (Extended Data Fig. 10f, g), spatially localized adjacent to the forebrain. This spatial pattern closely aligns with the established developmental trajectory of the olfactory system, wherein olfactory sensory neurons progressively form connections with the forebrain as embryogenesis progresses⁵⁻⁷. Together, these findings provide a high-resolution view of

neural and sensory system formation, underscoring the power of Spatial-DMT in resolving intricate anatomical structures and capturing the spatiotemporal dynamics.”

Fig. 1g-h:

Extended Data Fig. 10:

We acknowledge that multiple cells captured within a single pixel result in averaging signals from neighboring cells, a known limitation of current sequencing-based spatial technologies. This averaging effect can obscure cellular heterogeneity, particularly at the boundaries between distinct spatial domains or in regions containing diverse cell populations. This effect might also limit the resolution of cell-cell interactions. However, in regions predominantly composed of similar cell types, such averaged signals still provide biologically informative insights. We have added a discussion of these limitations in the revised manuscript and proposed that future developments, including advanced spatial computational deconvolution methods, could help infer single-cell resolution data from averaged signals, thus mitigating potential biases and enhancing biological interpretability.

“Lower-resolution pixels (e.g., 50 μm) generally capture more CpGs, UMIs, and expressed genes, likely due to the inclusion of a greater number of cells within each pixel. This effect depends on factors such as tissue heterogeneity and cell type variability, and results in each pixel representing a mixture of signals from neighboring cells. Further development of advanced spatial computational deconvolution methods is essential for inferring single-cell resolution, minimizing potential biases and enhancing biological interpretability.”

2. Analytical Limitations and Data Interpretation:

(1) Simultaneous Profiling of 5hmC and 5mC:

While the authors claim that their method profiles 5hmC and 5mC simultaneously using TET oxidation followed by APOBEC deamination, the chemistry used cannot distinguish between endogenous 5fC, 5caC, 5hmC, and 5mC.

Suggestions:

Include a discussion on these limitations and the potential impact on data interpretation.

Response:

We regret that our initial submission contains ambiguous language. As the reviewer pointed out, our current approach, utilizing TET oxidation followed by APOBEC deamination, does not differentiate between endogenous 5fC, 5caC, 5hmC, and 5mC. To avoid ambiguity, we now refer to our assay target as “the sum of 5mC and 5hmC” or “total cytosine modifications” instead of “5mC and 5hmC”, which may have unintentionally suggested that our method co-assays the two modifications separately.

Specialized approaches, such as oxidative bisulfite sequencing, can resolve these modifications. Recently developed single-cell methods, such as bACE-seq³², which enzymatically protects 5hmC via glycosylation from APOBEC3A, and SIMPLE-seq³³, which utilizes glucosylated 5hmC (5ghmC)-dependent restriction endonuclease AbaSI, have enabled the discrimination between 5hmC and 5mC. These techniques could potentially be integrated into our platform in the future but is beyond this manuscript's scope. We have now included a discussion of this limitation and its potential impact on data interpretation in our manuscript and highlighted potential future directions to address this limitation.

“Spatial-DMT, which is based on an Enzymatic Methyl-seq, does not distinguish between 5-methylcytosine and 5-hydroxymethylcytosine modification, the latter of which accumulates in certain brain regions¹³. This lack of resolution between distinct cytosine modifications complicates the interpretation of their respective roles in regulating gene expression. This limitation could be overcome by employing emerging methods that enable simultaneous measurement of the full spectrum of cytosine base modifications^{33,34}.”

(2) Correlation Between DNA Methylation and Gene Expression:

Different genes exhibit varying correlations with DNA methylation. The lack of differentiation between various methylation states (5mC and 5mC oxidation) could lead to misinterpretation.

Suggestions:

Conduct a more detailed analysis to distinguish the effects of different methylation states on gene expression.

Explore whether positive correlations observed are due to the presence of 5hmC, 5fC, or 5caC, which are generally associated with DNA de-silencing.

Response:

Thank you for your comments. We agree that not discriminating 5mC from its oxidative derivatives obfuscates the interpretation of their impact on gene expression regulation, particularly given that 5hmC has its separate reader proteins from 5mCs³⁵, and 5hmC is known to localize differently from 5mC in the genome. Indeed, our recent paper suggests that 5hmC at Polycomb targets and enhancers do contribute to the positive association between total modification and gene expression³⁶. It can be speculated that 5mC and 5hmC may each have unique regulatory roles influencing gene expression. Unfortunately, as discussed in our last response above, our current method relies on TET oxidation followed

by APOBEC deamination, which does not discriminate between these specific modifications. Therefore, we are not able to provide concrete data to resolve the contribution of different cytosine modification forms on gene expression. Incorporating more refined chemical or enzymatic methods, such as oxidative bisulfite sequencing or ACE-seq, into our spatial platform is planned for future implementations but represents a significant technical advancement that will require further development. We have included a discussion of this limitation and its potential impact on data interpretation in our manuscript and highlighted potential future directions to overcome this challenge.

“Spatial-DMT, which is based on an Enzymatic Methyl-seq, does not distinguish between 5-methylcytosine and 5-hydroxymethylcytosine modification, the latter of which accumulates in certain brain regions¹³. This lack of resolution between distinct cytosine modifications complicates the interpretation of their respective roles in regulating gene expression. This limitation could be overcome by employing emerging methods that enable simultaneous measurement of the full spectrum of cytosine base modifications^{33,34}.”

3. Novelty and Contribution:

(1) Importance and Benefits of Spatial Multi-Omics Technology:

While the paper emphasizes the ability to co-profile DNA methylation and transcriptome, the authors should better highlight the novelty and significance of their work compared to previous studies.

Suggestions:

Provide a clearer comparison with the previous spatial epigenome-transcriptome co-profiling study, particularly focusing on the new contributions made by the ability to map DNA methylation spatially.

Emphasize the potential benefits of this technology over traditional single-cell profiling strategies, such as identifying novel spatial patterns and region-specific regulatory mechanisms.

Response:

Thank you for suggesting a comparison of Spatial-DMT with other epigenome-transcriptome co-profiling methods. As mentioned by the reviewer, our approach uniquely profiles DNA methylation, enabling direct mapping of epigenetic states to their original tissue locations, which is a significant advantage over single-cell technologies reliant on computational inference. Incorporating gene expression co-profiling further enabled direct assessment of

correlations between gene expression levels and DNA methylation changes across distinct spatial regions.

In our study, we systematically assessed the correlations between CpG and non-CpG (predominantly CpA) methylation patterns and gene expression across spatially defined brain regions (Fig. 4c-h). Notably, genes such as *Cux1* showed stronger correlations between CpA methylation and expression compared to CpG methylation in specific regions, suggesting a unique spatially restricted regulatory role for non-CpG methylation. While these observations provide new spatial insights, we acknowledge that fully delineating the distinct regulatory roles of these methylation marks requires further experimental validation beyond the scope of this study. Future functional perturbation studies could further clarify these mechanisms.

Fig. 4c-h:

Furthermore, as the reviewer suggested, spatial technology enables analysis of regions that are challenging to dissect and resolve using single-cell sequencing. To address this comment, we conducted spatial mapping at cellular resolution (10- μm pixel size) to analyze the facial and forebrain region of an E11 mouse embryo and observed refined spatial patterns (Fig. 1g and Extended Data Fig. 10a), which revealed distinct clusters corresponding precisely to known anatomical regions of the developing mouse brain areas difficult to dissect. Notably, clusters W7 and W11 delineated key telencephalic compartments: W11 was enriched for telencephalon cells in the ventricular zone of the pallium, a neurogenic region characterized by active cell division and proliferation; W7 was enriched for GABAergic cortical interneurons localized in the mantle zone, where newborn neurons migrate, accumulate, differentiate, and begin to form the initial cortical architecture^{4,16,17} (Fig. 1h, Extended Data Fig. 10d). Gene ontology (GO) enrichment analyses further supported these regional identities, highlighting biological processes such as neurogenesis and progenitor proliferation in W11, and neuron projection and migration in W7 (Extended Data Fig. 10e). Moreover, the cell-type lineage tree constructed from the scRNA-seq reference confirmed the developmental trajectory, clearly showing telencephalon progenitors as precursors to GABAergic cortical interneurons¹⁸. In addition to these brain structures, our spatial analysis also resolved refined sensory structures in the developing olfactory system (W5 and W10) (Fig. 1g and Extended Data Fig. 10a). Sensory neurons were notably enriched in cluster W5 (Extended Data Fig. 10f, g), positioned closer and adjacent to the forebrain. This spatial pattern closely aligns with established developmental trajectories of the olfactory system, wherein olfactory sensory neurons progressively connect to the forebrain as embryogenesis advances⁵⁻⁷. Collectively, our results demonstrate the capacity of Spatial-DMT to resolve intricate anatomical details and spatiotemporal dynamics of DNA methylation and gene expression profiles during neural and sensory system development.

Fig. 1g-h:

Extended Data Fig. 10:

In addition, spatial analysis enabled the resolution of subtle region-specific epigenetic variations overlooked in single-cell sequencing. For instance, regional differential methylation analysis comparing the ventricular and mantle zone of hindbrain/spinal cord revealed subtle methylation differences at neural progenitor cell specific promoter and enhancers, as shown by the enrichment of corresponding H3K4me1 marks (Extended Data Fig. 16d). These differential methylated loci colocalized with enhancers involved in brain and neural tube development (mEnhA9), which are associated with key neuronal differentiation transcription factors such as FOXO4¹⁹, NEUROG2²⁰, and HOXC9²¹. Additionally, spatial methylome profiling distinguished epigenetic signatures within identical WNN clusters located at distinct spatial positions (Extended Data Fig. 16e). In the forebrain, hypomethylation is found at the binding sites of FOXI1, transcription factors crucial for hearing and visual development²², whereas hypomethylated loci specific to the spinal cord region is associated with the binding of TLX1, a transcription factor essential for spinal cord development and neuronal differentiation²³. These findings further highlight the power of

spatial methylome profiling in resolving subtle epigenetic variation that is related to spatial context.

Extended Data Fig. 16d-e:

To further explore the biological insights from spatial DNA methylation profiling:

(1) We observed clustering inconsistencies between 5mC and RNA in the E11 embryo, highlighting complementary information provided by DNA methylation and gene expression profiling. For instance, RNA cluster R3 subdivided into two distinct DNA clusters, D0 and D4. DNA methylation analysis of these clusters revealed significant differences, notably enriched in key developmental transcription factors. Specifically, cluster D0 exhibited enrichment of *Pitx1*, a transcription factor associated with facial morphogenesis²⁴, while cluster D4 showed enrichment for transcription factors such as *Hoxa1* and *Gata6*, known regulators crucial for cardiovascular development^{25,26}. In contrast, differential RNA expression analysis between these same clusters yielded fewer distinct markers, resulting in lower spatial resolution. These findings underscore the unique, orthogonal insights provided by Spatial-DMT-seq and demonstrate its enhanced capability in distinguishing subtle but biologically meaningful epigenetic states during development (Extended Data Fig. 16f, g).

Extended Data Fig. 16f, g:

f

g

(2) We examined partially methylated domains (PMDs), lamina-associated and late-replicating genomic regions known to progressively lose DNA methylation through mitotic divisions, making them useful indicators of cellular proliferation. By spatially mapping PMD methylation in E11 and E13 mouse embryos (Extended Data Fig. 16a, b), we identified distinct tissue-specific patterns. For instance, regions such as the forebrain and hindbrain/spinal cord exhibited higher PMD methylation, while embryonic heart tissue demonstrated lower PMD methylation levels, aligning with its known active heart chamber formation at these developmental stages²⁷. Intriguingly, we observed spatial gradients in PMD methylation, decreasing from the center to the periphery in the heart, and from the mantle to ventricular zone in the forebrain and hindbrain/spinal cord. These gradients potentially reflect the spatial organization of progenitor cells and their differentiation trajectories. In the postnatal P21 brain (Extended Data Fig. 16c), cortical layers displayed higher PMD methylation, consistent with reduced proliferative capacity typical of differentiated neurons³⁷. In contrast, the dentate gyrus exhibits comparatively lower PMD methylation, consistent with the presence of neural stem/progenitor cells in the subgranular zone that continues to undergo mitotic division and neurogenesis²⁹.

Extended Data Fig. 16a-c:

These findings have been integrated into the revised Results and Discussion section demonstrating that Spatial-DMT is a powerful and biologically informative tool for uncovering novel regulatory mechanisms within their tissue context.

(2) Comparisons with Public Datasets:

Given the novel nature of this technology, validating these findings against publicly available datasets would add credibility.

Suggestions:

Integrate publicly available datasets to validate or benchmark the observed patterns, especially for differentially regulated genes in the brain.

Response:

Thank you for your comments regarding the validation against publicly available datasets. To address your suggestions, we performed additional integrative analyses:

- Co-clustering with published spatial data: We integrated our E13 embryo dataset (50 μm resolution) with a comparable Spatial-ATAC-RNA sequencing dataset (Di et al., Nature, 2023)⁹. The integrated data (Extended Data Fig. 12) demonstrates well-resolved spatial domains and clear, coherent cell-type clustering across both datasets.
- Integration with published single-cell RNA-seq data: We further integrated our P21 brain sample (20 μm resolution) with a publicly available mouse brain single-cell RNA-seq dataset³⁸, which resulted in consistent and refined clustering (Extended Data Fig. 15). We then examined the correlation between gene expression in our data and the single-cell RNA dataset, observing a high correlation within clusters corresponding to specific tissues (Extended Data Fig. 15d, e). Additionally, we conducted a new 10 μm resolution experiment from the E11 mouse embryo

craniofacial and forebrain region, which showed strong concordance with a published single-cell RNA-seq dataset (Extended Data Fig. 10b, c).

- Examine differentially regulated genes in the brain. We have analyzed key markers for major cell types, including neurons (Syt1, Rbfox3), oligodendrocytes (Mbp, Plp1), fibrous astrocytes (Gfap), upper-layer neurons (Cux2, Cux1, Satb2), and deep-layer neurons (Bcl11b). For each marker, we assessed both RNA expression and promoter DNA methylation levels. The results, now included in Extended Data Fig. 14, reveal spatially restricted expression patterns consistent with known brain organization.

These comprehensive analyses further confirm the quality, robustness, and accuracy of our Spatial-DMT approach.

Extended Data Fig. 12:

Extended Data Fig. 15:

Extended Data Fig. 10b, c:

Extended Data Fig. 14:

4. Data Presentation and Clarity:

Workflow and Fig.s:

The use of multiple Fig.s, such as Fig. 1B, lacks clarity and requires revision for better interpretation.

Suggestions:

Revise Fig. 1B for better clarity. Include a more detailed explanation of the “retention rate” in Fig. 1.

Address why the number of CpGs detected is significantly lower compared to other studies (e.g., Hernando 2019 and Liu 2021).

Provide a clear definition for terms like “variably methylated regions” and “retention rate” to enhance readability.

Response:

Thank you for your comments. In response:

- To improve readability, we moved the detailed experimental pipeline to Extended Data Fig. 1 and introduced a simplified workflow in Fig. 1B. This update allows readers to follow the key protocol steps without being overwhelmed by technical details.
- For the terminology, we have now explicitly defined “retention rate” in the manuscript. It refers to the percentage of cytosines that remain unconverted, serving as a measure of methylation stability. This metric has been used in previous studies^{12,39}.
- Additionally, variably methylated regions (VMRs) are defined as genomic regions that exhibit high variance in methylation across cells or spatial pixels (based on the definition in a recent Nature Methods paper⁴⁰). We have explicitly defined it in the manuscript to improve clarity.
- Regarding the number of CpGs detected, our coverage is lower than that reported by Liu et al. (2021)¹³ and Hernando et al. (2019)³¹, primarily due differences in sequencing depth. For instance, Liu et al. (2021) sequenced over 1.5 million reads per cell¹³, whereas our average reads per pixel are approximately one-third of that. However, it is important to note that our method is a spatial technology, which inherently differs from single-cell methods. The protocols used in those single-cell studies are not compatible with spatial profiling. Despite this, our CpG coverage exceeds that of some existing single-cell technologies (Fig. 1C), while also offering the unique advantage of simultaneous, spatially resolved gene expression profiling.

Fig. 1B:

Fig. 1C:

5. Spatial and Temporal Resolution:

(1) Spatial Resolution and Single-Cell Analysis:

The resolution of 20-50 μm does not fully achieve single-cell resolution, which needs to be clearly addressed.

Suggestions:

Discuss the potential to achieve improved spatial resolution and the limitations imposed by the current setup.

Provide details on the number of tissue slices and replicates analyzed for reproducibility. Address if the findings are based on a single sample or multiple biological replicates.

Response:

Thank you for your comments regarding the resolution of our method. Initially, we employed a spatial resolution range of 20–50 μm for Spatial-DMT primarily to demonstrate the feasibility of simultaneously mapping DNA methylation and RNA transcriptome profiles within a tissue context. Although this resolution is not at single-cell level, it has already provided significant biological insights in both embryonic and postnatal brain tissues. The choice of resolution depends on the specific biological questions being investigated. For instance, broader coverage at 50 μm resolution is beneficial for studying extensive regions such as a whole embryo, whereas higher resolutions (e.g., 20 μm or 10 μm) enable more detailed analysis of localized regions. Additionally, the number of cells captured per pixel varies based on several tissue-specific characteristics, including tissue species (e.g.,

human versus mouse tissues), cell density (e.g., densely populated hippocampal regions versus less dense cortical regions), cell size (e.g., smaller lymphocytes compared to larger neurons), and distribution and organization of the region of interest within the tissue.

To further demonstrate the feasibility of achieving near single-cell resolution, we performed experiments at a 10 μm resolution specifically mapping the craniofacial and forebrain regions of an E11 embryo (Fig. 1g-h, Extended Data Fig. 10). This higher resolution allowed us to resolve finer cellular subtypes that were challenging to distinguish using a 50 μm chip. A dedicated results section discussing these new findings from the 10 μm analysis has now been included, underscoring that our Spatial-DMT technology is adaptable and capable of achieving higher resolution as needed.

Fig. 1g, h:

Extended Data Fig. 10:

To evaluate the reproducibility of our findings, we performed an independent replication experiment using a separate E11 mouse embryo sample. We observed strong correlations ($R > 0.95$) in both DNA methylation and gene expression levels between the original and replicate datasets (Extended Data Fig. 3a), confirming the consistency and robustness of our Spatial-DMT approach. UMAP visualizations of spatial clusters across replicates reveal highly similar clustering patterns, which were further supported by comparable spatial cluster assignments observed in the spatial maps (Extended Data Fig. 3b and c). Furthermore, we re-examined several previously analyzed marker genes and observed consistent spatial expression patterns across replicates (Extended Data Fig. 3d-f). These results further support the reproducibility of our method. Details of all tissue samples and replicates used in this study are provided in Supplementary Table 5.

Extended Data Fig. 3:

(2) Pseudo-time Analysis:

The terms “temporal” and “dynamics” are used frequently, despite the absence of time-resolved data.

Suggestions:

Use more appropriate terminology that reflects the pseudo-temporal nature of the data.

Response:

Thank you for your comments regarding our use of the term “temporal” and “dynamics” in the manuscript. We agree that more precise terminology should be used when referring to pseudo-time analysis, in order to avoid potential misinterpretation. To address this concern, we have updated our terminology, explicitly using “pseudo-temporal” and “pseudo-dynamics” when describing results derived from pseudo-time analysis. However, we have retained the original terms “temporal” and “dynamics” in contexts describing biological differences between actual embryonic stages (E11 to E13), as these reflect true developmental progression.

6. Integration of Multi-Modal Data:

Joint Analysis of DNA Methylation and Transcriptome Data:

The authors showed that different genes in the mouse brain are regulated by mCG, mCA, or both. However, they do not provide a thorough explanation for why this regulation differs.

Suggestions:

Compare or integrate these findings with other datasets to validate the distinct roles of mCA and mCG in epigenetic regulation.

Conduct a more detailed analysis on why specific genes are differentially regulated by mCG or mCA.

Response:

Thank you for your comments. While our current study focuses on correlative analyses between gene expression and variably methylated regions (VMRs), we offer the following insights into how specific genes may be differentially influenced by these methylation marks, which is potentially due to the distinct genomic distribution of these VMRs:

- Prox1 expression in the dentate gyrus correlates with both mCG and mCA VMRs (Fig. 4c, f). These overlapping VMRs are located within active enhancer regions (mEnhA), suggesting that both methylation marks may contribute to fine-tuning Prox1 expression in this context.

CA/CG VMR location from UCSC genome browser with chromHMM annotation

- *Ntrk3* shows a primary correlation with mCG VMRs (Fig. 4d, g), which largely overlap with weak enhancers (mEnHwK) located in exonic regions. This indicates that mCG may play a more dominant role in regulating *Ntrk3* expression through these enhancer contexts.

CA/CG VMR location from UCSC genome browser with chromHMM annotation

- Cux1 is more strongly correlated with mCA VMRs (Fig. 4e, h), likely due to their overlap with more transcribed enhancers (mTxEnh).

CA/CG VMR location from UCSC genome browser with chromHMM annotation

We acknowledge that our current approach does not establish causality. Further functional perturbation studies will be necessary to definitively parse out the distinct roles of mCG and mCA and to elucidate the mechanisms by which these methylation VMR marks differentially regulate gene expression, which are beyond the scope of this work.

7. Specific Figure and Data Points:

Fig. 1F:

The authors overlay their spatial omics data with a histological image of a mouse embryo, but it's unclear how well the results align with known anatomy.

Suggestions:

Quantify the alignment of spatial data with known anatomical features.

Response:

Thank you for your comments. To rigorously assess the alignment between Spatial-DMT data with known anatomical features, we individually overlay each cluster from our molecular

data with the bright-field images and evaluate whether the pixels from these clusters fall exclusively to known anatomic structures. In the figures below, we present the cases of the E11 embryonic heart and different regions of P21 mouse brain. In both cases, the pixels fill the corresponding anatomic regions from the bright-field images, marking heart, CA1, CA2, CA3, and dentate gyrus (DG) regions and their respective anatomical subdivisions precisely.

Furthermore, as suggested by Reviewer 3, we have updated several figures (e.g., Fig. 1f-h) to include the original histological image alongside the spatial cluster maps. This side-by-side presentation enables a more intuitive and direct visual comparison, highlighting the anatomical relevance of the spatial clustering results.

Fig. 1f-h:

Fig. 4B, 4F, 4G, and 4H:

These figures display RNA and DNA methylation data integration, yet some expression patterns appear random.

Suggestions:

Provide further explanation or possible hypotheses for these seemingly random patterns. Include benchmarking with known datasets to confirm the biological relevance of these clusters.

Response:

Thank you for this observation. Although the brain can be divided into distinct areas, each area comprises a diverse mix of cell types that together form a local microenvironment. For example, while neurons are more abundant in the cortex compared to other brain regions, glial cells are also present to support neuronal functions. Therefore, due to local cellular heterogeneity, some degree of randomness in gene expression and DNA methylation in the specific brain region is expected.

Despite some randomness, the spatial distribution still follows a pattern governed by the known differences in cell composition between brain regions. For example, canonical markers for neurons (*Syt1*, *Rbfox3*), oligodendrocytes (*Mbp*, *Plp1*), astrocytes (*Gfap*), and upper- and deep-layer neurons (*Cux2*, *Cux1*, *Satb2*, *Bcl11b*) are clearly differentially expressed and methylated across brain regions. These genes exhibit distinct and spatially localized expression patterns consistent with their expected anatomical distributions in the postnatal brain (Extended Data Fig. 14). Additionally, our spatial clustering results show strong concordance with established hippocampal subregions, including CA1, CA2, CA3, and the dentate gyrus (DG) (Fig. 4b). This alignment supports the biological validity of our spatial data.

Following the reviewer's suggestion, we further integrated our RNA expression data with a publicly available mouse brain single-cell RNA-seq dataset³⁸. The resulting analysis demonstrates a strong concordance between our spatial clusters and annotated cell types in the reference dataset (Extended Data Fig. 15), validating the biological relevance of clusters defined in our dataset.

Extended Data Fig. 14:

Fig. 4a-b:

Extended Data Fig. 15:

8. Additional Considerations:

Replicability and Statistical Analysis:

The lack of replicates or clear statistical validation is a significant gap.

Suggestions:

Include more biological replicates and detailed statistical analysis for all key findings to enhance reproducibility.

Response:

Thank you for your comments. To evaluate the reproducibility of our findings, we performed an independent replication experiment using a separate E11 mouse embryo sample. We observed strong correlations ($R > 0.95$) in both DNA methylation and gene expression levels between the original and replicate datasets (Extended Data Fig. 3a), confirming the consistency and reproducibility of our Spatial-DMT approach. UMAP visualizations of spatial clusters across replicates reveal highly similar clustering patterns, which were further supported by comparable spatial cluster assignments observed in the spatial maps (Extended Data Fig. 3b, c). Furthermore, we re-examined several previously analyzed marker genes and observed consistent spatial expression patterns across replicates (Extended Data Fig. 3d-f). The concordance between replicates, as well as the differences in gene expression and methylation levels across developmental stages, are statistically significant (Fig. 3e, f, Wilcoxon tests).

Extended Data Fig. 3:

Fig. 3e, f:

Referee #3 (Remarks to the Author):

In this manuscript, Lee & Fu et al., present spatial-DMT for spatial co-profiling of DNA methylation and transcriptome. The authors spent efforts to combine existing approaches for 5mC and transcriptomics measurement to enable joint spatial profiling of 5mC and RNA, which is certainly of high interest. The authors demonstrated the utilities of this technology to mouse embryogenesis and postnatal brain and revealed spatial context of known methylation and expression patterns. While the authors showed this new technology is able to identify the concordant and distinct DNA methylation and gene expression relationships in different spatial contents, the current manuscript lacks some necessary information from both technical and analytical aspects. Specific comments are listed below.

Response:

We would like to thank the reviewer for the positive feedback and insightful suggestions. To address the reviewer's concerns, we have incorporated additional evaluations and benchmarking analyses, including direct comparisons with existing spatial and single-cell datasets, improved technical clarifications, and performed additional analyses to strengthen the robustness of our findings further. Detailed responses to each specific point are provided below.

As a manuscript reporting a new technology, systematically (quantitative) evaluation and benchmarking is required. Some concerns about the specificity of this method are from lack of control or technical details. For example.

1. One key information for DNA methylation analysis – conversion rates – is missing throughout the manuscript. The accurate calculation of conversion rates can be performed on synthetic spike-in DNA with know 5mC sites. In addition, in Line 130 and Fig. 1d, it is unclear what is “retention rate”. Is this meant to be “modification level”?

Response:

Thank you for the comments. To address the concern regarding conversion rate, we have included an analysis of the linker sequence present in the raw reads, in which all cytosines are expected to convert due to the absence of methylation. Our analyses showed that all samples exhibit a high conversion rate exceeding 99% (Extended Data Fig. 4e). The term "retention rate" refers to the percentage of cytosines that remain unconverted. This metric reflects the outcome from both biological modification levels and unconverted cytosines resulting from technical incomplete conversion in the assay. For this reason, we shy away from using “modification level,” which specifically refers to biological methylation. Terminology, retention rate, has been widely used in previous methylation assay

development studies^{12,39}. For clarity, we have now explicitly defined “retention rate” in the revised manuscript as below.

“The retention rate, defined as the percentage of unconverted cytosines due to methylation or incomplete conversion.”

Extended Data Fig. 4e:

2. Did the authors performed technical replicates to demonstrate reproducibility?

Response:

Thanks for the comments. To evaluate the reproducibility, we included an analysis of an independent replication experiment using a separate E11 mouse embryo sample. Our analysis suggests that the two replicate datasets exhibit strong correlations ($R > 0.95$) in both DNA methylation and gene expression levels (Extended Data Fig. 3a). UMAP visualizations of spatial clusters across replicates reveal highly similar clustering patterns, which were further supported by comparable spatial cluster assignments observed in the spatial maps (Extended Data Fig. 3b, c). In addition, we re-examined several established marker genes and observed consistent spatial expression patterns across replicates (Extended Data Fig. 3d-f). Collectively, these results demonstrate the high reproducibility of our Spatial-DMT technology.

Extended Data Fig. 3:

3. Another missing piece is the comparison with public/reference datasets. For example, how many identified 5mC sites are also supported by previous methods? What is the genome-wide correlation for 5mC modification levels? This is particularly information is no positive control (spike-in DNA) can support the successful conversion of 5mC bases. The comparison of gene expression levels detected by this method with prior methods is also lacking.

Response:

We agree that benchmarking against publicly available reference datasets is essential and regret that our initial submission did not address this thoroughly. As was mentioned in our response to a previous comment, we have included an analysis of unmodified linker DNA, which is expected to be fully converted and can serve the utility of spike-in control (e.g.,

lambda DNA). This analysis indicated >99% conversion of unmodified cytosines (Extended Data Fig. 4e). Following the reviewer's suggestion, the revised manuscript also conducted the following comparison with public data sets to validate our methods and data further:

Comparison with public DNA methylome data: As specifically suggested by the reviewer, we compared our DNA methylation data and publicly available bulk whole-genome bisulfite sequencing (WGBS) datasets from E11 and E13 mouse embryos⁴¹ (results are shown below). We observed a high overlap of methylated CpG sites (methylation levels >0.5 in pseudo-bulk) and a strong genome-wide correlation in 5mC levels. It is worth noting that Extended Data Fig. 5b showed that our data is highly congruent with four public single-cell methylome datasets in the modification measurements stratified by genomic territories, displaying low modifications at transcription start sites and high modifications at gene bodies. These analyses provide evidence to support the accuracy and biological relevance of our DNA methylation profiling.

Extended Data Fig. 5b

Comparison with public transcriptome data: To evaluate the transcriptomic modality, we compared our spatial RNA expression data from the CA1 (W5) and dentate gyrus (W3) regions of the P21 mouse brain with publicly available single-cell RNA-seq data from the

same anatomical regions³⁸ (Extended Data Fig. 16d, e). We observed a high concordance in gene expression profiles, further supporting the quality and biological relevance of our spatial transcriptomic data.

Together, these benchmarking analyses confirm the reliability and biological validity of both the DNA methylation and gene expression modalities profiled using Spatial-DMT. These results and discussions have been included in the revised manuscript.

Extended Data Fig. 16d, e:

4. The authors used HCl to disrupt nucleosome to minimize biased tagmentation, which is supported by Fig. S2e. However, there is still non-negligible enrichments toward heterochromatin and depletion from enhancers. What background is used for calculating odds ratio, reference genome annotation or real data from WGS? If the reference annotation is used as background model, a positive control from public whole genome sequencing data should be added to this panel.

Response:

Thank you for the comment regarding our data uniformity, which is shown in our chromatin feature enrichment analysis. To clarify, the reference annotation used for the background was derived from taking the consensus of the mouse chromHMM dataset across multiple tissues⁴², which we utilized to annotate the genomic state of each CpG site. This reference served as the universe set for calculating odds ratios of CpG overlaps using the KYCG toolkit (<https://github.com/zhou-lab/knowYourCG>), which is now explicitly detailed in the revised

Methods section (Transcription factor motif enrichment). Specifically, our enrichment analysis was performed at the single-CpG level by comparing observed methylated CpGs against the entire annotated CpG set rather than relying on broader region-level overlap calculations.

We agree with the reviewer that real data positive controls should be included to evaluate uniformity besides the reference assembly, which may yield distorted copy number changes from sub-optimally mapped genomic regions. For this purpose, we used multiple previous single-cell whole-genome bisulfite sequencing (scWGBS) datasets instead of whole-genome sequencing data. We think that it better captures the status quo of high-resolution DNA methylome profiling (Extended Data Fig. 5a). As our analysis suggests, the uniformity of Spatial-DMT is comparable to these single-cell methylome datasets. The log2 odds ratios of the depletion at enhancers and enrichment at heterochromatin, as noted by the reviewer, remain consistently within ± 1 (the y-axis in our enrichment plots). We interpret the depletion or enrichment from our datasets as non-significant. We also note that our dataset readily resolves cell identities and the enhancer transcription factor binding signatures linked to each cell type (Fig. 1f-h, Fig. 2e). This aligns with our interpretation that enhancer coverage depletion is slight. The uniform enrichment profiles observed across diverse chromatin states further reinforce the robustness and reliability of our Spatial-DMT approach.

Extended Data Fig. 5a:

5. Will HCl treatment affect the diffusion pattern of DNA and RNA?

Response:

Thanks for the comments. The use of 0.1N HCl treatment in our protocol is based on previously reported methods demonstrated by Zhao et al. (Nature, 2022)⁴³ to preserve tissue architecture. Cerebellar sections were exposed to treatment as stated in the figure below and stained with DAPI in their study⁴³:

To validate spatial integrity further, we overlaid individual clusters with corresponding bright-field images, confirming that the heart-specific cluster in the E11 mouse embryo and the CA1, CA2, CA3, and dentate gyrus (DG) clusters in the P21 mouse hippocampus closely aligned with their respective anatomical regions.

Furthermore, we investigated transcriptional regulation at fine spatial resolution within the craniofacial and forebrain regions of the E11 embryo, using a 10 μm pixel-size microfluidic chip, achieving near single-cell resolution (Fig. 1g, h and Extended Data Fig. 10). Notably, two spatially defined clusters, W7 and W11, captured key telencephalic compartments. W11 was enriched for telencephalon progenitors in the ventricular zone of the pallium, a neurogenic niche characterized by active cell division and proliferation, while W7 corresponded to GABAergic cortical interneurons located in the mantle zone, where newborn neurons migrate, accumulate, and differentiate to establish cortical architecture^{4,16,17} (Fig. 1h and Extended Data Fig. 10d). Gene ontology (GO) enrichment analysis further supported these regional identities, highlighting biological processes associated with neurogenesis and progenitor proliferation in W11, and neuron projection and migration in W7 (Extended Data Fig. 10e). Additionally, the cell-type lineage tree constructed from the scRNA-seq reference confirmed the developmental trajectory, positioning telencephalon progenitors as direct precursors to GABAergic cortical interneurons¹⁸. Beyond the forebrain, our spatial analysis also resolved refined sensory

structures within the developing olfactory system (cluster W5 and W10) (Fig. 1g and Extended Data Fig. 10). Sensory neurons were notably enriched in W5 (Extended Data Fig. 10f, g), spatially localized adjacent to the forebrain. This spatial pattern closely aligns with the established developmental trajectory of the olfactory system, wherein olfactory sensory neurons gradually establish connections with the forebrain as embryogenesis progresses⁵⁻⁷. Collectively, these findings strongly suggest that our HCl treatment effectively preserves the spatial integrity of DNA and RNA throughout the experimental process.

Fig. 1g-h:

Extended Data Fig. 10d-g:

6. In Lines 126-128 and Fig.1c, the comparison is unfair. For example, Shareef 2021 is expanded RRBS which was designed to capture only a subset of the genome but not whole genome as in this method. Also, the numbers from Nichols 2022 is inconsistent with the publication: sciMETv1.LA was reported to have 2,158,578 CG and sciMETV2.SL have 325,034-534,728 CG sites - which are 2-10 folds of the coverage by this method.

Response:

Thank you for pointing out this lack of clarity in our comparisons. We have revised the figures to ensure that the comparisons are now presented more clearly and fairly. Specifically, we have removed the dataset from Shareef et al.'s, as their expanded RRBS method was designed to capture only a subset of the genome. Regarding comparisons with the Nichols et al. (2022) dataset, our initial analysis included only mouse samples. The numbers listed by the reviewer were based on human samples from that study. To address this inconsistency, we have now included data from their human samples (sciMETv2.LA) and merged sciMETv2.SL_N and sciMETv2.SL_NH datasets for a more comprehensive and accurate comparison. As shown in the revised Fig. 1C, our methods are comparable to the splint ligation version of sciMETv2 but fall short compared to the linear amplification version (despite the larger human genome compared to the mouse). This result is consistent with our use of splint ligation in our methods. We expect that the future incorporation of linear amplification will further improve the CpG coverage. These revisions are reflected in the updated Fig. 1c.

Fig. 1C:

7. In addition, the authors should clearly note that there could be multiple cells in a 20um or 50um pixel, and the numbers from reference data are from single cells. Did the author observed 6.25-fold of #CG for 50um compared to 20um $((50/20)^2)$?

Response:

Thank you for the comment. We agree with the reviewer that the relationship between pixel size and the expected number of cells is important background information and that comparison with multi-cell pixel and single-cell data is not entirely fair. In the revised manuscript, we have added the clarification that “Lower-resolution pixels (e.g., 50 μm) captured more UMIs and expressed genes, likely due to the inclusion of more cells within each pixel.”, and we have also noted that the reference datasets used for comparison are derived from single-cell data. To allow for more direct comparisons, our revised manuscript included a 10 μm map of the E11 mouse embryo, with each pixel containing approximately one cell. The 10 μm map captures a similar number of CpGs as sciMETv2 human data and more CpGs than the mouse data.

As expected, larger pixel sizes (e.g., 50 μm) tend to capture more than one cell and thus more CpGs compared to smaller pixels (e.g., 20 μm or 10 μm), given the difference in pixel area. However, the observed increase does not strictly adhere to this theoretical expectation $((50/20)^2 = 6.25)$. Several factors beyond pixel area contribute to these deviations, including:

- Tissue Characteristics: Variability in cell distribution, cell density, cell size, and regional heterogeneity can significantly impact the number of CpGs recovered. For instance, densely populated regions such as the hippocampus typically yield higher read counts compared to less dense regions like the cortex.
- Cell Type Variability: The differing types of cells (e.g., smaller lymphocytes versus larger neurons) may contribute differently to CpG recovery.
- The nonlinear dependence of CpG coverage on library complexity. All other variables are controlled, and the number of cells does not scale linearly with the number of CpGs measured. It is easy to see that in the extreme case, a very large number of cells would saturate the number of CpGs measured, and a further increase in cell number would not increase CpG coverage.
- The nonlinear dependence of CpG coverage on sequencing depth: Independent of pixel size, the total number of sequencing reads has a major effect on the number of CpGs detected. Our libraries were not sequenced to the same sequencing depth to allow a fair comparison.

- Tissue Preparation & Capture Efficiency: Variations in sample preparation and capture efficiency can introduce deviations from the theoretical scaling based purely on pixel dimensions.

Despite these complications, we fully concurred with the reviewer that the relationship between the pixel size is an important technical aspect of our method and influences one's choice of pixel resolution for different applications. We included a discussion of this topic in the Discussion section: "Lower-resolution pixels (e.g., 50 μm) generally capture more CpGs, UMIs, and expressed genes, likely due to the inclusion of a greater number of cells within each pixel. This effect depends on factors such as tissue heterogeneity and cell type variability and results in each pixel representing a mixture of signals from neighboring cells. Further development of advanced spatial computational deconvolution methods is essential for inferring single-cell resolution, minimizing potential biases, and enhancing biological interpretability."

8. What is the cutoff in calling a 5mC site? As a pixel could capture fragments from multiple cells, are there sites methylated in one cell but unmethylated in another cell within one pixel?

Response:

Thank you for the comment. We apologize for any ambiguity in our original description. To clarify, we do not binarize methylation calls at individual cytosines. Instead, methylation levels are represented as continuous values ranging from 0 to 1, reflecting the fraction of methylated reads at each site. This approach allows us to capture heterogeneous methylation states within each pixel, acknowledging that a single pixel may contain DNA fragments from multiple cells, some of which may have a cytosine methylated while others do not.

Consequently, the reported methylation level reflects an averaged methylation state across these fragments—analogue to bulk bisulfite sequencing data. We have updated the Methods section (Data Preprocessing) to clarify this in the revised manuscript:

"We employed the BISulfite-seq CUI Toolkit (BISCUIT, version 0.3.14) to align the DNA sequences to the mouse reference genome (mm10). Methylation levels at individual CG and CH sites were stored as continuous values between 0 and 1, representing the fraction of methylated reads after quality filtering."

It is possible that a site is methylated in one cell but unmethylated in another cell within the same pixel. We plotted the proportion of such sites across samples with different pixel sizes. These sites generally account for less than 3% of the sequenced sites, and smaller pixel sizes tend to have a lower proportion.

9. Another technical concerns about the purity of DNA modality. Can streptavidin beads pull down complete remove cDNA from supernatant that will be used for DNA library preparation? Since a SLP5 adaptor ligation is used for ssDNA library prep after APOBEC treatment, will this SLP5 adaptor also ligated to cDNA that remained in DNA library (if the beads pull down is not 100.00%)? A simple way to check is the calculate the percentage of reads that start with a TSO sequence in DNA library.

Response:

Thank you for raising this important point regarding the purity of our DNA library. We agree that it is crucial to ensure that streptavidin beads effectively remove residual cDNA from the supernatant prior to DNA library preparation. To evaluate potential cDNA contamination, we analyzed the raw FASTQ reads from the DNA library for the presence of poly-A (≥ 30 consecutive adenines), poly T (≥ 30 consecutive Thymine), and the TSO sequence (AAGCAGTGGTATCAACGCAGAGTACATGGG). As expected, no reads containing the TSO sequence were detected since there is no TSO reaction for the DNA library preparation, and this sequence is removed during RNA library construction via the Nextera XT reaction in our protocol (Fig. 1a and b, Extended Data Fig. 1). Additionally, the proportions of reads containing poly A or poly T were consistently below 0.3% across all samples (Extended Data Fig. 4f). These results confirm that our DNA library is free of cDNA contamination and demonstrate the high purity of the DNA modality captured by our Spatial-DMT protocol.

Extended Data Fig. 4f:

10. Lines 86-87, the authors stated 30 min - 1 hour tagmentation for twice is the optimized condition for improving gDNA yield. Can the authors provide the comparisons of different conditions during their optimization. This information could be useful for readers as to balance yield and experimental time.

Response:

Thanks for the comments. The use of multi-tagmentation strategy in our protocol is based on previously reported methods demonstrated by Zhao et al. (Nature, 2022)⁴³. They demonstrated that a multi-round tagmentation can increase the yield of genomic DNA. In their study, they quantified DNA fragments per bead across different tissues using the Slide-DNA-seq array and showed that the “4x” protocol variant—incorporating four rounds of tagmentation—produced a notably higher DNA fragment yield, as illustrated in the figure below:

In our protocol, we implemented two rounds of tagmentation instead of four, aiming to balance DNA yield with experimental time, and to minimize the risk of RNA degradation. In our optimization experiments, we compared single versus two rounds of tagmentation and found that performing two rounds can also result in higher gDNA yields compared to a single round (as shown below). In the revised manuscript, we have cited the work from Zhao et al. (Nature, 2022)⁴³ and clarified the rationale for adopting a two-round tagmentation strategy:

“To further reduce the size of the large gDNA fragment and improve yield, we adopted a multi-tagmentation strategy as previously demonstrated⁴³. Specifically, we implemented two rounds of tagmentation to balance DNA yield with experimental time and to minimize the risk of RNA degradation.”

1x Tn5 tagmentation (1 hour):

Region Table

From [bp]	To [bp]	Average Size [bp]	Conc. [ng/μl]	Region Molarity [nmol/l]	% of Total	Region Comment	Color
200	1200	715	8.97	22.4	58.86		■

2x Tn5 tagmentation (2x 1 hour = 2 hours):

Region Table

From [bp]	To [bp]	Average Size [bp]	Conc. [ng/μl]	Region Molarity [nmol/l]	% of Total	Region Comment	Color
200	1200	619	23.5	67.8	71.49		■

There are multiple places that could be improved from the analytical aspect. For example: 11. A general comment is that the histological images could be provided side-by-side with molecular profiles when applicable.

Response:

Thanks for the suggestion. Currently, it remains a technical challenge to perform H&E staining and spatial multi-omics analysis on the same slide due to potential interference with downstream molecular profiling. As an alternative, we provide corresponding bright-field histological images as reference, displayed side-by-side with the molecular profiles in the revised manuscript (Fig. 1f–h, Fig. 4a-b, and Extended Data Fig. 11a), to facilitate visual comparison and spatial interpretation.

Fig. 1f-h:

Fig. 4a-b:

Extended Data Fig. 11a:

12. Lines 210-212, Fig2. abc and Fig. 3ef. It is inappropriate to compare the expression levels of genes with the overlapping VMR. There is no evidence the given VMR is regulating this gene, and what is there are multiple VMR overlapping with one gene but showing different methylation levels?

Response:

Thank you for the comment. We agree that the overlap of a VMR with a gene alone does not necessarily establish a direct regulatory relationship; we did not intend to suggest causality. Instead, we aimed to highlight only correlations. This analytical approach is consistent with methods used in many previous studies (e.g., Kremer et al. *Nature*, 2024)⁴⁴. However, we fully acknowledge that functional validation through perturbation experiments is required to confirm any direct regulatory roles.

Additionally, as the reviewer noted, multiple VMRs may overlap with a single gene, each potentially exhibiting different methylation levels. This scenario underscores the complexity inherent in epigenetic regulation, wherein methylation occurring in different genomic elements—such as promoters, enhancers, or gene bodies—can have distinct and context-dependent effects on gene expression. For example, Fig. 4c-h demonstrates how distinct CA and CG VMRs overlapping the same gene may correlate differently with gene expression levels, possibly due to their presence within different enhancer types and genomic contexts. The following genome browser view indicates the genomic locations of the CG and CA VMRs with respect to the *Prox1*, *Ntrk3*, *Cux1* genes highlighted in Fig. 4f-h.

We revised the manuscript and added the following language in the Discussion to state this limitation and the complexity of the methylation-expression relationship clearly:

“Importantly, the multi-modal nature of Spatial-DMT uncovered intricate relationships between DNA methylation and transcription within the tissue context. Integrating DNA methylation and transcription from the same tissue section delineated the cell-type and region-specific mechanisms of mCG and mCH regulation and their potential roles in transcriptional programs. Our data indicate that gene expression may be differentially influenced by mCG and mCA in a spatially restricted manner, potentially due to the distinct genomic distribution of these methylation marks. While the observed associations are correlative, they offer valuable insights into the complex interplay of regulatory elements on gene expression.”

Response figure showing CA/CG VMR locations with respect to Prox1, Ntrk3, Cux1 gene transcripts, and their chromHMM annotation:

13. In Fig. 1f, while the inconsistency of clustering between 5mC and RNA (typically lower resolution in 5mC) is reasonable, but there is a subset of the cells was grouped as one cell type (R3) by RNA, but it was combined with R2 and classified as D0 in DNA (D0 and D4). What is the biological basis of RNA modality have higher resolution for D4 cells (R1, R3 and R12) but lower resolution for D0 cells (R2 & R3)? A consensus matrix can be provided to show the DNA-RNA clustering relationships more clearly.

Response:

Thank you for making this keen observation. Indeed, we do not believe there is a simple subset-superset relationship between DNAm and RNA modalities in resolving cell identity. Rather, the two molecular layers work in concert to define cell identity, each offering complementary information. In some cases, a cell population may be more clearly

distinguished by its transcriptomic profile, while in others, epigenetic features provide stronger discriminatory power.

In the case mentioned by the reviewer, RNA cluster R3 is subdivided into two distinct DNA clusters, D0 and D4. Indeed, further DNA methylation analysis of these clusters revealed significant epigenetic differences, notably enriched in key developmental transcription factors. Specifically, cluster D0 exhibited enrichment of *Pitx1*, a transcription factor associated with facial morphogenesis²⁴, while cluster D4 showed enrichment for transcription factors such as *Hoxa1* and *Gata6*, known regulators crucial for cardiovascular development^{25,26}. In contrast, differential RNA expression analysis between these same clusters yielded fewer distinct markers, resulting in lower spatial resolution (Extended Data Fig. 16f, g).

We think this is potentially reflective of regulatory redundancy where distinct epigenetic mechanisms converge to produce similar transcriptional outcomes. Conversely, the opposite scenario is equally plausible—transcriptional states can diverge despite similar epigenetic landscapes, as epigenetic regulation is only one layer influencing gene expression. We revised the manuscript to include a discussion of the biological basis of RNA-DNA cooperation in shaping the molecular cell identity. In the Discussion, we now wrote: “This integrative data analysis also demonstrated that each modality captures distinct yet complementary aspects of cellular states, enhancing cell states differentiation beyond what is achievable with single-modality approaches. This may reflect regulatory redundancy, where distinct epigenetic mechanisms converge to produce similar transcriptional outcomes. Conversely, the opposite scenario is equally plausible—transcriptional states may diverge despite similar epigenetic landscapes, as epigenetic regulation represents only one layer influencing gene expression.”

Extended Data Fig. 16f, g:

Following the reviewer’s suggestion, we have now included a consensus matrix to

summarize the correspondence between DNA and RNA clusters. The matrix aligns with the reviewer's observation: R3 contains D0 and D4. R2 and R3 combined to D0.

DNA-RNA clustering consensus matrix:

14. Fig. 1f, there are a few straight horizontal lines in DNA spatial distribution that grouped as D3. Is this technical artifact or true biology?

Response:

Thank you for your comments. We investigated these features and determined they are likely technical artifacts arising from lower read counts during the barcoding flow step.

To evaluate reproducibility, we performed an independent replicate experiment, which confirmed that these artifacts could be resolved. The updated results (Extended Data Fig. 3b, c) demonstrate that these patterns are technical artifacts rather than biological effects that can be mitigated.

Extended Data Fig. 3b, c:

15. What is the genome distribution patterns of VMR? Did the CG and CA VMR separately calculated? What are the statistics in calling these VMRs?

Response:

Thank you for your comments. We employed the MethSCAN tool (Kremer et al. *Nature Methods*, 2024)⁴⁰ to identify VMRs, and their genome-wide distribution patterns are shown in Extended Data Fig. 5D.

Extended Data Fig. 5D:

To clarify, CG and CA VMRs were calculated separately. VMRs were defined as fused genomic intervals exhibiting methylation variance within the top 2 percentile, following the methodology described in the original MethSCAn publication^{40,44}. We have updated the Methods section (Data Preprocessing) to provide a clearer explanation of this approach:

"These processed CG/CH files were then independently analyzed using the MethSCAn pipeline to identify variably methylated regions (VMRs), defined as fused genomic intervals exhibiting methylation variance within the top 2 percentile. We used default parameter settings when running MethSCAn, and the MethSCAn filter min-sites parameter was determined from the read coverage knee plot (Extended Data Fig. 4c)."

Minor:

1. The size of slides/tissue sections should be clearly labeled.

Response:

We have now added the size of the slide to the corresponding Figures legend.

2. Line 194, TET2 full name is ten-eleven translocation methylcytosine dioxygenase 2.

Response:

Thank you for pointing this out. We have updated the manuscript by providing the full name of TET2 to ten-eleven translocation methylcytosine dioxygenase 2 upon its first occurrence.

3. Unit label are missing from multiple Fig. panels, for example, Fig. 2c, 3e,f, h.

Response:

We have now added the unit label (methylation percentage and log normalized expression) for these Figure panels in the corresponding Figure legend.

4. Line 235, are there any literature support for the statement that Dnmt1 expression are positively associated with mitotic division?

Response:

Thank you for your comment. It was based on the established role of Dnmt1 as part of the DNA replication program. In cancer, where cell division is more active, Dnmt1 is more expressed in order to maintain the epigenetic identity. Several reports were aligned with this.

For example, Fig. 7g from Zhou et.al (*Nature Genetics*, 2018)⁴⁵ demonstrates the positive association between Dnmt1 expression and mitotic division. Similarly, other papers^{46,47} showed that inactivation of Dnmt1 leads to mitotic cycle arrest. However, we recognized that embryonic development is a different biological setting, and we only observed differences between two developmental stages. Therefore, we were likely speculative in drawing this conclusion. We removed this discussion in the revised manuscript for rigor.

Fig. 7g from Zhou et.al (*Nature Genetics*, 2018)

5. Lines 244-246, Fig. S8d cannot support this statement.

Response:

Thank you for pointing out our lack of clarity. From this Extended Data Figure (as shown below), the mCG VMRs-gene pairs have a more polarized distribution than mCA VMR-gene pairs. This is supported not by a difference in the mean of the distribution but by their difference in the variances. This is consistent with the more predominant mCG modification in most tissues and their stronger biochemical interaction with transcriptional machinery, compared to mCA modifications. Further, we note that a greater proportion of variably methylated regions exhibit a negative correlation with gene expression levels. To eliminate

any ambiguity, we have revised the Result section (Spatial co-mapping of mCH, mCG, and RNA transcription on postnatal mouse brain) accordingly:

" Across both sequence contexts, negative correlations between DNA methylation and gene expression are more prevalent than positive ones, highlighting the predominantly repressive nature of these epigenetic modifications (Extended Data Fig. 13d)."

Extended Data Fig. 13d:

6. Lines 272, Fig. 4h did show mCG-dependent RNA expression level.

Response:

Thank you for your comment. We agree that the CA3 region also exhibits an mCG-dependent RNA expression level. To accurately reflect this observation, we have updated the Result section (Spatial co-mapping of mCH, mCG, and RNA transcription on postnatal mouse brain) as follows:

"Lastly, the silencing of *Cux1*, a TF involved in neuronal development and function, exhibited a negative correlation with CA and CG hypermethylation in the CA3 region. In contrast, in the CA1/2 region, *Cux1* expression showed a negative correlation only with CA hypermethylation and appeared independent of mCG levels (Fig. 4e, h)."

References

- 1 Qiu, C. *et al.* A single-cell time-lapse of mouse prenatal development from gastrula to birth. *Nature* **626**, 1084-1093 (2024). <https://doi.org/10.1038/s41586-024-07069-w>
- 2 Kriegstein, A. & Alvarez-Buylla, A. The glial nature of embryonic and adult neural stem cells. *Annu Rev Neurosci* **32**, 149-184 (2009). <https://doi.org/10.1146/annurev.neuro.051508.135600>
- 3 Marin, O., Valiente, M., Ge, X. & Tsai, L. H. Guiding neuronal cell migrations. *Cold Spring Harb Perspect Biol* **2**, a001834 (2010). <https://doi.org/10.1101/cshperspect.a001834>
- 4 Hu, J. S., Vogt, D., Sandberg, M. & Rubenstein, J. L. Cortical interneuron development: a tale of time and space. *Development* **144**, 3867-3878 (2017). <https://doi.org/10.1242/dev.132852>
- 5 Tufo, C. *et al.* Development of the mammalian main olfactory bulb. *Development* **149** (2022). <https://doi.org/10.1242/dev.200210>
- 6 Kim, B.-R., Rha, M.-S., Cho, H.-J., Yoon, J.-H. & Kim, C.-H. Spatiotemporal dynamics of the development of mouse olfactory system from prenatal to postnatal period. *Frontiers in Neuroanatomy* **17** (2023). <https://doi.org/10.3389/fnana.2023.1157224>
- 7 Treloar, H. B., Miller, A. M., Ray, A. & Greer, C. A. *Development of the olfactory system*. Vol. 20092457 (2010).
- 8 Liu, Y. *et al.* High-Spatial-Resolution Multi-Omics Sequencing via Deterministic Barcoding in Tissue. *Cell* **183**, 1665-1681.e1618 (2020). <https://doi.org/https://doi.org/10.1016/j.cell.2020.10.026>
- 9 Zhang, D. *et al.* Spatial epigenome–transcriptome co-profiling of mammalian tissues. *Nature* **616**, 113-122 (2023). <https://doi.org/10.1038/s41586-023-05795-1>
- 10 Deng, Y. *et al.* Spatial profiling of chromatin accessibility in mouse and human tissues. *Nature* **609**, 375-383 (2022). <https://doi.org/10.1038/s41586-022-05094-1>
- 11 Cao, Y. *et al.* Single-cell bisulfite-free 5mC and 5hmC sequencing with high sensitivity and scalability. *Proceedings of the National Academy of Sciences* **120**, e2310367120 (2023). <https://doi.org/doi:10.1073/pnas.2310367120>
- 12 Nichols, R. V. *et al.* High-throughput robust single-cell DNA methylation profiling with sciMETv2. *Nature Communications* **13**, 7627 (2022). <https://doi.org/10.1038/s41467-022-35374-3>
- 13 Liu, H. *et al.* DNA methylation atlas of the mouse brain at single-cell resolution. *Nature* **598**, 120-128 (2021). <https://doi.org/10.1038/s41586-020-03182-8>
- 14 Lister, R. *et al.* Global Epigenomic Reconfiguration During Mammalian Brain Development. *Science* **341**, 1237905 (2013). <https://doi.org/doi:10.1126/science.1237905>
- 15 Kinde, B., Gabel, H. W., Gilbert, C. S., Griffith, E. C. & Greenberg, M. E. Reading the unique DNA methylation landscape of the brain: Non-CpG methylation, hydroxymethylation, and MeCP2. *Proceedings of the National Academy of Sciences* **112**, 6800-6806 (2015). <https://doi.org/doi:10.1073/pnas.1411269112>

- 16 Kriegstein, A. & Alvarez-Buylla, A. The Glial Nature of Embryonic and Adult Neural Stem Cells. *Annual Review of Neuroscience* **32**, 149-184 (2009).
<https://doi.org/https://doi.org/10.1146/annurev.neuro.051508.135600>
- 17 Marín, O., Valiente, M., Ge, X. & Tsai, L.-H. Guiding neuronal cell migrations. *Cold Spring Harbor perspectives in biology* **2**, a001834 (2010).
- 18 Qiu, C. *et al.* A single-cell time-lapse of mouse prenatal development from gastrula to birth. *Nature* **626**, 1084-1093 (2024). <https://doi.org/10.1038/s41586-024-07069-w>
- 19 Liu, W., Li, Y. & Luo, B. Current perspective on the regulation of FOXO4 and its role in disease progression. *Cellular and Molecular Life Sciences* **77**, 651-663 (2020).
<https://doi.org/10.1007/s00018-019-03297-w>
- 20 Lu, C. *et al.* Essential transcription factors for induced neuron differentiation. *Nature Communications* **14**, 8362 (2023). <https://doi.org/10.1038/s41467-023-43602-7>
- 21 Mao, L. *et al.* HOXC9 Links Cell-Cycle Exit and Neuronal Differentiation and Is a Prognostic Marker in Neuroblastoma. *Cancer Research* **71**, 4314-4324 (2011).
<https://doi.org/10.1158/0008-5472.Can-11-0051>
- 22 Vidarsson, H. *et al.* The Forkhead Transcription Factor Foxi1 Is a Master Regulator of Vacuolar H⁺-ATPase Proton Pump Subunits in the Inner Ear, Kidney and Epididymis. *PLOS ONE* **4**, e4471 (2009). <https://doi.org/10.1371/journal.pone.0004471>
- 23 Xu, Y. *et al.* Tlx1 and Tlx3 Coordinate Specification of Dorsal Horn Pain-Modulatory Peptidergic Neurons. *The Journal of Neuroscience* **28**, 4037-4046 (2008).
<https://doi.org/10.1523/jneurosci.4126-07.2008>
- 24 Stelzer, G. *et al.* The GeneCards Suite: From Gene Data Mining to Disease Genome Sequence Analyses. *Current Protocols in Bioinformatics* **54**, 1.30.31-31.30.33 (2016). <https://doi.org/https://doi.org/10.1002/cpbi.5>
- 25 Maitra, M., Koenig, S. N., Srivastava, D. & Garg, V. Identification of GATA6 Sequence Variants in Patients With Congenital Heart Defects. *Pediatric Research* **68**, 281-285 (2010). <https://doi.org/10.1203/PDR.0b013e3181ed17e4>
- 26 Makki, N. & Capecchi, M. R. Cardiovascular defects in a mouse model of HOXA1 syndrome. *Human Molecular Genetics* **21**, 26-31 (2011).
<https://doi.org/10.1093/hmg/ddr434>
- 27 Savolainen, S. M., Foley, J. F. & Elmore, S. A. Histology Atlas of the Developing Mouse Heart with Emphasis on E11.5 to E18.5. *Toxicologic Pathology* **37**, 395-414 (2009). <https://doi.org/10.1177/0192623309335060>
- 28 Frade, J. M. & and Ovejero-Benito, M. C. Neuronal cell cycle: the neuron itself and its circumstances. *Cell Cycle* **14**, 712-720 (2015).
<https://doi.org/10.1080/15384101.2015.1004937>
- 29 Goldman, S. A. & Chen, Z. Perivascular instruction of cell genesis and fate in the adult brain. *Nature Neuroscience* **14**, 1382-1389 (2011).
<https://doi.org/10.1038/nn.2963>
- 30 Shareef, S. J. *et al.* Extended-representation bisulfite sequencing of gene regulatory elements in multiplexed samples and single cells. *Nature Biotechnology* **39**, 1086-1094 (2021). <https://doi.org/10.1038/s41587-021-00910-x>

- 31 Hernando-Herraez, I. *et al.* Ageing affects DNA methylation drift and transcriptional cell-to-cell variability in mouse muscle stem cells. *Nature Communications* **10**, 4361 (2019). <https://doi.org/10.1038/s41467-019-12293-4>
- 32 Caldwell, B. A. *et al.* Functionally distinct roles for TET-oxidized 5-methylcytosine bases in somatic reprogramming to pluripotency. *Molecular Cell* **81**, 859-869.e858 (2021). <https://doi.org/10.1016/j.molcel.2020.11.045>
- 33 Bai, D. *et al.* Simultaneous single-cell analysis of 5mC and 5hmC with SIMPLE-seq. *Nature Biotechnology* (2024). <https://doi.org/10.1038/s41587-024-02148-9>
- 34 Fabyanic, E. B. *et al.* Joint single-cell profiling resolves 5mC and 5hmC and reveals their distinct gene regulatory effects. *Nature Biotechnology* **42**, 960-974 (2024). <https://doi.org/10.1038/s41587-023-01909-2>
- 35 Spruijt, Cornelia G. *et al.* Dynamic Readers for 5-(Hydroxy)Methylcytosine and Its Oxidized Derivatives. *Cell* **152**, 1146-1159 (2013). <https://doi.org/10.1016/j.cell.2013.02.004>
- 36 Goldberg, D. C. *et al.* Scalable Screening of Ternary-Code DNA Methylation Dynamics Associated with Human Traits. *bioRxiv*, 2024.2005.2017.594606 (2025). <https://doi.org/10.1101/2024.05.17.594606>
- 37 Frade, J. M. & Ovejero-Benito, M. C. Neuronal cell cycle: the neuron itself and its circumstances. *Cell Cycle* **14**, 712-720 (2015). <https://doi.org/10.1080/15384101.2015.1004937>
- 38 Zeisel, A. *et al.* Molecular Architecture of the Mouse Nervous System. *Cell* **174**, 999-1014.e1022 (2018). <https://doi.org/10.1016/j.cell.2018.06.021>
- 39 Morrison, J. *et al.* Evaluation of whole-genome DNA methylation sequencing library preparation protocols. *Epigenetics & Chromatin* **14**, 28 (2021). <https://doi.org/10.1186/s13072-021-00401-y>
- 40 Kremer, L. P. M. *et al.* Analyzing single-cell bisulfite sequencing data with MethSCAN. *Nature Methods* **21**, 1616-1623 (2024). <https://doi.org/10.1038/s41592-024-02347-x>
- 41 He, Y. *et al.* Spatiotemporal DNA methylome dynamics of the developing mouse fetus. *Nature* **583**, 752-759 (2020). <https://doi.org/10.1038/s41586-020-2119-x>
- 42 Ernst, J. & Kellis, M. Chromatin-state discovery and genome annotation with ChromHMM. *Nature Protocols* **12**, 2478-2492 (2017). <https://doi.org/10.1038/nprot.2017.124>
- 43 Zhao, T. *et al.* Spatial genomics enables multi-modal study of clonal heterogeneity in tissues. *Nature* (2021). <https://doi.org/10.1038/s41586-021-04217-4>
- 44 Kremer, L. P. M. *et al.* DNA methylation controls stemness of astrocytes in health and ischaemia. *Nature* **634**, 415-423 (2024). <https://doi.org/10.1038/s41586-024-07898-9>
- 45 Zhou, W. *et al.* DNA methylation loss in late-replicating domains is linked to mitotic cell division. *Nature Genetics* **50**, 591-602 (2018). <https://doi.org/10.1038/s41588-018-0073-4>
- 46 Milutinovic, S., Zhuang, Q., Niveleau, A. & Szyf, M. Epigenomic Stress Response: KNOCKDOWN OF DNA METHYLTRANSFERASE 1 TRIGGERS AN INTRA-S-PHASE ARREST OF DNA REPLICATION AND INDUCTION OF STRESS RESPONSE GENES*.

- Journal of Biological Chemistry* **278**, 14985-14995 (2003).
<https://doi.org/https://doi.org/10.1074/jbc.M213219200>
- 47 Jackson-Grusby, L. *et al.* Loss of genomic methylation causes p53-dependent apoptosis and epigenetic deregulation. *Nature Genetics* **27**, 31-39 (2001).
<https://doi.org/10.1038/83730>

Spatial joint profiling of DNA methylome and transcriptome in mammalian tissues

Responses to reviewers' comments

We sincerely appreciate the thoughtful and constructive feedback provided by the reviewers. Their comments have been highly valuable in enhancing the clarity, rigor, and overall quality of our manuscript. In response, we have conducted new experiments, expanded analyses, and thoroughly addressed all the points raised by the reviewers. Below, we provide detailed, point-by-point responses to each reviewer's comment, along with descriptions of the corresponding updates made to the revised manuscript.

Table of Contents:

Referee #1	2
Referee #2	9
Referee #3	10
References	17

Referee #1 (Remarks to the Author):

In the resubmission of this paper by Lee et al, the authors have responded to most comments from me and other reviewers and added additional data and analyses. These additional details, figures and data alleviate some of my concerns, and I commend the authors for being responsive and comprehensive in addressing the comments. However, I still have significant concerns about the quality of the data that the method can produce, some of which are actually highlighted by the additional data the authors provide. My concerns are especially about the quality of the RNA data, which does not seem to be very useful to identify well-known cell types that should be resolved due to their spatial distribution. Additionally, 10um experiments should provide near single-cell resolution, but the paper does not show that specific cell types can be identified from RNA-seq data. Also, there is no comparison of the DNA methylation data to sc DNA methylation data (there are several datasets available). More details below:

1. I have concerns with the quality of RNA data, especially for the postnatal brain. From Extended Data Fig. 10c, it is evident that the spatial clusters do not map well on the scRNA-seq data, with spatial clusters distributed widely across scRNA-seq clusters. It is in contrast to similar data from Zhang et al., *Nature*, 2023 (Figure 2d, 5c), which the authors cite, which shows much better correspondence between scRNA-seq and spatial clusters.

Response:

Thanks for the comments. We would like to clarify that Extended Data Fig. 10c, which integrates spatial-DMT RNA data (10 μ m resolution, colored by WNN clusters) with scRNA-seq data from the E11 mouse embryo¹, does show good correspondence between spatial clusters and single-cell transcriptomic clusters. The perceived lack of correspondence may stem from the scRNA-seq embedding (Extended Data Fig. 10b), which was dominated by more distinct cell types, masking separation within broader populations. For clarity, we enumerated the correspondence of the spatial cluster to scRNA-seq clusters.

- Spatial cluster W11 (Telencephalon cells; Fig. 1h) maps to Neuroectoderm and glia cluster in single-cell data, which include telencephalic populations.
- Spatial cluster W7 (GABAergic cortical interneurons cells; Fig. 1h) aligns with CNS neuron single-cell clusters, where GABAergic interneurons are represented.
- Spatial cluster W5 (Extended Data Fig. 10f (right)) corresponds to Olfactory sensory neurons.
- Spatial cluster W10 (Olfactory epithelial cells; Extended Data Fig. 10f (middle)) maps to epithelial cells single-cell cluster.
- Spatial clusters W0-W2 correspond to the mesoderm single-cell cluster.

To benchmark against a comparable tissue type and spatial resolution reported by Zhang et al., *Nature*, 2023, we further evaluated our P21 mouse brain dataset and observed good correspondence with expected cell types (Fig. 4b; Extended Data Fig. 15b, c). For example,

- Spatial cluster W3 aligns with dentate gyrus granule neurons and neuroblasts single-cell clusters (Extended Data Fig. 15b, c), localized to the dentate gyrus granule cell layer (Fig. 4b).
- Spatial clusters W5 and W6 correspond to Telencephalon projecting excitatory neurons single-cell cluster (Extended Data Fig. 15b, c), located in CA1/2 and CA3 regions, respectively (Fig. 4b).
- Spatial clusters W1 and W4 map to a distinct excitatory neuron population (Extended Data Fig. 15b, c), which are located in the cerebral cortex (Fig. 4b).
- Spatial cluster W0 corresponds to Oligos and Astrocyte single-cell cluster (Extended Data Fig. 15b, c), located in the stratum lacunosum-moleculare, stratum oriens, and stratum radiatum of CA1-CA3, as well as the molecular layer of dentate gyrus (Fig. 4b).

Extended Data Fig. 15b, c:

In comparison, the spatial RNA data from Zhang et al. (*Nature*, 2023)² (Fig. 2d, P21 mouse brain, 20 μ m) showed that some spatial clusters did not co-embed with the reference scRNA-seq clusters, while others exhibited mixing across different cell types within the same single-cell cluster (as highlighted by red arrows in their Fig. 2d below).

- Fig. 2d in Zhang et al. *Nature*, 2023²:

In addition to visual inspection, the RNA data quality from Spatial-DMT is supported in other analyses, some of which were suggested by the reviewers' other comments below:

- (1) As shown in Fig. 1e, key RNA quality control metrics indicate that the transcriptomic performance of Spatial-DMT is comparable to that of DBiT-seq.
- (2) Unsupervised clustering of spatial RNA data identified distinct tissue and cell types in both the embryos (Fig. 1f, 1g; Extended Data Fig. 11) and P21 mouse brain (Fig. 4b). Marker gene expression and GO analysis further supported the spatial identity of these clusters, correlating well with known histological features in embryonic tissues and brain regions (Fig. 2b, 2c; Extended Data Figs. 6, 7b,c, 8b,c, 9, 10d-g, and 14).
- (3) We performed cell-type decomposition of spatial transcriptomic pixels using scRNA-seq references^{1,3}. For example, in the E11 embryo, spatial clusters W7 and W11 delineated key telencephalic compartments—W11 enriched for telencephalon progenitors in the ventricular zone of the pallium, and W7 corresponding to GABAergic cortical interneurons in the mantle zone (Fig. 1h; Extended Data Fig. 10d). These assignments were supported by spatial marker expression and validated by ISH images from the Allen Developing Mouse Brain Atlas (Extended Data Fig. 10d). Similarly, we were able to resolve different cell types in P21 using spatial RNA data, and aggregated gene expression across pixels in each cluster was highly correlated with the corresponding single-cell reference transcriptomes (Extended Data Fig. 15d, e, shown for W3 (DG) and W5 (CA1)).
- (4) Co-embedding with the spatial RNA data from Zhang et al. (*Nature*, 2023)² revealed strong concordance (Extended Data Fig. 12).

In summary, our comprehensive evaluation demonstrates that the RNA data from Spatial-DMT are of comparable, if not better, quality to existing spatial methods and are well-suited for cell type identification. The method not only achieves robust global metrics but also enables biologically meaningful interpretations.

2. I did not see any data with UMAP/clustering of spatial P21 data to show that at least the main cell types can be identified (oligos, deep-layer neurons, granule neurons of the hippocampus etc).

Response:

We appreciate the reviewer's request for a more explicit demonstration of cell type identification in the P21 brain dataset. In our previous revision, as the reviewer suggested, we had indirectly addressed this by showing spatial expression of marker genes localized to expected anatomical regions (Extended Data Fig. 14). To resolve cell identities, we integrated our Spatial-DMT dataset with a reference scRNA-seq dataset³ to associate spatial clusters with previously defined cell types. The integration UMAP shows Spatial-DMT data co-clustering with their matching cell type clusters from scRNA-seq, indicating that our spatial transcriptomes capture the defining transcriptomic signatures of each major cell type in the P21 brain (Extended Data Fig. 15a-c). For instance, W0, W3, and W5 were identified as oligodendrocytes, dentate gyrus granule neurons, and telencephalon-projecting excitatory

neurons, respectively. Furthermore, aggregated gene expression across pixels in each spatial cluster was highly correlated with the corresponding reference single-cell transcriptomes (Extended Data Fig. 15d, e, shown for W3 (DG) and W5 (CA1)).

To further refine the cell type identification, we conducted cell-type deconvolution using the scRNA-seq reference³ (New Extended Data Fig. 15f). This analysis revealed well-defined spatial organization of diverse cell types, consistent with known brain anatomy. For example, cortical excitatory neurons were arranged in distinct laminar patterns: TEGLU7, TEGLU8, TEGLU4, and TEGLU3 were enriched in layers 2/3, 4, 5, and 6, respectively. In the hippocampus, TEGLU24 and TEGLU23 mapped to CA1/2 and CA3 regions, while granule neurons DGGRC2 localized to the dentate gyrus. Additionally, DEGLU1 was enriched in the thalamic region. Each of these cell types mapped to anatomically appropriate brain regions in agreement with the scRNA-seq reference³. These findings further demonstrate that the RNA data generated by Spatial-DMT exhibit good quality and are capable of accurately identifying and spatially resolving complex cell types in the brain. We have updated the Results section to reflect these findings.

Extended Data Fig. 15f:

3. The paper is almost fully focused on embryonic mouse analysis. It gives the impression that the technique only works well with this particular sample type, with very little data from other ages and specific brain regions. This is in contrast to similar papers that demonstrate the performance of their method with several types of samples. Using the method to analyze cortical lamina (the entire span of the cortex), as well as striatum or thalamus, in addition to the hippocampus would alleviate some of these concerns.

Response:

We thank the reviewer for the comment about the diversity of samples to demonstrate the Spatial-DMT technology. Our primary aim in this study is to introduce and validate a novel spatial multi-omics technology (joint DNA methylome and transcriptome profiling within the same tissue section), and for that initial validation, we focused on mouse embryonic tissues as proof-of-concept. We chose embryos because they inherently contain a wide range of tissue types (brain, spinal cord, heart, craniofacial regions, etc.), all in one sample, allowing us to test the method's performance across diverse cell types and organs within a single experiment. We agree, however, that showing the method's applicability beyond embryonic tissue is important to demonstrate its general utility. To address this, we have included data from a postnatal day 21 (P21) mouse brain in our manuscript, which we had presented before and have now expanded. This P21 sample covers both the hippocampus and the adjacent cerebral cortex, and also covers part of the thalamus. Therefore, the P21 dataset already encompasses multiple distinct brain regions, as the reviewer mentioned, and the results clearly show that Spatial-DMT works effectively in more mature tissues. For example, we were able to resolve the cellular organization of the hippocampus (including granular and neuron populations) and identify layered structures in the cortex (as noted above, distinct cortical lamina were distinguished in our integrated analysis). Additionally, the P21 data allowed us to identify neurons that were enriched in the thalamus within the same section. These findings demonstrate that the technique is not limited to embryonic samples – it performs well in other ages and different brain regions. We have updated the Results section to reflect these findings.

Furthermore, to demonstrate the broader applicability of Spatial-DMT, we performed additional experiments on a human adult lymph node sample (unpublished data from ongoing work). This result illustrates that Spatial-DMT can be extended to non-embryonic, non-brain samples and applied across diverse tissue types and species.

Spatial-DMT mapping on a human lymph node:

Unsupervised clustering of spatial DNA methylation, RNA, and WNN data revealed distinct spatial domains within the lymph node, including the T cell zone (W0), germinal centers (GC) (W1), and B follicle zone (W2). To resolve specific immune cell types, we performed cell-type deconvolution using a scRNA-seq reference⁴. This analysis identified naive B cells enriched in the follicle zone, germinal center dark zone B cells (B-GC-DZ) and light zone B cells (B-GC-LZ) enriched in GC regions, and CD4+ T cells localized to the T cell zone.

While additional validation across more tissue types, developmental stages, and species is possible, we consider such expansion to be part of future studies. Our current results lay a strong foundation for extending Spatial-DMT to broad biological contexts. We have included this discussion in the revised manuscript.

4. There is no comparison or integration of the spatial DNA methylation data with publicly available single-cell DNA methylation data. Without these analyses, it is impossible to judge whether the method truly works. Such analyses have been done with Spatial ATAC.

Response:

We thank the reviewer for their suggestion to compare our spatial DNA methylation data with public single-cell DNA methylation datasets.

First, we would like to clarify that, as suggested in the previous round of review, we have performed comprehensive benchmarking of our spatial DNA methylation data against existing single-cell methylation datasets:

- (1) Key DNA methylation quality control metrics, including CpG coverage (Fig. 1c) and duplication rate (Extended Data Fig. 4b), were directly compared with those from previously published single-cell DNA methylation studies in mouse embryos and brain⁵⁻⁸. Our results demonstrate comparable performance to these published datasets.
- (2) We also assessed the genomic distribution of CpG coverage and found it to be uniformly distributed across genomic regions (Extended Data Fig. 5a), in agreement with public single-cell DNA methylation references⁵⁻⁸. Additionally, DNA methylation levels across various chromatin states were consistent with known biology and were comparable with those published databases⁵⁻⁸ (Extended Data Fig. 5b).
- (3) Unsupervised clustering of spatial DNA methylation data revealed distinct tissue and cell types in both the embryos (Fig. 1f, 1g; Extended Data Fig. 11) and P21 mouse brain (Fig. 4b). The methylation levels of well-characterized marker genes further supported the spatial identity of these clusters and showed strong concordance with known histological features (Fig. 2a, 2c; Extended Data Figs. 6, 7a,c, 8a,c, 9a, 10d-g, and 14).

To further address the reviewer's suggestion, we additionally performed a direct comparative analysis between our spatial DNA methylation data and a publicly available single-cell methylome dataset from the mouse brain⁵. The new analysis demonstrates a strong concordance between our spatial clusters and annotated cell types in the reference dataset

(New Extended Data Fig. 15e), further validating the data quality and biological relevance of our spatial DNA methylation profiles.

Extended Data Fig. 15e:

Referee #2 (Remarks to the Author):

Thanks for addressing my comments and questions satisfactorily. I do not have further concerns.

Response:

Thank you for your thoughtful and constructive feedback, which greatly improved the quality of our manuscript. We appreciate your time and are pleased that your concerns have been fully addressed.

Referee #3 (Remarks to the Author):

The authors' responses addressed most of the issues, including quality control of the data, reproducibility of techniques, and consistency of published data. However there are still several issues.

Response:

Thank you for your continued feedback and thoughtful assessment. We appreciate the additional comments and have addressed all remaining concerns in detail below.

Comments 4: While scWGBS data is a relevant comparator for methylome profiling, it may not adequately control for tagmentation or fragmentation biases, since it also suffers from sparse coverage and noise. Could you clarify why bulk WGS was not included as a control in your enrichment panel, and whether such a comparison might better reflect the expected uniform coverage?

Response:

We thank the reviewer for this suggestion and regret the lack of clarity in our rationale. We acknowledge that compared with bulk WGS, most single-cell methods, even some other library prep methods for bulk, do not rise to the same level of sequence uniformity. Our Spatial-DMT is no exception. We agree with the reviewer that this is likely due to fragmentation bias. However, the purpose of the uniformity analysis is to benchmark the uniformity of Spatial-DMT against comparable, sparsely covered datasets. We selected single-cell WGBS (scWGBS) as the comparator because it represents a well-established technology that, like Spatial-DMT, must contend with low DNA input---a defining challenge for both approaches, while achieving high cell throughput. Such challenges are absent in bulk WGS or WGBS datasets, where global coverage of the whole genome can be readily achieved.

We also acknowledge the reviewer's point that tagmentation-based methods may suffer more from coverage non-uniformity. To partially address this, we included a scWGBS dataset that did not rely on tagmentation (Liu et al. *Nature*, 2021)⁵, which used Zymo EZ-96 DNA Methylation-Direct Kit (cat. #D5023). This dataset showed the best sequence uniformity as expected. However, the plate-based method is incompatible with spatial mapping, and the liquid handler required to perform scalable profiling is not accessible to most labs and hence has limited cell throughput compared to alternatives such as sciMET.

A technical complication that caused us to avoid bulk WGS or bulk WGBS data in our initial \log_2 odds ratio enrichment panel is that the enrichment metric reported by the KnowYourCG tool depends on sequencing depth and may lead to a misleading perception of enrichment in datasets that fully cover the CpG landscape. This tool calculates enrichment based on the overlap between query CpGs and defined genomic annotations. Since bulk WGS/WGBS datasets capture nearly all CpG sites, they will exhibit high overlap, leading to inflated odds

ratios. Therefore, we benchmarked against single-cell DNA methylation datasets of similar data sparsity for \log_2 odds ratio enrichment analysis.

To address the reviewer’s suggestion of including a bulk control, we performed a fold enrichment analysis (observed overlaps divided by expected overlaps), which is more robust to sequencing depth variations and appropriate for comparison with a bulk WGBS dataset. This analysis is shown in the revised manuscript (New Extended Data Fig. 5a), and includes a bulk WGBS dataset from mouse embryos (He et al. *Nature*, 2020)⁹.

The bulk WGBS showed consistent baseline fold enrichment across annotation categories, and our spatial-DMT data showed a uniform distribution across genomic regions as well. We agree with the reviewer that tagmentation-based single-cell or spatial approaches might suffer from sparse coverage and potential fragmentation bias. Therefore, we have now acknowledged this limitation in the revised Discussion section, where we wrote “Further optimization of the Spatial-DMT protocol may improve coverage and minimize potential tagmentation bias”. Additionally, we have updated the Results section to reflect these analyses. Methods sections have also been updated to describe the rationale and methodology behind the fold enrichment analysis.

Extended Figure 5a and 5b:

You interpret \log_2 odds ratios within ± 1 as indicating “non-significant” enrichment or depletion. Could you elaborate on whether statistical testing was performed to support this threshold, rather than relying on visual inspection alone?

Response:

We regret using the term “non-significant” on an arbitrary effect size cutoff without providing explicit statistics. We removed the term in the revised manuscript’s language for rigor. We would like to further clarify that we did investigate statistical significance. As the statistical significance is based on per-cell (single-cell data)/per-pixel (spatial data)/per tissue sample (bulk data) analysis, we used a bar plot to characterize the overall proportion of cells/pixels/tissue samples that pass the FDR threshold. As shown in the figure below, we performed Fisher’s exact tests for each genomic annotation, followed by Benjamini–Hochberg correction for multiple testing (FDR < 0.05). The results present the proportion of spatial pixels (and single cells/tissue samples in other datasets) exhibiting statistically significant enrichment. These analyses confirmed that most categories in our spatial-DMT data do not show significant enrichment, with the exception of the “quiescent” category. This category, the largest genomic category containing over 10 million CpGs, tends to produce inflated p-values due to its size (a pattern also observed in other single-cell datasets). In contrast, and as expected, bulk WGBS datasets showed widespread statistical significance, which is attributed to their high CpG site coverage. As discussed above, applying Fisher’s test to assess uniformity is confounded by differences in the overall genome coverage. To avoid potentially misleading interpretations, particularly for readers unfamiliar with these biases, we chose not to include it in the manuscript.

Fisher exact test significant cells/pixels/tissue samples proportions:

Comments 8: Thank you for the clarification. However, I still have some concerns regarding the interpretation of continuous methylation values in the context of low cell numbers per pixel. Given that each pixel captures limited DNA fragments from only a few cells, the total read count per site is likely low and each CG site per cell can even only be either methylated or unmethylated. In this scenario, continuous values (e.g., 0.25, 0.5, 0.75) may simply reflect stochastic sampling noise rather than true biological heterogeneity.

Response:

We appreciate the reviewer’s point that using continuous values to capture methylation levels in the low-cell-number scenario is counterintuitive. As the reviewer suggested, when each pixel captures only a few cells, and the total read count per CpG site is low, the readouts per CpG should be either methylated or unmethylated. In fact, this is indeed the case from our analysis. For example, in our previous revision, we quantified the per-CpG methylation levels on sites covered by more than one read, and we found that the proportion of reads exhibiting discrepant methylation states, leading to neither 0 nor 1 methylation levels, was rare. In other words, sites are rarely seen with both methylated and unmethylated reads. As expected, this proportion is associated with the pixel size. Bigger pixels capture more cells and display greater methylation heterogeneity. Fewer than 3% of CpG sites in the E11 50 μm dataset and fewer than 1% in the E11 10 μm dataset (near single-cell resolution) exhibited such intermediate values (as shown in the figure below).

We would like to clarify that we reported continuous values in some of our analyses solely for technical reasons. For example, due to the inherent sparsity of single-cell and spatial DNA methylation data, it is impractical to analyze methylation status solely at the individual CpG level. Binary information at sparse loci cannot be directly used to construct a feature matrix suitable for downstream analysis (e.g., PCA)¹⁰. A widely adopted approach in single-cell methylation studies is to divide the genome into fixed- or variable-sized tiles and calculate the average methylation level across CpGs within each tile for each cell (Kremer et al. *Nature*, 2024; Kremer et al. *Nature Methods*, 2024; Liu et al. *Nature*, 2023; Fabyanic et al. *Nature Biotechnology*, 2024; Acharya et al. *Genome Biology*, 2024)¹⁰⁻¹⁴. For example, in a given cell, if five CpG sites are covered in a tile and four are methylated, the methylation level or methylation fractions for that region would be reported as 0.8. This results in a continuous-valued matrix, where rows correspond to cells and columns represent genomic tiles, with values ranging from 0 to 1.

In our study, we applied this strategy and adopted the Variably Methylated Regions (VMRs) framework, which was recently developed to capture more informative and biologically variable genomic regions (Kremer et al. *Nature Methods*, 2024)¹⁰. The methylation levels shown in figures such as Fig. 2c, 3f, and 4f–h represent the average methylation across all CpGs within each VMR for each pixel. This approach has also been applied in recent single-

cell methylation studies (Kremer et al. *Nature*, 2024; Kremer et al. *Nature Methods*, 2024)^{10,11}, demonstrating its ability to identify biologically meaningful regions involved in the core functions of different cell types. To clarify this point, we have updated the Methods section to include the source of non-binary continuous methylation readings and the rationale of the VMR-based quantification strategy. In the revised Methods section, we wrote “Due to the inherent sparsity of DNA methylation data, it is impractical to analyze methylation status solely at the individual CpG level. Binary information at sparse loci cannot be directly used to construct a feature matrix suitable for downstream analysis. In our study, we adopted the Variably Methylated Regions (VMRs) framework, which divides the genome into variable-sized tiles and calculate the average methylation level across CpGs within each tile for each pixel. This approach results in a continuous-valued matrix, where rows correspond to pixels and columns represent genomic tiles, with values ranging from 0 to 1.”

Comment 13: Although DNAm and RNA-based cell identity do not have a simple subset-superset relationship - but there must be biology behind the observation. For example, epigenetic primed sub-population but share similar transcriptional states, and sub-types of distinct transcriptional states not divided by DNAm (but other epigenetic mechanisms). From the confusion matrix looks there are quite a few differences between the two modalities. One possibility is RNA signal has much larger signal range (e.g., expression level from 0 to thousands of transcripts per cell) but DNAm is nearly binary. One suggestion is to highlight the advantage of using WNN for spatial-multiomics analysis - for example, can you estimate the fraction of cell types within the pixel based on WNN and independent DNAm/RNA clustering? The confusion matrix should be included in the manuscript, along with quantitative comparison between WNN-based clustering and single-modality clustering results, which will be helpful to the readers.

Response:

We thank the reviewer for this insightful comment regarding the differences in clustering between the RNA and DNA methylation modalities. We are indeed intrigued by the difference between the complementary roles DNA methylation and RNA expression play in jointly defining cells' identities and the possibility that epigenetic differences could have primed subsequent transcriptomic differences before they actually manifest. We appreciate the reviewer's further suggestion to highlight the differences between the two modalities in WNN analysis. In the revised manuscript, we have added this explanation: “This may reflect epigenetically primed subpopulations that share similar transcriptional states.” “This may reflect regulatory redundancy, where distinct epigenetic mechanisms converge to produce similar transcriptional outcomes. Conversely, the opposite scenario is equally plausible—transcriptional states may diverge despite similar epigenetic landscapes, as epigenetic regulation represents only one layer influencing gene expression.” Additionally, we further included the confusion matrix and alluvial plot (Extended Data Fig. 5e, f), which visualizes the correspondence between clustering results derived from RNA and DNA methylation. We also added a discussion noting that “RNA expression profiles exhibited a broader dynamic range, which may result in distinct clustering granularity across modalities.”

Extended Data Fig. 5e:

We further analyze the pixels shared and unique to each WNN-based cluster and single-modality cluster. We incorporated an alluvial diagram that illustrates the correspondence between clusters identified by different modalities (Extended Data Fig. 5f). The alluvial diagram demonstrates that WNN clustering incorporates complementary information from both modalities, resulting in refined cluster definitions.

Extended Data Fig. 5f:

Following the reviewer's suggestion, we also quantitatively assessed the relative contribution of each modality to WNN clustering by calculating modality weights for each pixel (Extended Data Fig. 5g). This analysis revealed that some WNN clusters are more

strongly driven by RNA signals (e.g., W6 – heart), while others have higher DNAm modality weights (e.g., W11 – craniofacial region). These findings demonstrate the complementary roles the two modalities play in shaping cells' identities. They also underscore the utility of spatial multi-omics integration in capturing cell type diversity that may be underrepresented when considering a single modality alone. The Results section has been updated to reflect these new analyses and interpretations.

Extended Data Fig. 5g:

References

- 1 Qiu, C. *et al.* A single-cell time-lapse of mouse prenatal development from gastrula to birth. *Nature* **626**, 1084-1093 (2024). <https://doi.org/10.1038/s41586-024-07069-w>
- 2 Zhang, D. *et al.* Spatial epigenome–transcriptome co-profiling of mammalian tissues. *Nature* **616**, 113-122 (2023). <https://doi.org/10.1038/s41586-023-05795-1>
- 3 Zeisel, A. *et al.* Molecular Architecture of the Mouse Nervous System. *Cell* **174**, 999-1014.e1022 (2018). <https://doi.org/10.1016/j.cell.2018.06.021>
- 4 Kleshchevnikov, V. *et al.* Cell2location maps fine-grained cell types in spatial transcriptomics. *Nature Biotechnology* **40**, 661-671 (2022). <https://doi.org/10.1038/s41587-021-01139-4>
- 5 Liu, H. *et al.* DNA methylation atlas of the mouse brain at single-cell resolution. *Nature* **598**, 120-128 (2021). <https://doi.org/10.1038/s41586-020-03182-8>
- 6 Shareef, S. J. *et al.* Extended-representation bisulfite sequencing of gene regulatory elements in multiplexed samples and single cells. *Nature Biotechnology* **39**, 1086-1094 (2021). <https://doi.org/10.1038/s41587-021-00910-x>
- 7 Hernando-Herraez, I. *et al.* Ageing affects DNA methylation drift and transcriptional cell-to-cell variability in mouse muscle stem cells. *Nature Communications* **10**, 4361 (2019). <https://doi.org/10.1038/s41467-019-12293-4>
- 8 Nichols, R. V. *et al.* High-throughput robust single-cell DNA methylation profiling with sciMETv2. *Nature Communications* **13**, 7627 (2022). <https://doi.org/10.1038/s41467-022-35374-3>
- 9 He, Y. *et al.* Spatiotemporal DNA methylome dynamics of the developing mouse fetus. *Nature* **583**, 752-759 (2020). <https://doi.org/10.1038/s41586-020-2119-x>
- 10 Kremer, L. P. M. *et al.* Analyzing single-cell bisulfite sequencing data with MethSCAN. *Nature Methods* **21**, 1616-1623 (2024). <https://doi.org/10.1038/s41592-024-02347-x>
- 11 Kremer, L. P. M. *et al.* DNA methylation controls stemness of astrocytes in health and ischaemia. *Nature* **634**, 415-423 (2024). <https://doi.org/10.1038/s41586-024-07898-9>
- 12 Liu, H. *et al.* Single-cell DNA methylome and 3D multi-omic atlas of the adult mouse brain. *Nature* **624**, 366-377 (2023). <https://doi.org/10.1038/s41586-023-06805-y>
- 13 Fabyanic, E. B. *et al.* Joint single-cell profiling resolves 5mC and 5hmC and reveals their distinct gene regulatory effects. *Nature Biotechnology* **42**, 960-974 (2024). <https://doi.org/10.1038/s41587-023-01909-2>
- 14 Acharya, S. N. *et al.* sciMET-cap: high-throughput single-cell methylation analysis with a reduced sequencing burden. *Genome Biology* **25**, 186 (2024). <https://doi.org/10.1186/s13059-024-03306-7>